# Titration of RAS alters senescent state and influences tumour initiation

Adelyne S. L. Chan[1,13], Haoran Zhu[1,13], Masako Narita[1], Liam D. Cassidy[1], Andrew R. J. Young[1], Camino Bermejo-Rodriguez[2], Aleksandra T. Janowska[1], Hung-Chang Chen[1], Sarah Gough[1], Naoki Oshimori[3], Lars Zender[4,5,6,7], Sarah J. Aitken[1,8,9], Matthew Hoare[1,10,11] & Masashi Narita[1,12] ✉

Oncogenic RAS-induced senescence (OIS) is an autonomous tumour suppressor mechanism associated with premalignancy[1,2]. Achieving this phenotype typically requires a high level of oncogenic stress, yet the phenotype provoked by lower oncogenic dosage remains unclear. Here we develop oncogenic RAS dose-escalation models in vitro and in vivo, revealing a RAS dose-driven non-linear continuum of downstream phenotypes. In a hepatocyte OIS model in vivo, ectopic expression of NRAS(G12V) does not induce tumours, in part owing to OIS-driven immune clearance[3]. Single-cell RNA sequencing analyses reveal distinct hepatocyte clusters with typical OIS or progenitor-like features, corresponding to high and intermediate levels of NRAS(G12V), respectively. When titred down, NRAS(G12V)-expressing hepatocytes become immune resistant and develop tumours. Time-series monitoring at single-cell resolution identifies two distinct tumour types: early-onset aggressive undifferentiated and late-onset differentiated hepatocellular carcinoma. The molecular signature of each mouse tumour type is associated with different progenitor features and enriched in distinct human hepatocellular carcinoma subclasses. Our results define the oncogenic dosage-driven OIS spectrum, reconciling the senescence and tumour initiation phenotypes in early tumorigenesis.

Senescence is a state of stable exit from the cell cycle with functional alterations, represented by an altered composite of secretory factors (senescence-associated secretory phenotype (SASP))[1,2,4]. This shift in cellular function can be in the form of loss, gain and/or augmentation. Cellular function is largely dictated by lineage-specific genes, and our recent studies have suggested that senescent cells adapt an epigenetic mechanism akin to terminal differentiation for altering lineage-specific gene expression[5,6]. This suggests that senescence is another layer of the dynamic fate-determination process, but how the senescence phenotype evolves is not entirely clear[7]. This idea is particularly relevant in OIS. RAS is frequently mutated in human cancer, but an oncogenic *RAS* allele alone is insufficient for cancer development; instead, a further increase in the activity of mutant RAS and its downstream effectors, such as the MAPK pathway, appears necessary[8–11]. Of note, OIS also requires excessive RAS activity[12]. The relationship between OIS and tumour initiation remains elusive, and we reasoned that it can be modelled by the phenotype conferred by a range of oncogenic RAS levels in a normal or non-transformed diploid cellular context.

## RAS dose and non-linear gene regulation

To test this, we first utilized a mouse liver model, which involves stable delivery of transposable elements containing oncogenic *NRAS^{G12V}* by hydrodynamic tail-vein injection (HDTVi), in which the transgenes are taken up by a subset of hepatocytes. These cells have been reported to become OIS by day 6 post-injection, which is followed by a CD4+ T lymphocyte-dependent and macrophage-dependent clearance of NRAS(G12V)-expressing cells by days 12–30 post-injection[3,13].

Immunohistochemical (IHC) analysis for RAS on day 6 post-injection, before immune clearance, demonstrated substantial heterogeneity in RAS intensity (Fig. 1a). Next, we asked whether this heterogeneity in NRAS dose translates to downstream transcriptomic differences at a single-cell level, we performed single-cell RNA sequencing (scRNA-seq) on flow-sorted mVenus (thus NRAS-mutant)-expressing hepatocytes on day 6 in control (non-oncogenic NRAS(G12V/D38A)) and experimental (NRAS(G12V)) mice. In *t*-distributed stochastic neighbour embedding (*t*-SNE) space by single-cell gene expression profile, control and a subset of experimental cells showed a good

[1]Cancer Research UK Cambridge Institute, Li Ka Shing Centre, University of Cambridge, Cambridge, UK. [2]Department of Molecular and Clinical Cancer Medicine, University of Liverpool, Liverpool, UK. [3]Department of Cell, Developmental and Cancer Biology, Knight Cancer Institute, Oregon Health and Science University, Portland, OR, USA. [4]Department of Medical Oncology and Pneumology, University Hospital Tuebingen, Tuebingen, Germany. [5]German Cancer Research Consortium (DKTK), Partner Site Tübingen, German Cancer Research Center (DKFZ), Heidelberg, Germany. [6]iFIT Cluster of Excellence EXC 2180 Image Guided and Functionally Instructed Tumor Therapies, University of Tuebingen, Tuebingen, Germany. [7]Tuebingen Center for Academic Drug Discovery and Development (TüCAD2), Tübingen, Germany. [8]Medical Research Council Toxicology Unit, University of Cambridge, Cambridge, UK. [9]Department of Histopathology, Cambridge University Hospitals NHS Foundation Trust, Cambridge, UK. [10]Early Cancer Institute, Hutchison Research Centre, University of Cambridge, Cambridge, UK. [11]Department of Medicine, University of Cambridge, Cambridge, UK. [12]Tokyo Tech World Research Hub Initiative (WRHI), Institute of Innovative Research, Tokyo Institute of Technology, Yokohama, Japan. [13]These authors contributed equally: Adelyne S. L. Chan, Haoran Zhu. ✉e-mail: Masashi.Narita@cruk.cam.ac.uk

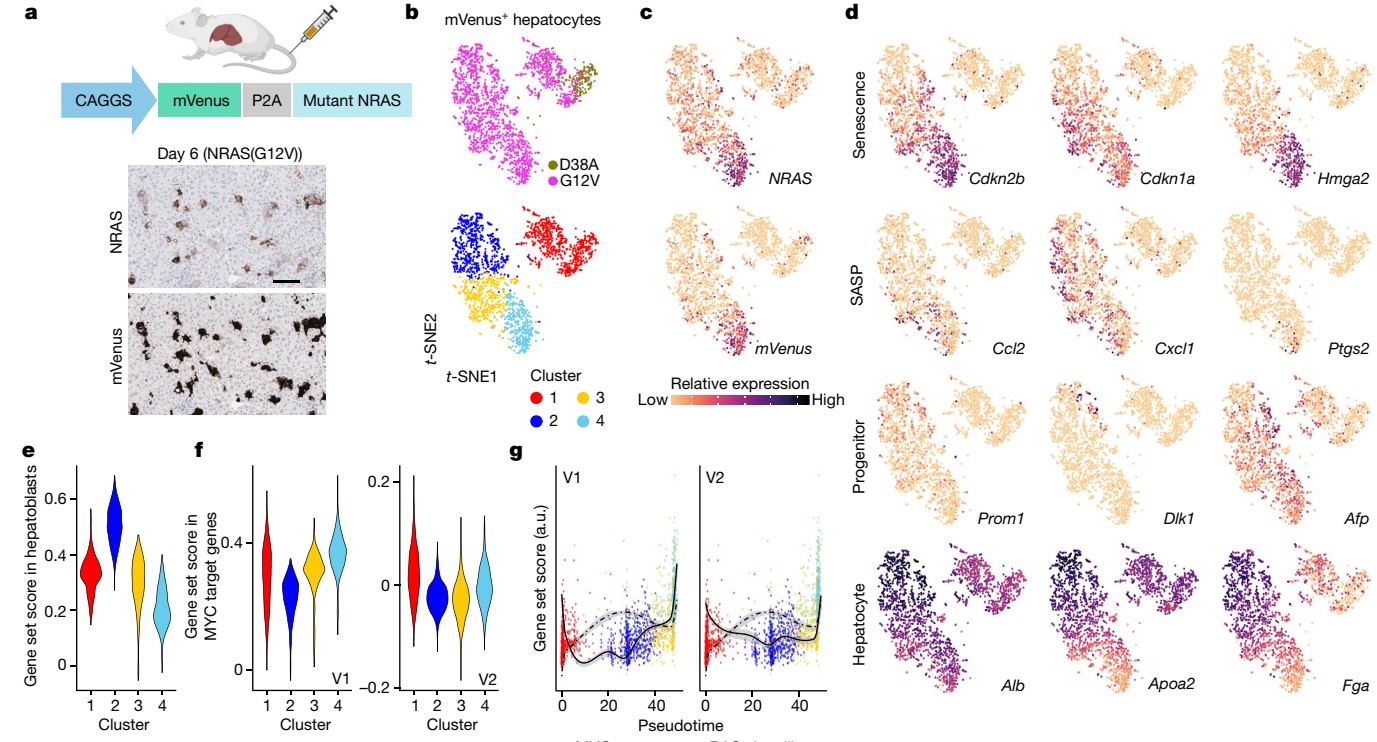

**Fig. 1 | Single-cell transcriptomics reveals OIS spectrum driven by oncogenic dosage in vivo. a**, Schematic of the HDTVi setup. IHC shows consequent heterogeneity in expression levels of ectopic NRAS(G12V) in experimental mice used for scRNA-seq. Scale bar, 100 μm. Schematic in **a** was created with BioRender.com. **b–d**, *t*-SNE embeddings of single-cell-sequenced hepatocytes (*n* = 2,179 cells from *n* = 2 NRAS(G12V) and *n* = 1 NRAS(G12V/D38A) mice), coloured by experimental condition: cluster (**b**), expression of *Nras* or *mVenus* (**c**) and selected genes (**d**) as indicated. **e,f**, Changes in expression of hepatoblast-associated signature (Descartes Cell Types and Tissue library, Enrichr; **e**) and two versions of MYC target genes (MSigDB Hallmark) across clusters (**f**). **g**, Correlation between expression levels of MYC (V1 and V2) and RAS signatures (KRAS_SIGNALLING_UP, MSigDB Hallmark) over pseudotime calculated with AddModuleScore in Seurat. The height of the dot indicates the curated gene set score derived from senescence-associated genes. a.u., arbitrary units.

separation from the rest of the experimental cells; overall, they formed four clusters by similarity of expression pattern (Fig. 1b). Both *NRAS* and *mVenus* expression increased across the clusters (Fig. 1c). Pseudotime analysis-exhibited progression of pseudotime values corresponded well with these cell clusters and *NRAS* was one of the top 50 hits driving the pseudotime, suggesting that NRAS dose is a primary driver of the observed clustering (Extended Data Fig. 1a–c). Genes associated with cell-autonomous effects of senescence, such as CDK inhibitors and chromatin modulators, tended to linearly correlate with the *NRAS* level, indicating that cluster 4 has typical OIS features, whereas SASP genes had a more heterogeneous expression pattern across clusters 3 and 4 (Fig. 1d). Using a previously annotated secretome gene set[13,14], these cell clusters with relatively high levels of NRAS expressed genes associated with the 'cytokine–cytokine receptor interaction' term in the Kyoto Encyclopedia of Genes and Genomes (KEGG) pathway database, including known SASP factors: *Il1a*, *Il1b* and *Ccl2* (Extended Data Fig. 1d). Both oncogenic stress and the SASP have been linked to the DNA damage response[15–17]. Although DNA damage-related gene sets were often higher in clusters 3 and 4, than in cluster 1 or 2, overall differences were modest (Extended Data Fig. 1e).

For a more unbiased view of gene expression differences across the clusters, we defined markers for each cluster and then performed pathway enrichment. Cluster 2 enriched for a hepatoblast signature, with upregulated progenitor genes represented by *Afp*, *Prom1* and *Dlk1* (Fig. 1d,e). This cluster also exhibited an upregulation of hepatocyte-specific markers, such as *Alb* (Fig. 1d), suggesting a functionally augmented state alongside the more progenitor state. The functional augmentation of hepatocytes is consistent with the secretome analysis, where cluster 2 was associated with the term 'complement and coagulation cascade' (Extended Data Fig. 1d). Of note, both progenitor and functionally augmented states exhibited a non-linear trend with increasing dose of *NRAS* (Fig. 1c,d). Using MSigDB[18,19] hallmarks, we found that MYC target genes were downregulated in cluster 2, whereas a subset of MYC targets was unchanged, or rather upregulated, in the OIS cluster 4 (Fig. 1f). Of note, MYC is a direct downstream transcription factor in the RAS–MAPK (ERK) pathway, where MYC is a nuclear substrate of ERK[20]. Signatures of other ERK substrate transcription factors or downstream kinases showed, unlike MYC targets, a largely linear upregulation along the cell clusters (Extended Data Fig. 1f).

To evaluate pathway-level changes along the pseudotime, we computed module scores for each cell between RAS and MYC signatures and found an overall negative correlation except for the OIS cluster 4, where it switched to a positive correlation in both MYC target gene sets, indicating a complex non-linear relationship between RAS and MYC signatures (Fig. 1g).

We then asked whether a similar dose-dependent trend exists in tissues expressing oncogenic RAS from the endogenous locus in a different premalignancy model. We used publicly available data[21] in a KRAS(G12D)-driven pancreatic tumour model, *Ptf1a-CreER;LSL-Kras-G12D;LSL-tdTomato* (PRT) mice, in which, upon 4-hydroxytamoxifen (4-OHT) administration, acinar cells are genetically labelled (with tdTomato) and express oncogenic *Kras^G12D* from the endogenous locus[21]. In this dataset, mice were sampled at different timepoints encompassing different disease stages, in which OIS was previously implicated in pancreatic intraepithelial neoplasia[22–24], and, consistently, a *Cdkn2a^+* (encoding p16) OIS cluster has been identified by the original authors.

We reanalysed this scRNA-seq data, focusing on the oncogenic *Kras*-expressing *tdTomato^+* cluster, and first located this cluster of

OIS cells (Extended Data Fig. 2a,b). Consistent with the idea that both senescence and tumorigenesis require a high level of oncogenic RAS[11,12], spontaneous upregulation of *Kras* level was detected with disease stage, in which the OIS cluster exhibited higher expression than both their non-senescent pancreatic intraepithelial neoplasia counterparts and the more advanced pancreatic ductal adenocarcinoma stage (Extended Data Fig. 2c–e). The spontaneous increase in oncogenic RAS expression during tumorigenesis was also supported by the analysis of The Cancer Genome Atlas (TCGA) datasets[25], in which *RAS* transcript in human pancreatic ductal adenocarcinoma (*KRAS*) and other types of tumours tended to be higher in tumours with mutant *RAS* than with wild-type *RAS* (Extended Data Fig. 2f). *KRAS* upregulation in cancer cells was also found in public scRNA-seq data in human pancreatic and lung cancer[26,27] (Extended Data Fig. 2g,h). Furthermore, although levels of some progenitor markers, such as *Prom1*, *Pdx1* and *Notch1*, were upregulated during tumorigenesis, this upregulation was weaker in the OIS cells than in cells in the same stage (Extended Data Fig. 2e), suggesting an inverse correlation between the progenitor and senescent states. MYC basal levels in control cells were generally low in the pancreas but, similar to the liver model, a subset of MYC targets were higher in the OIS cluster (Extended Data Fig. 2d). Together, these results suggest that oncogenic RAS provokes a dose-dependent, non-linear spectrum of phenotype in preneoplastic conditions.

To systematically explore the response to differing levels of RAS in a more homogeneous manner, we developed an in vitro and in vivo system for titrating down the dose of ectopic RAS expressed in cells. For both, we co-expressed the fluorescent marker mVenus and a mutant RAS on the same open reading frame, separated by P2A, a self-cleaving peptide that mediates co-translational cleavage into the constituent proteins.

## RAS triggers slow-cycling RPE1 cells

In vitro, we chose RPE1 cells, an hTERT-immortalized diploid epithelial cell line of human retinal pigment origin, because they are resistant to flow-sorting stress, yet maintain a diploid karyotype[28,29]. We used a predictive form of our reporter construct; although mVenus is constitutively expressed, mutant HRAS is introduced in the form of an inducible ER–HRAS(G12V) fusion protein, which is only stabilized upon 4-OHT administration[26] (Fig. 2a). We then added 4-OHT to induce HRAS(G12V) and sampled cells for analysis by flow cytometry at defined timepoints post-HRAS(G12V) induction. Population fluorescence intensity gradually shifted towards a distinct peak, corresponding to a relatively low level of mVenus, suggesting that this level of HRAS(G12V) provides the optimal selective advantage in this RPE1 cell system (Fig. 2b). This provides direct evidence for non-linear dose-dependent effects of oncogenic RAS on non-transformed cells in culture[12,30].

This system permits sorting of this heterogeneous cellular population into highly homogeneous subpopulations differing in the expression level of mVenus before inducing HRAS(G12V) (Fig. 2c). The HRAS(G12V)-induced phenotype was characterized in four subpopulations, selected to maximize separation between them (denoted 'S', 'M', 'L' and 'XL' to indicate increasing mVenus intensities) and plain RPE1 cells ('N' denotes no *mVenus-P2A-ER–HRAS^G12V* transduction; Fig. 2c,d). We first validated that this separation is stable in long-term culture (Extended Data Fig. 3a). The low-RAS 'S' subpopulation remained proliferative with no significant increase in senescence-associated β-galactosidase (SA-β-gal) activity after HRAS(G12V) induction. By contrast, higher HRAS(G12V)-expressing subpopulations ('M', 'L' and 'XL') exhibited a significant increase in SA-β-gal activity and reduction in cell-cycle progression compared with matched uninduced control cells (Fig. 2e,f and Extended Data Fig. 2b).

Of note, despite this dose-dependent decrease in proliferative capacity, a substantial number of BrdU-positive cells remained in the high HRAS(G12V)-expressing subpopulations (Fig. 2f). The existence of OIS escapers within a population would lead to their eventual grow-out in a heterogeneous context, but this property is not expected in sorted subpopulations; indeed, we observed no sign of eventual grow-out in (X)L cells. To assess the fate of these residual BrdU-incorporating XL cells on day 6 post-induction, using membrane-permeable Hoechst-33342 quantification of DNA content as a proxy of cell-cycle phase in live cells, we flow-sorted cells of the S and XL subpopulations that were in mid-S phase on day 6 post-induction, returned them to culture and reassessed their phenotype 3 days later (in the presence of 4-OHT throughout; Fig. 2g). As expected, S cells showed a slight increase in BrdU incorporation, probably due to a synchronization effect (Fig. 2h). However, in XL cells, there was no such increase but rather a slight decrease in the number of BrdU-positive cells. These cells stained positive for IL-8 (Extended Data Fig. 3c), demonstrating that they remain functionally viable. The data reinforce that the OIS-like state with reduced, but not complete loss of, proliferative capacity is stable and that the slow-cycling state is not due to proliferation of a rare subset of cells. We conducted similar experiments in TIG3 human diploid fibroblasts. In a mixed population of TIG3 cells with a wide range of HRAS(G12V) levels, the survival benefit of the low-RAS TIG3 cells was recapitulated, and high-RAS TIG3 cells showed senescence-like phenotype, including reduced proliferation, increased SA-β-gal activity and upregulation of the SASP components IL-6 and IL-8 (Extended Data Fig. 3d–f). Of note, an increased DNA damage response (a classic senescence marker), probed by phosphorylated H2AX (γH2AX) immunostaining, in high-RAS-expressing TIG3, but not RPE1, cells was detected, supporting the slow-cycling nature of the RPE1 system (Extended Data Fig. 3g).

To further characterize the sorted subpopulations in the RPE1 system, we performed RNA-seq analysis, pre-induction and on day 6 post-induction. Principal component analysis demonstrated that the induced subpopulations were transcriptionally distinct from one another (Extended Data Fig. 4a). Pathway enrichment analysis of differentially expressed (FDR < 0.05, |log fold change| > 1.2) genes showed increased numbers of pathway terms associated with higher HRAS(G12V) subpopulations (Extended Data Fig. 4b,c), particularly pathways related to the inflammatory response, largely driven by genes encoding well-described SASP factors[31–33], although not all other classical OIS markers, including *Cdkn2a* (encoding p16), were upregulated, even in the XL cells (Fig. 2i). Among the MSigDB hallmark gene sets[19] (Extended Data Fig. 4d), reduction of MYC and cell-cycle signatures represented the most notable changes in each subpopulation, including S cells, albeit more modestly (Fig. 2j,k), suggesting that the survival benefit observed in S cells in a heterogeneous population does not merely reflect their better growth capacity. Other RAS–MAPK substrate transcription factors examined failed to show such reduced activity in RAS-expressing RPE1 cells (Extended Data Fig. 5a). This unique suppression of MYC activity is unlike typical OIS cluster 4 of the liver dataset, but rather reminiscent of the progenitor-like cluster 2 (Fig. 1e–g). Indeed, publicly available data generated from OIS fibroblasts (Supplementary Table 2) also showed globally intact or often an increased MYC signature (Extended Data Fig. 5b). We also performed scRNA-seq analysis in these RPE1 subpopulations (*n* = 2) and found that, in *t*-SNE space, RAS signalling and the cell-cycle profile were orthogonal, in which MYC signatures appeared inversely correlated with the former; thus, the negative correlation between RAS and MYC signatures in RPE1 cells was not simply due to reduced cell proliferation (Extended Data Fig. 5c,d). Furthermore, markers of neural progenitors, which are RPE precursors, *NES* and *PAX6* were upregulated with RAS induction, whereas a number of RPE differentiation markers were downregulated, although some RPE-functional genes, such as *BEST1*, were upregulated (Fig. 2l). Thus, similar to subsets of oncogenic RAS-expressing cells in vivo (for example, cluster 2; Fig. 1b), in RPE1 cells, oncogenic RAS promotes a unique progenitor-like state, which we postulated is a part of the OIS spectrum.

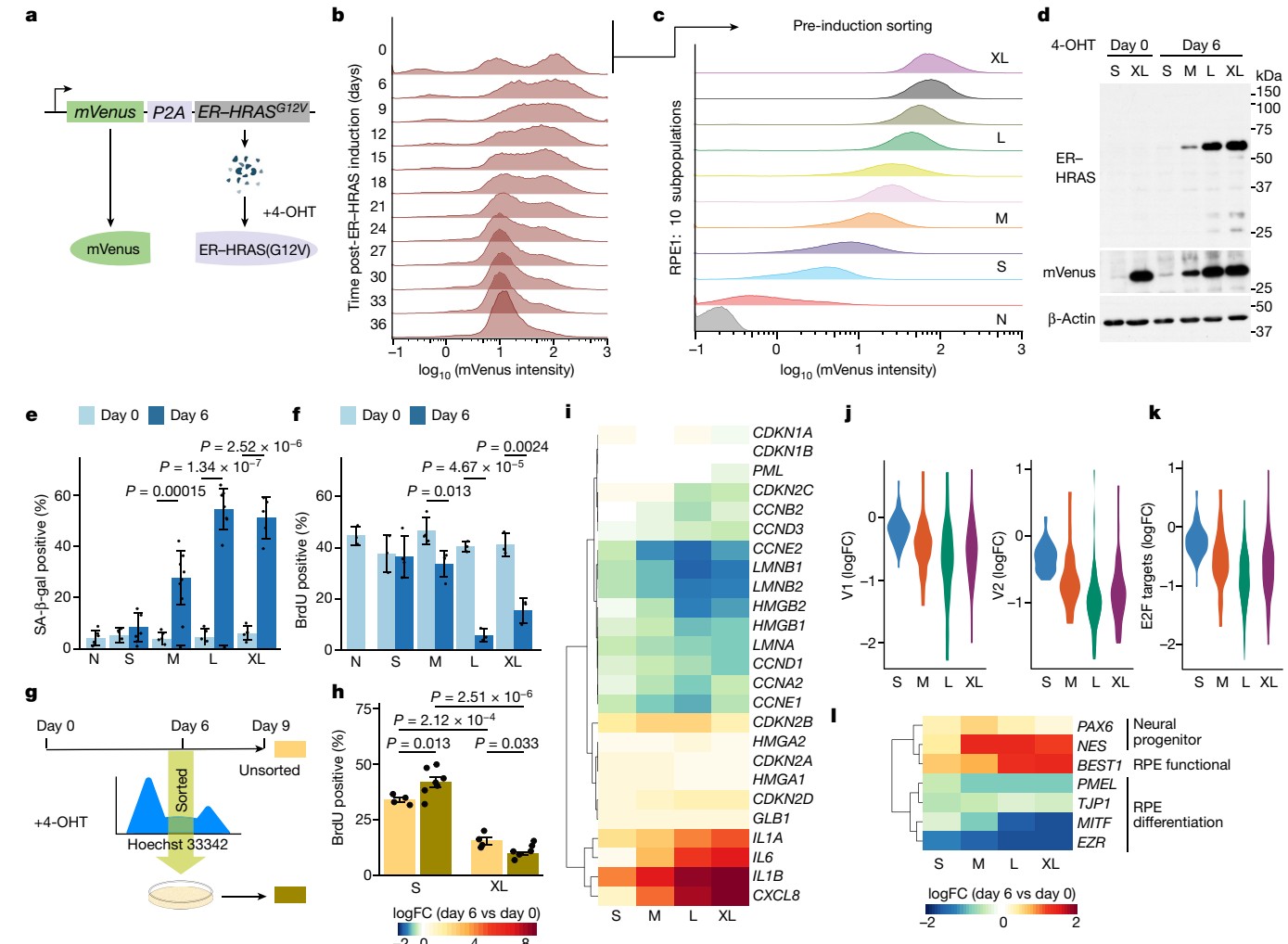

**Fig. 2 | Oncogenic RAS induces OIS-like slow-cycling phenotype in RPE1 cells. a**, The 'predictive reporter' system. Schematic in **a** was created with BioRender.com. **b**, Distribution of mVenus intensity over time by flow cytometry for a population of cells heterogeneously expressing the construct. **c**, mVenus intensity of subpopulations established by flow sorting without RAS induction. **d**, Western blotting for the indicated proteins in the sorted subpopulations pre-induction and on day 6 post-induction with 4-OHT. **e,f**, Senescence phenotype of the sorted subpopulations assessed by SA-β-gal positivity (**e**) and BrdU incorporation (**f**). From left to right, *n* = 6, 5, 5, 6, 8, 5, 7, 5 and 5 (**e**) and *n* = 3, 3, 4, 4, 4, 3, 3, 3 and 3 (**f**) independent experiments.

**g**, Flow-sorting experiment to enrich for cells in the S phase on day 6 post-induction. **h**, BrdU positivity for sorted versus unsorted cells on day 9 post-induction. From left to right, *n* = 4, 7, 4 and 6 independent experiments. FC, fold change. **i–l**, Differential expression of senescence-associated genes (**i**), MYC (**j**) and E2F target (**k**) gene sets, and RPE-associated genes (**l**) in the different subpopulations against matched control cells. Senescence-associated genes (**i**) were manually curated from pathway databases shown in Extended Data Fig. 4c. The scores for all hallmarks are in Extended Data Fig. 4d. Error bars denote s.d. (**e,f,h**). Statistical significance was determined using two-way Student's *t*-test with no correction for multiple testing.

## Tumour initiation by sub-OIS RAS dose

Such an overlapping feature of 'OIS intermediates' with increased progenitor markers and reduced levels of MYC targets is reminiscent of recently identified tumour-initiating cells (TICs), which are characterized by a TGFβ-responding slow-cycling state in a mouse model of ectopic HRAS(G12V)-driven early squamous cell carcinoma[34]. We reanalysed RNA-seq datasets derived from this mouse model and found a lower level of MYC and E2F targets in TICs than in the rest of the tumour cells, a trend that was also unique to MYC among the downstream transcription factors of the RAS–MAPK pathway examined (Extended Data Fig. 5e). Furthermore, similar to the pancreas, TCGA analysis[25] suggests a spontaneous upregulation of *RAS* in oncogenic RAS-driven head and neck squamous cell carcinoma in humans (Extended Data Fig. 2d).

To directly investigate the long-term implications of sub-OIS dosage oncogenic RAS in vivo, we applied our dose-titrating strategy in the mouse liver model. For this, we expressed the *mVenus-P2A-NRAS^G12V* construct under different promoters (Fig. 3a). We first validated

this dose difference by IHC analysis for RAS on day 6 post-injection; compared with the original strong promoter, CAGGS, the weaker PGK and UBC promoters resulted in lower and more homogenous expression levels of mutant NRAS (CAGGS > PGK > UBC; Fig. 3b). We assessed γH2AX-positive DNA damage foci and, in line with the subtle changes at the transcriptomic level (Extended Data Fig. 1e), found no significant increase in the frequency of cells with DNA damage foci in NRAS(G12V)-expressing hepatocytes at day 6, although more comprehensive measurements are still required (Extended Data Fig. 6a). Consistent with previous studies[4,15], NRAS-expressing cells were cleared by approximately days 12–30 post-injection in the *CAGGS-NRAS^G12V* mice (Fig. 3b). However, such senescence surveillance was weaker or absent in *PGK-NRAS^G12V* or *UBC-NRAS^G12V* mice, respectively, leading to persistent immune cell clusters around NRAS-expressing hepatocytes beyond day 12 (Fig. 3b and Extended Data Fig. 6b,c). Mice injected with *PGK*-driven or *UBC*-driven *NRAS^G12V* developed liver tumours with nearly 100% penetration (19 out of 20 mice) by 300 days post-HDTVi (Fig. 3c). By contrast, there was no tumour growth in mice injected

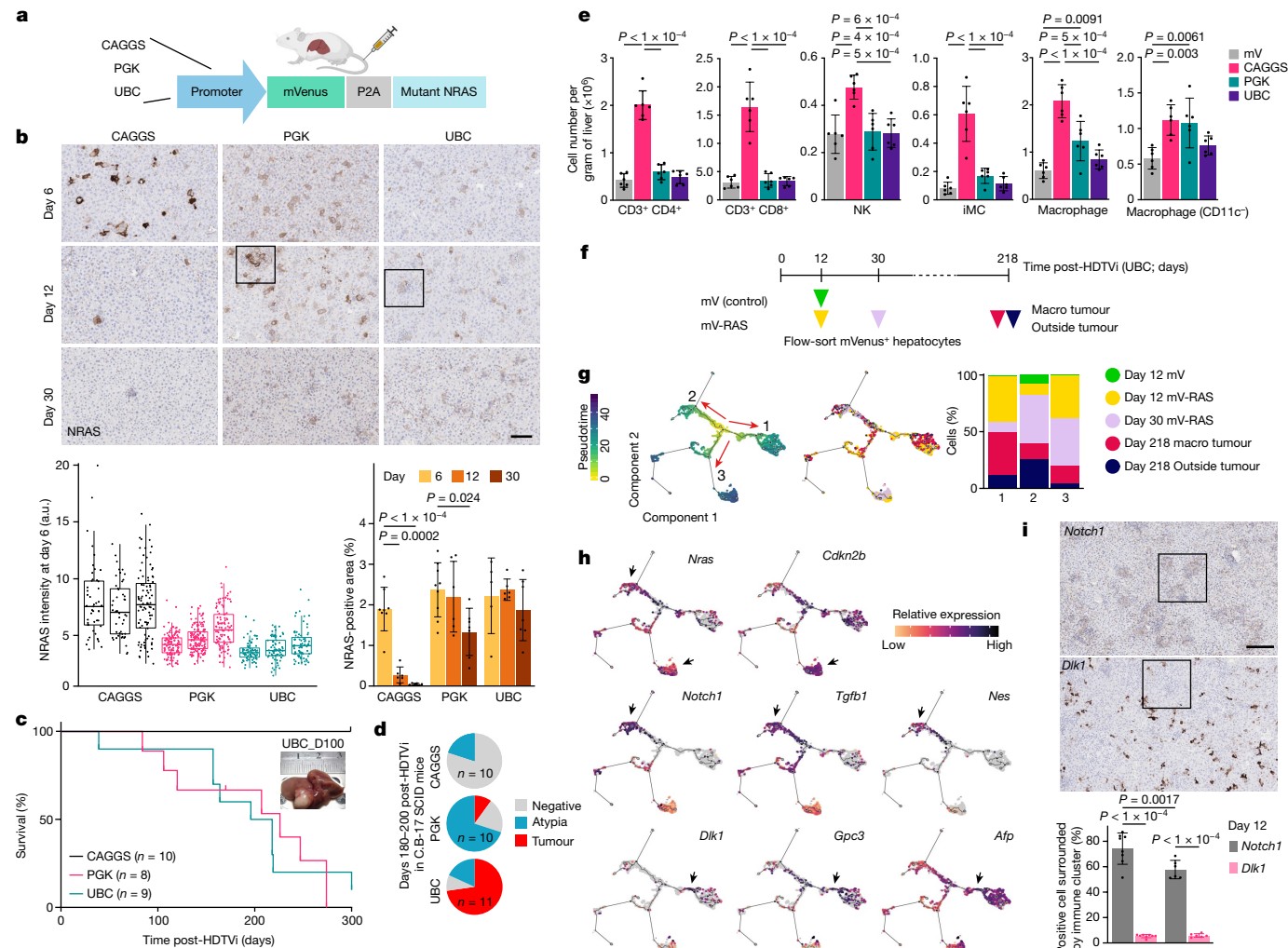

**Fig. 3 | Sub-OIS dosage of RAS is sufficient for tumorigenesis. a**, Schematic of experimental setup titrating down the dose of RAS introduced by HDTVi. The schematic in **a** was created with BioRender.com. **b**, NRAS IHC (top), quantification of NRAS intensity (three independent livers per condition; bottom left) and the percent of positive areas (bottom right). Magnified images are shown in Extended Data Fig. 6b. Scale bar, 100 μm. The box plot centre line indicates the median, the box limits indicate the first and third quartiles, and the whiskers indicate the largest values within 1.5 times the interquartile range. For the percent area, values are mean ± s.d. Two-way analysis of variance (ANOVA) with multiple comparisons followed by post hoc *t*-test with Bonferroni correction were used to determine significance. *n* = 8, 6, 6, 9, 6, 7, 5, 6 and 7 mice. **c**, Kaplan–Meier analysis for mice injected with the different plasmids. **d**, Tumour incidence in SCID mice injected with the indicated plasmids. *n* denotes the number of mice (**c**,**d**). **e**, Cell number per gram of liver for the indicated immune cell types (*n* = 6 mice per condition).

Values are mean ± s.d. One-way ANOVA followed by Tukey's honest significant difference test was used to determine significance. mV, mVenus; NK, natural killer cell. **f**, Experimental setup for scRNA-seq of mVenus-expressing hepatocytes (*n* = 2 per condition, 4,039 hepatocytes total). mVenus-expressing cells are derived from day 12 control (green), day 12 (yellow) and day 30 (purple) *UBC-NRAS^{G12V}*, macro-tumour (red) and outside tumour (dark blue). **g**,**h**, Pseudotime projection (left), coloured by sample of origin (right; **g**) and indicated genes of interest (**h**). The bar plot in **g** shows the percentage of cells from each sample, in each of the three pseudotime branches. The arrows in **h** indicate cell clusters expressing senescence-related (top) or progenitor-related genes. **i**, Percentage of *Notch1^+* or *Dlk1^+* hepatocytes within persistent immune cell clusters. Values are mean ± s.d. Two-way mixed-effects ANOVA followed by post hoc *t*-tests with Bonferroni correction were used to determine significance. *n* = 8, 8, 6 and 6 mice. Scale bar, 200 μm. Magnified images of the indicated areas are in Extended Data Fig. 8c.

with *CAGGS-NRAS^{G12V}* (Fig. 3c). Senescence surveillance in this context has previously been reported to depend on an intact CD4+ T cell and bone marrow-derived macrophages[3]. To focus on the cell-autonomous aspect of those RAS-expressing cells, we repeated this long-term experiment in an immunocompromised context in SCID mice lacking the entire adaptive immune component and found that this dose dependency was maintained, with the largest fraction of tumours found in the *UBC-NRAS^{G12V}* mice (Fig. 3d). These data suggest that, in addition to attenuating immune-mediated clearance, low-dose oncogenic RAS promotes tumorigenesis through the acquisition of cell-autonomous alterations such as increased plasticity.

To gain mechanistic insights into the resistance of immune surveillance in *PGK-NRAS^{G12V}* or *UBC-NRAS^{G12V}* mouse livers, we conducted

immune cell profiling using flow cytometry. Consistent with previous studies[3], we detected a significant increase in the numbers of CD4+ and CD8+ T cells, natural killer (NK) cells (CD3− and NK1.1^{hi}), immature monocytes (iMCs; Ly6C^{hi}, F4/80^{low}, CCR2^{hi}, CD11b^{hi}, CD11c^{low} and Gr1^{low}) and macrophages (F4/80^{hi}, CCR2^{hi}, CD11b+ and CD11c^{low}) in the *CAGGS-NRAS^{G12V}* livers at day 9 (Fig. 3e). However, such immune-cell recruitment was minimal in *PGK-NRAS^{G12V}* or *UBC-NRAS^{G12V}* livers. This is consistent with our secretome analysis, which suggested a weaker cytokine signature in hepatocyte clusters with lower NRAS(G12V) expression (Fig. 1d and Extended Data Fig. 1d), including *Ccl2*, which is required for recruitment of iMCs and thus senescence surveillance in this liver OIS model[35]. In addition, our recent study has shown that *Ptgs2* (encoding COX2) is also critical for senescence surveillance in

this model[36]. COX2 is an enzyme involved in the generation of prostaglandins, modulating the inflammatory SASP[36–38], and loss of *Ptgs2* promotes accumulation of immunosuppressive regulatory T ($T_{reg}$) cells in *CAGGS-NRAS^{G12V}* livers[36–38]. At the single-cell level, *Ptgs2* was only detected in the cluster 4 (OIS) hepatocytes at day 6 (Fig. 1d), and, consistently, we found a progressive accumulation of $T_{reg}$ cells in *PGK-NRAS^{G12V}* and *UBC-NRAS^{G12V}* livers (Extended Data Fig. 6d). These results suggest that insufficient activation of SASP regulators in hepatocytes that exhibit lower *NRAS^{G12V}* expression might in part contribute to their immune resistance.

To capture these dynamic changes during tumorigenesis, we performed scRNA-seq on flow-sorted hepatocytes from *UBC-NRAS^{G12V}* mice euthanized at different timepoints post-HDTVi (Fig. 3f). In the *t*-SNE space, there were two distinct clusters expressing a relatively high level of *NRAS* (Extended Data Fig. 7a). One of these, consisting of early timepoint cells, expressed markers of senescence including *Cdkn2b* (encoding p15), consistent with an OIS cluster. This cluster also showed elevated expression of MYC targets, reinforcing the positive correlation between RAS and MYC signatures in the OIS state (Extended Data Fig. 7b). The other high-NRAS cluster, which included tumour cells, exhibited elevated *Notch1* and TGFβ signalling (Extended Data Fig. 7a,b). Consistently, we and others have previously shown that NOTCH and TGFβ signalling is dynamically activated during OIS[13,39] and that co-introduction of NRAS(G12V) and a constitutively active form of NOTCH1 (intracellular domain; N1ICD) leads to liver tumour development in mice[13]. Within the population of lower-NRAS cells, we also identified a small cluster of cells, highly enriched for markers of hepatoblasts, such as *Dlk1* and *Afp*, with prominent upregulation of hepatocyte markers, such as *Alb* (Extended Data Fig. 7a).

When projected into a pseudotime, cells were mainly arranged into three developmental branches (Fig. 3g): two corresponded to the OIS (branch 3) and *Notch1* (branch 2) clusters described above, and the other branch contained the highest proportion of tumour cells, which expressed high *Afp* (branch 1; Fig. 3g,h). Serum AFP is a widely used hepatocellular carcinoma (HCC) biomarker and, in the *Afp^{hi}* branch 1, we identified an intermediate cellular state, largely corresponding to the *Dlk1^+/Gpc3^+* 'hepatoblastic cluster' (Fig. 3h, bottom). Upregulation of DLK1 and GPC3 has been associated with HCC[40,41]. Thus, the *Dlk1^+/Gpc3^+/Afp^+* progenitor-like cells potentially represent a tumour-initiating state for branch 1 tumours. These distinct clusters were also recapitulated by other trajectory inference methods (Extended Data Fig. 7c). Consistently, IHC analysis at early timepoints (days 6 and 9) identified a significantly higher *Dlk1^+* fraction of hepatocytes in the tumour-prone *PGK-NRAS^{G12V}* or *UBC-NRAS^{G12V}* mice than in *CAGGS-NRAS^{G12V}* mice (Extended Data Fig. 7d).

Although a large proportion of tumour cells were found along this *Dlk1/Gpc3/Afp^{hi}* branch, there were a small but substantial number of tumour cells along *Notch1/Tgfb1^{hi}* branch 2, which was characterized by another progenitor and stem marker: *Nes* (which encodes nestin; Fig. 3h, middle). Nestin has been implicated in undifferentiated liver tumorigenesis[42], thus we postulated that the *Nes^{hi}* cells along this branch represent a distinct population of TICs. This prompted us to re-evaluate the NRAS(G12V)-N1ICD-driven mouse liver tumour samples[13], and we found that all of these tumours stained positive for nestin ($n = 6$) and were poorly differentiated (Extended Data Fig. 8a). Although we observed that there were *Afp^+* hepatocytes in some lesions (in two of six mice), these were exclusive from the nestin^+ areas and showed barely detectable NRAS and NOTCH1 staining (Extended Data Fig. 8a, right), suggesting that they arose due to a local stress response or very low levels of the ectopic genes.

We next examined the spatial relationship of *Dlk1^+* or *Notch1^+* hepatocytes with immune cell clusters and found that most *Notch1^+* hepatocytes were within immune cell clusters, whereas *Dlk1^+* hepatocytes were largely excluded (Fig. 3i and Extended Data Fig. 8b,c).

Consistently, we previously showed that inhibiting Notch signalling in *CAGGS-NRAS^{G12V}*-expressing hepatocytes promotes OIS surveillance[13], suggesting that sustained activation of Notch signalling may also contribute to the resistance of NRAS(G12V)-expressing hepatocytes against immune surveillance. Then, we treated *UBC-NRAS^{G12V}* mice with sorafenib, an approved multi-kinase inhibitor that disrupts the RAS–MAPK cascade by targeting RAF and several upstream receptor tyrosine kinases[43]. At day 30 following *NRAS^{G12V}* transduction, as expected, lowering RAS–MAPK signalling did not affect immune surveillance, but the F4/80^+ macrophage aggregation, which was associated with *Notch1^+* hepatocytes, was reduced by the treatment (Extended Data Fig. 8d), further reinforcing the correlation between oncogenic RAS levels and their immunogenic activities in mouse livers.

## Dichotomous HCC tumour-initiating states

These results suggest that a modest level of oncogenic RAS leads to the development of liver tumours associated with at least two distinct tumour-initiating events. We next asked how we can translate this information to the tumours developed in *PGK-NRAS^{G12V}* or *UBC-NRAS^{G12V}* cohorts (Fig. 3c). Histologically, these tumours captured a wide range of histopathological differentiation (Fig. 4a), and the differentiation score was negatively correlated with the latency period (Fig. 4a,b). Tumours that developed early were predominantly undifferentiated (DS4), with pleomorphic tumour cells and sarcomatoid features. Although these tumours all stained positively for the biliary and progenitor marker CK19, they lacked specific histological features of cholangiocarcinoma (Extended Data Fig. 9a). By contrast, late-onset tumours were more well-differentiated HCC (Fig. 4b). Similar to NRAS(G12V)-N1ICD-driven tumours (Extended Data Fig. 8a), early-onset tumours with DS3–4 were mostly positive for nestin and NOTCH1, whereas the majority of late-onset differentiated tumours (DS1–2) were negative for nestin/NOTCH1 (Fig. 4a,b). Consistent with the scRNA-seq data (Fig. 3), the ectopic NRAS level tended to be higher in the NOTCH1/nestin^+ tumours (Fig. 4a and Extended Data Fig. 9b,c). By contrast, *Dlk1^+* cells were detected in all tumours irrespective of time of onset ($n = 15$) but retained their hepatocytic morphology and were spatially distinct from NOTCH1/nestin^+ regions, where *Dlk1^+* cells tended to exhibit lower NRAS expression (Extended Data Fig. 9b,c). This reinforces that although both types of TICs exist in the early stages, they develop tumours with different latency periods.

Together, our data suggest that, in the *PGK-* and *UBC-NRAS^{G12V}* models, a relatively high level of RAS can induce either senescence or a progenitor-like state (*Notch1* and *Nes*), the latter leading to aggressive undifferentiated tumours, whereas a low level of RAS induces a distinct progenitor-like state (*Dlk1*, *Gpc3* and *Afp*), developing more differentiated HCC with a longer latency period. We next investigated any relevance of our findings in human liver tumours. Two representative groups of human liver cancer cell lines have been proposed to mimic 'early-stage', well-differentiated (AFP^+) and 'late-stage', poorly differentiated (AFP^−) HCC, respectively[44,45]. Gene set enrichment analysis for tumour cells from each branch against those dichotomous datasets of human liver cancer cell lines[44,45] has revealed that genes driving the branch 1 and branch 2 tumours were significantly associated with well-differentiated and poorly differentiated states, respectively (Extended Data Fig. 9d). Similarly, when compared with previously defined human HCC subclasses[46], we observed a striking correlation: our branch 1 and branch 2 cells highly expressed genes associated with subclass S3 (well-differentiated HCC with better overall survival) and subclass S1 (typified by TGFβ and WNT activity), respectively, in a mutually exclusive manner (Fig. 4c). Next, we performed Kaplan–Meier analysis of patients with HCC from the TCGA dataset[25], comparing between patients in the top and bottom quartiles of expression levels for each of the subclass signatures. We found that we could improve

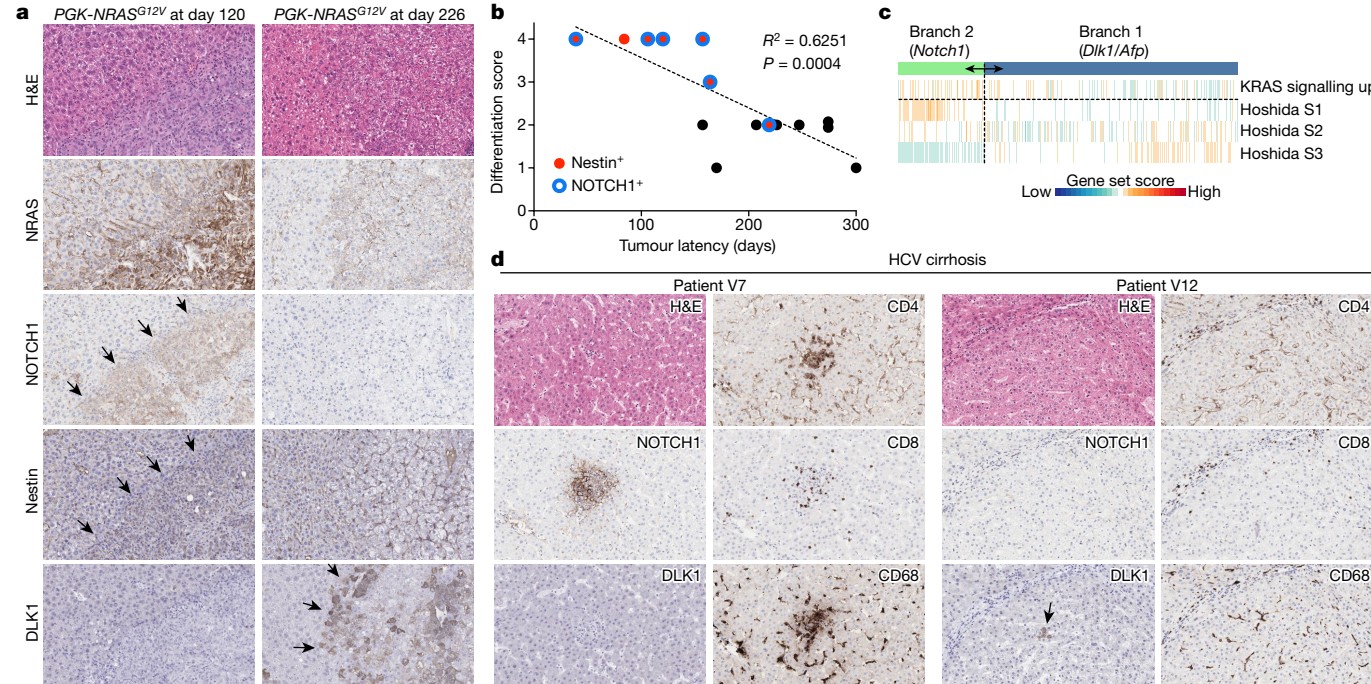

**Fig. 4 | Dichotomous *Dlk1*/*Afp*- and *Notch1*/*Tgfb1*/*Nes*-driven tumour-initiating events in mice and human HCC. a**, Representative haematoxylin and eosin (H&E) staining and IHC for the indicated proteins in undifferentiated, early-onset (day 120; left) and well-differentiated, late-onset (day 226; right) tumours in mice injected with *PGK-NRAS^G12V*. The arrows indicate areas positive for NOTCH1 and nestin (left) and DLK1 (right). Serial sections were used from 15 mice. **b**, Correlation between tumour latency (days) and differentiation score (from DS1 (well differentiated) to DS4 (undifferentiated)) in the PGK and UBC cohorts in Fig. 3c. Statistical significance and the strength of linear correlation between tumour latency and differentiation score were calculated using simple linear regression analysis. The dots are coloured by positivity for nestin and NOTCH1. Note that two mice at day 274 were scored as DS2. **c**, Two tumour branches correlate with distinct classes of human HCC. Gene set scores for the indicated human HCC gene signatures in tumour cells of the two branches are shown. **d**, Representative IHC for the indicated proteins in a patient with hepatitis C virus (HCV)-related liver cirrhosis showing that NOTCH1^+ hepatocytes were associated with immune cell clusters. Arrows indicate tumour borders. Serial sections were used for each sample. Total patients *n* = 28 (Extended Data Fig. 10a). Scale bars, 100 μm (**a**,**d**).

the diagnostic value of the subclass signatures, particularly in S1, by enriching for genes that we identified in branch 2 (or branch 1, for S3; Extended Data Fig. 9e).

Finally, we asked whether the distinct progenitor-like states identified in our scRNA-seq data could be detected in human liver cirrhosis, a major risk factor for liver tumour development (Supplementary Table 3). We identified positive DLK1 staining within the hepatocytes of 17 out of 28 cirrhotic human livers, whereas NOTCH1 staining was identified in 15 out of 28 cirrhotic human livers (Extended Data Fig. 10a). Of note, nine patients exhibited positive staining for both markers in spatially different regions (Extended Data Fig. 10b). Furthermore, NOTCH1^+ hepatocytes were invariably surrounded by immune cells, including CD68^+ myeloid cells, CD4^+ T cells and CD8^+ T cells; by contrast, NOTCH1^− cells did not evoke an immune response. These findings highlight that the two distinct molecular features of TICs identified in our mouse model may exist in human liver cirrhosis, both hepatitis C virus-related (Fig. 4d) and non-viral steatotic liver disease-related (Extended Data Fig. 10c).

We propose that our dose-titrating systems can model a non-linear OIS spectrum, including senescence intermediates such as slow-cycling (RPE1 cells) and immune-resistant tumour-initiating states (mouse livers), both characterized by increased progenitor features and a reduced MYC signature. The liver model provides insights into a RAS dose-associated evolution of senescence and immune microenvironment, revealing at least two distinct paths towards tumorigenesis in the liver: the *Dlk1*/*Afp* branch, corresponding to differentiated HCCs with longer latency, and the *Notch1*/*Tgfb1*/*Nes* branch, corresponding to undifferentiated tumours and associated with short latency and poor prognosis. These undifferentiated tumours were associated with

a relatively high level of oncogenic RAS activity, underscoring that oncogenic dosage is critical to define not only the senescence depth but also types of tumour-initiating states. The persistent immune cell clusters might also contribute to shaping a tumorigenic niche. Thus, beyond directly targeting specific TICs, modulating RAS–MAPK signalling or other crucial pathways at an early stage, such as NOTCH signalling, may have clinical relevance. Senescence is a dynamic process: at the end of the spectrum, OIS is a fate-determined state with tumour-suppressive properties, whereas more intermediate cellular states are associated with increased cell plasticity, a distinct immune reaction and a tumour-initiating capacity. Although our preclinical models are focused on young female mice, a separate long-term cohort in both sexes validated the similar tumorigenic activity of low-RAS expression in male mice (Extended Data Fig. 10d). A better understanding of specific TICs and their microenvironments, along with other factors such as sex, age and background chronic liver diseases, may offer therapeutic insights for early intervention in tumorigenesis.

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

## Methods

### Cell culture

hTERT-RPE1 cells (a telomerase-immortalized human retinal pigment epithelial cell line; CRL-4000, American Type Culture Collection) were grown in DMEM/F12/10% FCS, and TIG3 cells (a primary human embryonic lung fibroblast line; JCRB0506, JCRB Cell Bank)[47] were grown in DMEM/10% FCS in a 5% $O_2$/5% $CO_2$ atmosphere. Cells were obtained directly from the respective source cell banks. No authentication was performed by the authors of this paper. Cells were regularly tested for mycoplasma contamination. Of 4-OHT (H7904, Sigma), 100 nM was used for all ER–RAS induction experiments in vitro. Of etoposide (E1383, Sigma), 50 µM was used for the DNA damage experiments in RPE1 and TIG3 cells.

### BrdU incorporation and SA-β-gal assays

Cellular proliferation by BrdU incorporation and SA-β-gal analysis have been previously described[48]. RPE1 and TIG3 cells were incubated with BrdU for 2 h for the BrdU incorporation assay.

### Mice

HDTVi was performed as previously described[3]. In brief, at 6–8 weeks of age, 25 µg of appropriate vector and 5 µg of SB13 transposase-containing plasmid were diluted in sterile-filtered normal saline to a total volume of 10% of the body weight of the animal, before being injected into the lateral tail vein in under 10 s. Mice were randomized into control and experimental groups. C57BL/6 and Fox Chase SCID mice used in this study were purchased from Charles River. All mice used in these experiments were female, apart from the long-term monitoring cohort for identifying sex differences in tumour formation. All procedures were conducted in accordance with the UK Animal (Scientific Procedures) Act 1986, approved by the CRUK Cambridge Institute Animal Welfare and Ethical Review Body (AWERB) and conducted under the authority of the Project Licence number PP3912882.

Mice were housed in individually ventilated cages (Tecniplast) at a temperature of 19–23 °C, humidity of 45–65%, with up to 75 air exchanges per hour in the cages, and a 12–12-h light–dark cycle with the lights on at 07:00. The maximum caging density was five mice from the same litter and sex starting from weaning. As bedding, Aspen woodchip (Datesand) were provided. Mice were fed a standardized mouse diet LabDiet 5R58 breeding and maintenance diet or 5053 high-fat diet (IPS) and provided drinking water ad libitum. All materials, including individually ventilated cages, lids, feeders, bottles, bedding and water were autoclaved before use. Sentinel mice were negative for at least all Federation of Laboratory Animal Science Associations (FELASA)-relevant murine infectious agent as diagnosed by our health monitoring laboratory, Surrey Diagnostics.

### Tumour monitoring

The health of mice and impact of internal tumours were judged by external signs (for example, abdominal distension or weight gain exceeding 10% of normal body weight), clinical signs (for example, laboured breathing, rough hair coat, piloerection, inactivity, failure to eat or drink, fluid retention, neurological signs and digestive disturbances), aided by post-mortem assessment of morphological abnormalities in previously killed or deceased animals. To ensure early identification of health problems, animals with known or suspected pathologies received enhanced levels or surveillance (for example, hand checks). Primarily, mice were palpated, usually once a week, to detect the liver tumours. In the majority of cases, the liver tumours are detected before the development of clinical signs, and the animal was humanely culled by a schedule one method to alleviate any potential suffering. Occasionally, mice may develop clinical signs, as above, and were culled by a schedule one method to alleviate any further potential suffering. Limits specified by the project license were not exceeded in any of the experiments conducted.

### Plasmids

Predictive reporter plasmids for the in vitro experiments: *NLS-mVenus-P2A-ER–RAS* on either the pLNCX2 (retroviral, Clontech) and the pRRL.SIN-18 (lentiviral, described in ref. 49) backbones. The nuclear localization signal on all of these constructs is derived from SV40 large T-antigen (PKKKRKV). Plasmids for HTVIs: pPGK-SB13; pT/CAGGS-*NRAS^{G12V}-IRES-mVenus*, pT/CAGGS-*NRAS^{G12V/D38A}-IRES-mVenus*[15], pT/CAGGS-*mVenus-P2A-NRAS^{G12V}*, pT/PGK-*mVenus-P2A-NRAS^{G12V}*, pT/UBC-*mVenus-P2A-NRAS^{G12V}* and *UBC-mVenus-P2A*.

### Single-cell immune suspensions

Dissected livers were homogenized (130-105-807, Miltenyi Liver Dissociation Kit) and passed through a 70-µm filter. After centrifugation, samples were washed twice in PEB buffer (PBS, 5 µM EDTA and 0.5% BSA). Immune cells were enriched using an OptiPrep gradient (07820, STEMCELL Technologies). Immune cells along the gradient interphase were washed and resuspended in FACS buffer (PBS, 5 mM EDTA and 5% BSA) and individually placed within a 96-well round-bottomed tissue culture plate. Pellets were incubated with TruStain FcX Fc-blocking solution (101319, BioLegend) and then treated with cell-surface panels of fluorophore-conjugated antibodies: (1) CD45–BV510 (563891, BD), CD3–AF647 (100209, BioLegend), CD4–BUV496 (612952, BD), CD8a–BV711 (100747, BioLegend) and NK1.1–BV421 (108731, BioLegend); (2) CD45–BV510 (563891, BD), CD11b–Super Bright 645 (64-0112-82, eBioscience), CD11c–BV421 (117329, BioLegend), Ly6C–PerCP-Cy5.5 (128011, BioLegend), F4/80–PE-Cy7 (123113, BioLegend), Gr-1–FITC (108405, BioLegend), CCR2–BV785 (150621, BioLegend), MHC-II–Spark UV 387 (107670, BioLegend) and PDL1–APC (124312, BioLegend). The samples of all flow cytometric studies were incubated with a Fixable Viability Dye eFluor 780 (65-0865-14, eBioscience). Stained cells were analysed using an LSRFortessa Cell Analyzer (BD), and acquired results were analysed using FlowJo software (v10.9.0, FlowJo, BD). AccuCheck Counting Beads (PCB100, Invitrogen) were used for absolute cell number assessment.

### Flow cytometry

mVenus quantification was performed using a MACSQuantVYB (Miltenyi Biotech) flow cytometer. When DNA content quantification was required, Hoechst 33342 (stock 10 µg ml$^{-1}$) was added to the media of adherent cells in culture to a final concentration of 1 ng ml$^{-1}$. Cells were incubated on Hoechst-containing medium for 45 min before analysis.

Intrahepatic immune cells were prepared as above and then run on a BD Fortessa flow cytometer (Becton Dickinson); data were analysed using FlowJo v10. The gating strategy is provided in the Supplementary Information.

### Protein expression by immunoblotting and immunofluorescence

Immunofluorescence and immunoblotting, on SDS–PAGE on gels of various concentrations, were performed as previously described[48].

The primary antibodies (and their dilutions) for immunoblotting included: anti-β-actin (A5441, Sigma; AC15, mouse monoclonal, 1:5,000); anti-HRAS (sc29, Santa Cruz Biotechnology; F235, mouse monoclonal, 1:1,500); anti-GFP (632377, Clontech; rabbit polyclonal, 1:1,000); anti-IL-6 (MAB2061, R&D Biosystems; clone #1936, mouse monoclonal, 1:250); anti-IL-8 (MAB208, R&D Biosystems; clone #6217, mouse monoclonal, 1:500); anti-cyclin A (c4710, Sigma; CY-A1, mouse monoclonal, 1:1,000); and anti-p21 (sc-6246, Santa Cruz; F5, mouse monoclonal, 1:1,000). The primary antibodies (and their dilutions) for immunofluorescence included: anti-IL-8 (MAB208, R&D Biosystems; clone #6217, mouse monoclonal, 1:250); anti-BrdU (555627, BD Biosciences; 3D4, 1:500); and anti-phospho-histone H2A.X (Ser139) (05-636, Merck; JBW301, mouse monoclonal, 1:200, pH 8.0 for formalin-fixed paraffin-embedded sections).

The secondary antibody used was goat anti-mouse IgG (Alexa Fluor 555, 1:1,000; A-11034, Thermo Fisher) in PBS-T. Cells were counterstained with DAPI at 1 μM in the secondary antibody solution. Fluorescence images were obtained using Leica DMI6000B epifluorescence light microscope or Leica Stellaris 8 confocal microscope, using LAS X software versions 3.7.5.24914 or 4.7.0 (Leica), respectively. Uncropped immunoblot images can be found in the Supplementary Information.

## IHC

Formalin-fixed paraffin-embedded mouse and human tissues were stained with the primary antibodies listed at the concentrations below, after heat-induced epitope retrieval in citrate (pH 6) or Tris-EDTA (pH 9) buffers before visualization manually using the ImmPRESS IHC detection kit according to the manufacturer's instructions and counterstaining with haematoxylin. Alternatively, automated chromogenic immunohistochemical staining was performed on a Leica Bond Max (Leica) using the polymer refine detection and refine red detection kits (Leica). All tissue sections were scanned on a Leica AT2 at ×20 or ×40 magnification and a resolution of 0.5 μm per pixel.

The following primary antibodies (and their dilutions) were used: anti-GFP (ab13970, Abcam; chicken polyclonal, 10 μg ml$^{-1}$, pH 6.0); anti-RAS (ab52939, Abcam; EP1125Y, rabbit monoclonal, 1:1,000, pH 6.0); anti-p-ERK1/2 (9101, Cell Signaling Technology; rabbit polyclonal, 1:800, pH 6.0); anti-CK8 (MABT329, DSHB; TROMA-1, rat monoclonal, 2.98 μg ml$^{-1}$); anti-CK19 (MABT913, DSHB; TROMA-III, rat monoclonal, 0.058 μg ml$^{-1}$); anti-mouse nestin (MAB353, Chemicon; rat-401, mouse monoclonal, 1:200, pH 6.0); anti-human nestin (MAB5326, Chemicon; 10C2, mouse monoclonal, 1:120, pH 6.0); anti-AFP (sc-8399, Santa Cruz; C3, mouse monoclonal, 1:50, pH 6.0); anti-mouse DLK1 (FAB8634T, R&D Systems; 1168B, rabbit monoclonal, 1:200, pH 9.0); anti-human DLK1 (MAB1144, R&D Systems; 211309, mouse monoclonal, 4 μg ml$^{-1}$, pH 9.0); anti-NOTCH1 (3608, Cell Signaling Technology; D1E11, rabbit monoclonal, 1:200, pH 6.0); anti-TGFβ (3709, Cell Signaling Technology; 56E4, rabbit monoclonal, 1:100, pH 6.0); anti-mouse CD4 (ab183685, Abcam; EPR19514, rabbit monoclonal, 0.3205 μg ml$^{-1}$, pH 9.0); anti-mouse CD8α (98941, Cell Signaling Technology; D4W2Z, rabbit monoclonal, 1:200, pH 9.0); anti-mouse F4/80 (MCA497, Serotec; CLA3-1, rat monoclonal, 1:20, pH 6.0); anti-mouse FOXP3 (14-5773, eBioscience; FJK-16s, rat monoclonal, 5 μg ml$^{-1}$, pH 9.0); anti-human CD4 (M7310, Dako; 4B12, mouse monoclonal, 1:50, pH 9.0); anti-human CD8 (RM-9116-S, Thermo Fisher Scientific; SP16, rabbit monoclonal, 1:100, pH 9.0); and anti-human CD68 (NCL-L-CD68, Novocastra; 514H12, mouse monoclonal, 1:50, pH 9.0).

The following horseradish peroxidase (HRP) polymer kit was used for manual IHCs: M.O.M. ImmPRESS HRP Polymer Kit (MP-2400, Vector Laboratories); ImmPRESS HRP Horse Anti-Rabbit IgG Polymer Kit (MP-7401, Vector Laboratories); and ImmPRESS HRP Goat Anti-Rat IgG Polymer Kit (MP-7404, Vector Laboratories).

## Image analysis and quantification

For in vitro slides, quantification of γH2AX was performed in Fiji (ImageJ2 v2.14.0). In brief, a nuclear mask was applied based on the DAPI channel, and then the mean γH2AX intensity was measured per cell.

For in vivo liver tissue sections, quantification of γH2AX was performed manually after scanning using Axioscan 7 (Zeiss) at ×40 magnification. Random areas were selected and at least 100 NRAS$^+$ or NRAS$^-$ cells per liver section were counted. Representative images were taken using TCS SP5 confocal microscope (Leica). For measuring the perecnt of positive tissue areas, image analysis was performed using the HALO (Indicalabs, v3.3.2541) with the Area Quantification v1.0 algorithm following the digitization of tissue sections. IHC images were trained independently to provide the best accuracy for the positive area and all the slides were reviewed manually following analysis to assess accuracy. In brief, the total section area was highlighted using the Flood fill annotation tool, and a minimum tissue optical density at

0.035 was used to eliminate non-tissue areas. Percentage stain-positive tissue was used as readout for statistical analysis performed using GraphPad Prism 10.2.1 (339).

## Tumour scoring

Haematoxylin and eosin (H&E)-stained tissue sections were reviewed by a board-certified pathologist (S.J.A.) who was blinded to the experimental design. Tumours were graded according to the WHO classification of digestive system tumours[50]. Differentiation scores were assigned: DS1, well differentiated; DS2, moderately differentiated; DS3, poorly differentiated; and DS4, undifferentiated. For morphologically heterogeneous tumours, or where multiple lesions were present in the same liver, tumours were classified based on the worst grade.

## Bulk RNA-seq

RNA was extracted from five biological replicates per condition using the Qiagen RNeasy plus kit according to the manufacturer's instructions and quality checked using a Bioanalyser Eukaryote Total RNA Nano Series II chip (5067-1511, Agilent). Libraries were prepared using the TruSeq Stranded mRNA Library Prep Kit (20020594, Illumina) according to the manufacturer's instructions and sequenced using the HiSeq-4000 platform (Illumina). Reads were aligned to the human genome version GRCh38 (downloaded from https://www.ensembl.org/Homo_sapiens/Info/Index) using STAR[51], and per-gene read counting was performed using the featureCounts function of the subread package in R[52]. Low-quality reads (mapping quality less than 20) and known adapter contamination were filtered out using Cutadapt[53]. Differential expression analysis was performed with edgeR[54,55], comparing each of the induced samples with their uninduced equivalent. Differentially expressed genes were identified using edgeR's glmTreat function using a fold change of 1.2 in either direction and a false discovery rate cut-off of 0.05.

## Gene set enrichment and pathway analysis

Rank-based gene set enrichment analysis and generating the associated random-walk plots were performed using the fgsea R package[56]. Expression values were tested against gene sets curated as part of the MSigDB, a collection of gene sets representing coherently expressed signatures designed to represent well-defined biological states or processes[57]. Overlap-based pathway and gene ontology enrichment was performed using the web-based Enrichr platform[58,59].

All summary plots were generated in R, mostly using the ggplot2 package[60]. Upset plots were generated using the UpSetR package[61], and heatmaps were generated using the pheatmap package, which also implements hierarchical clustering for the ordering of columns and rows where indicated.

## Cancer Cell Line Encyclopedia and TCGA

Cancer Cell Line Encyclopedia expression data were downloaded from the DepMap Portal[62]. The liver cell lines were grouped into well-differentiated and poorly differentiated lines based on previous classification[44,45]. When projected into two dimensions, differentiation status of the cell lines was the primary driver of the first principal component. As such, genes were ranked from well to poorly differentiated based on their loadings along this principal component. TCGA expression and mutation data were downloaded from the GDC data portal[25]. Survival analysis and visualization of this data were performed using the survminer R package. For the diagnostic value of gene signatures, an intersect was taken between gene lists associated with the indicated Hoshida subclasses and either the *Notch1*-associated or *Dlk1*-associated branches in our data.

## Human premalignant liver patient cohort

All biological samples were collected with informed consent from Addenbrooke's Hospital, Cambridge, UK, according to procedures

approved by the Office for Research Ethics Committees Northern Ireland (ORECNI; 20/NI/0109). All participants consented to publication of research results.

## scRNA-seq and analysis

For hepatocyte scRNA-seq, livers were perfused with 0.05% collagenase in Hank's balanced salt solution (HBSS) to partial dissociation, then cut into pieces with a razor blade or scalpel, in HBSS with 0.015% collagenase and 0.2% dispase. The resulting cell suspensions were incubated with 0.02% DNase in HBSS before red blood cell lysis (00-4333-57, eBioscience; 5 min on ice) and then washed with HBSS with 0.02% DNase (centrifuged for 7 min at 400$g$ at 4 °C) to isolate hepatocytes. For RPE1 scRNA-seq, cells were trypsinized into single-cell suspension.

Cells isolated from the different conditions (RPE1) or mice (hepatocytes) were individually labelled with 1 µg of BioLegend TotalSeq Cell Hashing antibodies diluted in cell staining buffer (PBS, 3% FBS and 0.05% azide) for 30 min at 4 °C, and then washed three times with cell staining buffer (centrifuged for 7 min at 400$g$ at 4 °C). Hepatocytes were flow sorted for mVenus positivity according to the gating strategy in Supplementary Information. In each cohort (Figs. 1 and 3), we used two mice per condition, except for non-oncogenic *CAGGS-NRAS*[G12V/D38A] (one mouse) in the first cohort (Fig. 1). For RAS-induced RPE1 cells (day 6 post-4-OHT treatment), we used both individual subpopulations and a mixed population, with a mixed population (no 4-OHT treatment) as control. This allowed us to pool all conditions into the same experimental run. Cells were then pooled and resuspended to a concentration of 800 cells per microlitre for single-cell encapsulation using the Chromium Single Cell B Chip Kit (PN-1000073, 10X Genomics), followed by library prep using the Chromium Single Cell 3' GEM Library & Gel Bead Kit v3 (PN-1000075, 10X Genomics) for the gene expression library and the Chromium Single Cell 3' Feature Barcode Library Kit (PN-1000079, 10X Genomics) for the hashtag-oligo library. Both libraries were then pooled for paired-end sequencing on the HiSeq-4000 (OIS dataset and RPE1 dataset) or the Illumina NovaSeq 6000 platform (tumours dataset).

Hashtags used for each sample were: for the liver OIS dataset (TotalSeq-A anti-mouse), G12V-1 hashtag 1 (ACCCACCAGTAAGAC); G12V-2 hashtag 2 (GGTCGAGAGCATTCA); and D38A hashtag 3 (CTTGCCGCATGTCAT).

For the RPE1 dataset (TotalSeq-A anti-human), monoculture 'S' d6 hashtag 1 (GTCAACTCTTTAGCG); monoculture 'M' d6 hashtag 2 (TGATGGCCTATTGGG); monoculture 'L' d6 hashtag 3 (TTCCGCCTCTCTTTG); monoculture 'XL' d6 hashtag 4 (AGTAAGTTCAGCGTA); co-culture d0 hashtag 5 (AAGTATCGTTTCGCA); and co-culture d6 hashtag 6 (GGTTGCCAGATGTCA).

For the liver tumours dataset (TotalSeq-B anti-mouse), mVenus only-1 hashtag 1 (ACCCACCAGTAAGAC); mVenus only-2 hashtag 2 (GGTCGAGAGCATTCA); day 12-1 hashtag 3 (CTTGCCGCATGTCAT); day 12-1 hashtag 4 (AAAGCATTCTTCACG); day 30-1 hashtag 5 (CTTTGTCTTTGTGAG); day 30-2 hashtag 6 (TATGCTGCCACGGTA); tumour-1 hashtag 7 (GAGTCTGCCAGTATC); tumour-2 hashtag 8 (TATAGAACGCCAGGC); non-tumour-1 hashtag 9 (TGCCTATGAAACAAG); and non-tumour-2 hashtag 10 (CCGATTGTAACAGAC).

Resulting reads were aligned using the CellRanger pipeline to the mm10 genome assembly for the hepatocyte datasets and hg38 for the RPE1 dataset. Demultiplexing based on expression of hashtag oligos was performed using the CITE-seq-Count command, with no mismatches allowed. As all conditions to be compared were pooled into the same experimental run, direct analysis could be performed without the need for integration or batch correction. After quality-control filtering to remove low-quality sequenced cells, all downstream analysis, including pseudotime analysis, a technique that models single-cell transcriptional change as a continuum, was performed using the Seurat[63,64], Monocle[65] or dynverse[66] implementations in R.

## Statistical analysis

Statistical analyses were carried out in R (v4.1.1) or using the Prims10 built-in analysis (v10.1.1). The number ($n$) of biologically independent samples is described in the figure legends and Methods, and the data points are shown with the bar charts. Tests used to assess statistical differences between conditions are described in the respective figure legends. See Source Data.

For the mouse scRNA-seq experiments, in each cohort (Figs. 1 and 3), we used two mice per condition, except for non-oncogenic *CAGGS-NRAS*[G12V/D38A] (one mouse) in the first cohort (Fig. 1). The western blot in Fig. 2d was repeated in three independent experiments, and results were reproduced. Figure 4e shows representative images from a cohort of 13 patients with hepatitis C (further patient details are in Supplementary Table 3). The immunofluorescence in Extended Data Fig. 3b was repeated in three independent experiments. The IHCs in Extended Data Figs. 6 and 10 were repeated for the number of $n$ mice as indicated on the figure, and results were reproduced as shown in the associated quantifications.

## Reporting summary

Further information on research design is available in the Nature Portfolio Reporting Summary linked to this article.

## Data availability

Details of publicly available datasets are provided on the respective figure panels, and in the Methods and Supplementary Table 2 (refs. 13,67–74). The scRNA-seq datasets were downloaded from https://www.ncbi.nlm.nih.gov/query/acc.cgi?acc=GSE141017 (mouse premalignant pancreas), https://www.ncbi.nlm.nih.gov/geo/query/acc.cgi?acc=GSE155698 (human pancreas) and https://www.ncbi.nlm.nih.gov/geo/query/acc.cgi?acc=GSE131907 (human lung), respectively. The URLs for downloading the bulk RNA-seq datasets used in this study are provided in Supplementary Table 2. TCGA data were downloaded from the GDC portal (https://portal.gdc.cancer.gov/). The RNA-seq and scRNA-seq data generated in this study have been deposited in the Gene Expression Omnibus under the accession code GSE222951. Source data are provided with this paper.

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

**Acknowledgements** We thank I. Olan and other members of the Narita laboratory for their inputs; the staff of the Cancer Research UK Cambridge Institute (CRUK-CI) core facilities (Histopathology, Flow Cytometry, Biological Resources Unit, Microscopy, Genomics, Biorepository, Bioinformatics and Research Instrumentation); and S. Jackson laboratory for technical support. The Human Research Tissue Bank is supported by the NIHR Cambridge Biomedical Research Centre. The Narita laboratory is supported by a CRUK-CI core grant (C9545/A29580). Masashi Narita, L.D.C. and H.Z. are supported by the Biotechnology and Biological Science Research Council (BB/T013486/1). Masashi Narita is also supported by the Biotechnology and Biological Science Research Council (BB/S013466/1) and Diabetes UK via BIRAX and the British Council (65BX18MNIB). S.J.A. is supported by a Cancer Research UK Clinician Scientist Fellowship (RCCCSF-May23/100001) and Medical Research Council Toxicology Unit core funding (RG94521). M.H. is supported by a Cancer Research UK Programme Foundation Award (DRCPFA-Jun22\100001), an MRC Research grant (MR/X00970X/1) and a Cancer Research UK–Oregon Health and Science University Joint Award (C52489/A29681). N.O. is supported by the US NIH/NCI (R01CA245535). L.Z. was supported by the Deutsche Forschungsgemeinschaft (German Research Foundation): SFB-TR240 and EXC 2180-390900677, iFIT 'Image Guided and Functionally Instructed Tumor Therapies'. Schematics were partly created with BioRender.com.

**Author contributions** A.S.L.C., H.Z. and Masashi Narita conceived the study. A.S.L.C. performed most of the experiments on in vitro samples. H.Z. performed most of the experiments on mouse samples. L.D.C. and A.R.J.Y. set up the in vivo system (Figs. 1a and 3a,b). C.B.-R. and Masako Narita set up the in vitro system (Fig. 2a–d). A.S.L.C. analysed the RNA-seq and scRNA-seq data. N.O. provided materials and analysed the RNA-seq data relating to the squamous cell carcinoma model (Extended Data Fig. 5e). A.T.J. performed the DNA damage immunofluorescence experiments in TIG3 cells (Extended Data Fig. 3g). H.-C.C., S.G. and H.Z. analysed the flow cytometry data (Fig. 3e). S.J.A. performed the pathological analysis. L.Z. and M.H. provided materials, supervised in vivo experiments and interpreted the data. A.S.L.C., H.Z. and Masashi Narita wrote the paper with input from the other authors.

**Competing interests** M.H. has received an unrestricted research grant from Pfizer and consults for AstraZeneca, Boston Scientific and Quotient Therapeutics. All of the other authors declare no competing interests.

**Additional information**
**Correspondence and requests for materials** should be addressed to Masashi Narita.

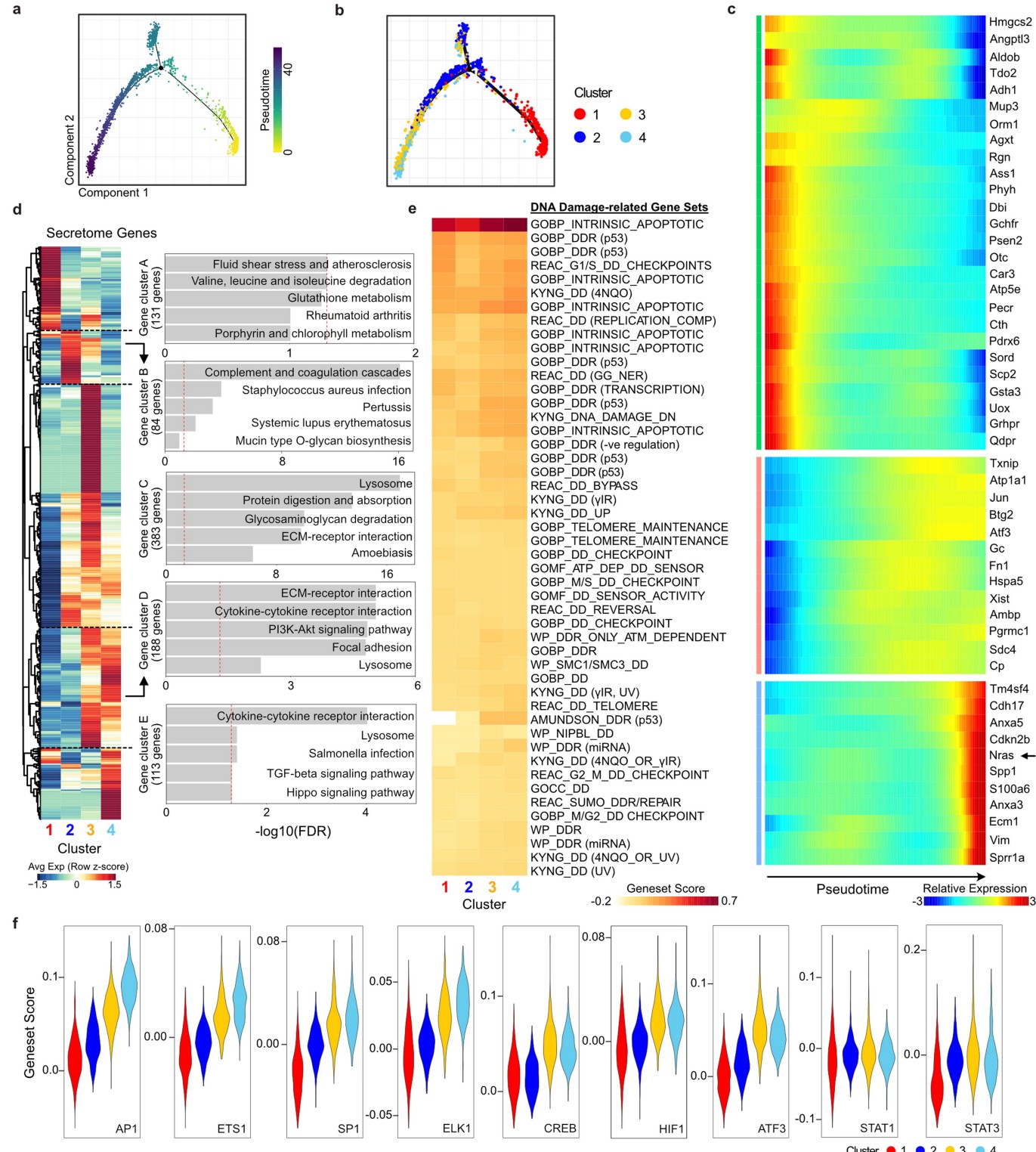

**Extended Data Fig. 1 | Characterisation of dose-dependent response to RAS expression at single cell level in the liver model. a,b,** Projection of single cells coloured by pseudotime (a) or cluster as in Fig. 1b (**b**). **c,** Top 50 genes driving pseudotime ordering, arranged in order of similarity of expression pattern across pseudotime. **d,** Heatmap of average gene expression across single cells in each cluster for 899 secretome genes. The top 5 enriched KEGG pathways in each of the clusters are shown. Values, -log10(FDR), red dotted line indicates significance level of 0.05. **e,** Expression of all DNA damage-related gene sets from entire MsigDB across clusters. Terms were manually trimmed but the full descriptions are in Supplementary Table 1. GOBP = Gene Ontology Biological Process, REAC = Reactome, GOMF = Gene Ontology Molecular Function, WP = WikiPathways. **f,** Distribution of geneset scores for target genes of the indicated RAS downstream transcription factors across clusters.

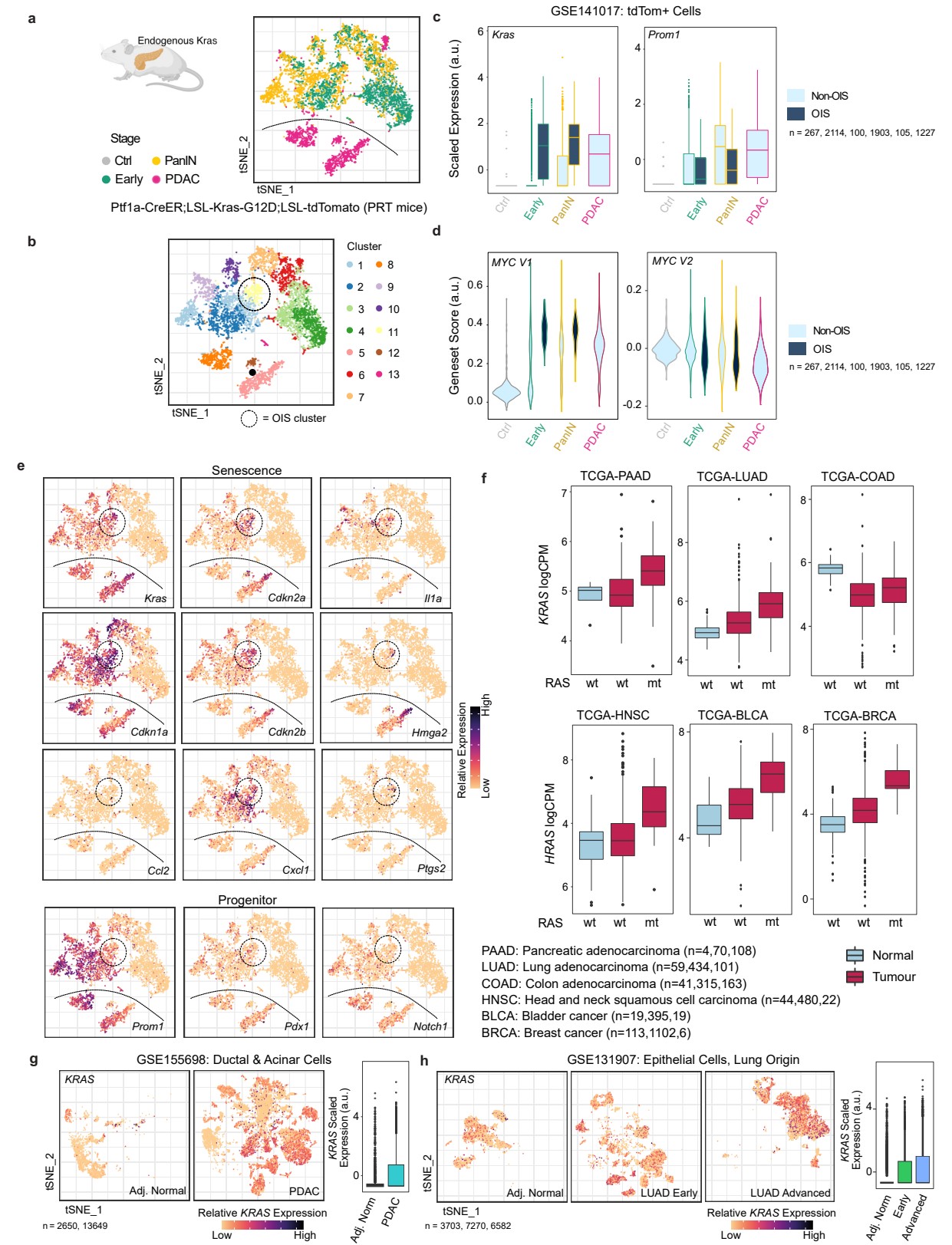

**Extended Data Fig. 2 | RAS dose-dependency in non-liver contexts. a,b,** tSNE projection of tdTom+ cells from the pancreas model coloured by sample-of-origin (**a**) and cluster (**b**). Schematic in panel **a** was created using BioRender (https://biorender.com). **c,d,** Distribution of expression levels at single-cell level for the indicated genes (**c**) and gene signature (**d**) in endogenous Kras^(G12D)-driven pancreatic tumour model (PRT mice). Values for preneoplastic "Early" and "PanIN" were divided into two based on the clustering in Extended Data Fig. 1b, as indicated by the colour of the box/violin. Cells from the *Cdkn2a/p16* positive Cluster 11 were designated as "OIS", whilst all the other "Early" and "PanIN" cells were designated as "non-OIS". PDAC, pancreatic ductal adenocarcinoma. n values indicate number of cells. **e,** tSNE projections coloured by indicated genes-of-interest. **f,** Expression of *KRAS* or *HRAS* in TCGA samples of the indicated tumour types, separated by RAS mutation status. wt, wild-type; mt, mutant. n values indicate number of patients. **g,h,** Upregulation of *KRAS* in human pancreatic (**g**) and lung (**h**) cancer cells, compared to normal epithelial cells, in public scRNA-seq datasets. Ductal cell clusters were identified using KRT19 expression, acinar cell clusters by CPA1 and CPA2 (**e**). *KRAS* expression in lung epithelial cells of human lung adenocarcinoma samples, comparing between adjacent normal and tumour cells from different disease stages (**f**). Lung epithelial cell subset is based on annotation by the original authors. n values indicate number of cells. All boxplot centre line indicates median, box limits indicate first- and third-quartiles and whiskers indicate largest values within 1.5 * interquartile range.

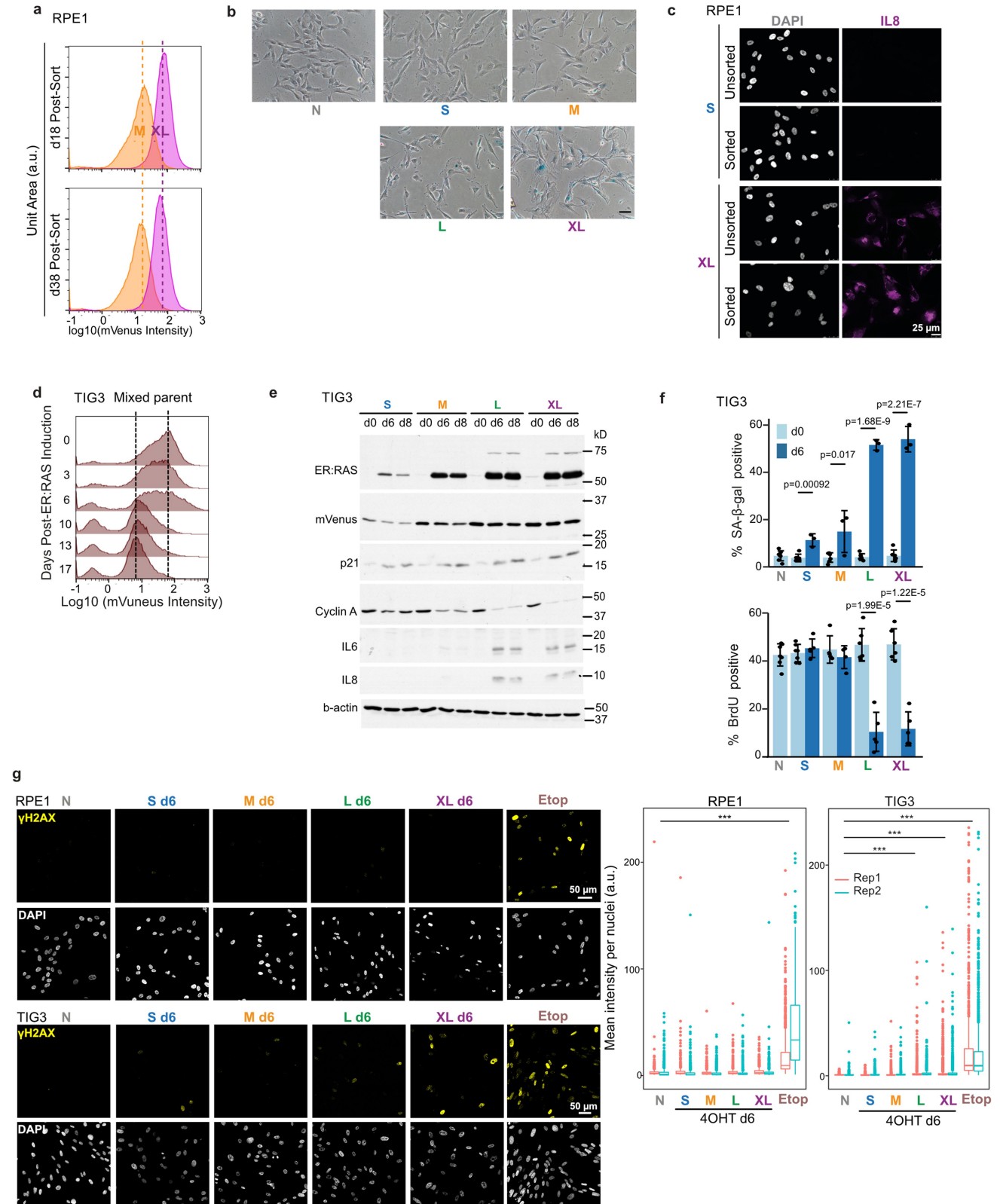

**Extended Data Fig. 3** | See next page for caption.

**Extended Data Fig. 3 | Characterisation of the in vitro predictive reporter system. a**, Flow cytometry analysis for mVenus intensity in uninduced cells, at different timepoints post-sorting in RPE1 cells. **b**, Representative phase contrast pictures of SA-β-gal assay (quantifications in Fig. 2e). Scale bar = 50 µm. **c**, IL-8 immunofluorescence for 'S' and 'XL' RPE1 cells on Day 9 comparing unsorted cells and cells, which were sorted on Day 6 to enrich for S-phase cells. **d**, Distribution of mVenus intensity over time by flow cytometry for a mixed population of TIG3 cells expressing the predictive reporter construct. **e,f**, Senescence phenotype of the TIG3 sorted subpopulations was assessed by Western blotting for the indicated proteins (**e**), SA-β-gal positivity and BrdU incorporation (**f**). Error bars, s.d. Statistical significance was determined using two-way pairwise student's t-test with no correction for multiple testing. **g**, γH2AX staining (left) and quantification of mean γH2AX intensity within a nuclear mask for the indicated conditions (right) in RPE1 and TIG3 cells. Images are representative from n = 2 per condition where n = independent experiments. N, plain. Etop, Etoposide (50 µM) treatment for 24 h as positive controls. Individual replicates are shown in quantification. Boxplot centre line indicates median, box limits indicate first- and third-quartiles and whiskers indicate largest values within 1.5 * interquartile range. One-way ANOVA followed by Tukey's HSD test. ***p < 0.001. Statistical significance was calculated between the indicated conditions, pooling values from both replicates.

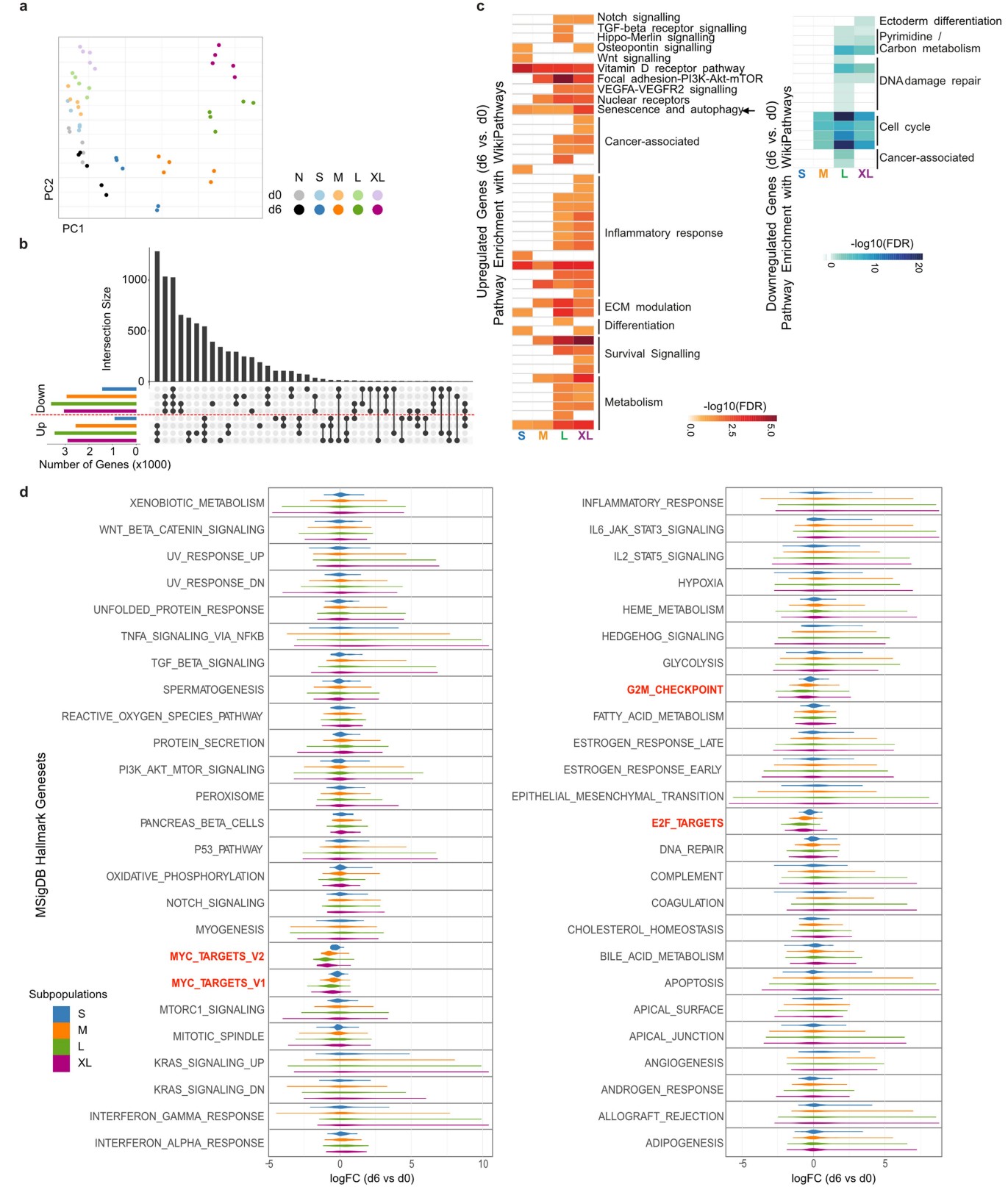

**Extended Data Fig. 4 | RNA-seq analysis for individual cell populations in RPE1 cells expressing predictive reporter construct. a-c**, Principal component analysis (**a**), number of differentially expressed (DE) genes (**b**), and pathway enrichment analysis (**c**) for the sorted subpopulations. N, plain RPE1 cells (no mVenus-P2A-ER:HRAS^GI2V transduction). n = 5 independent samples for each condition. **d**, Distribution of log-2-fold change values for genes in MsigDB Hallmark genesets for each of the subpopulations, comparing each Day 6 condition with its respective uninduced control. MYC targets and cell cycle genes are highlighted in red.

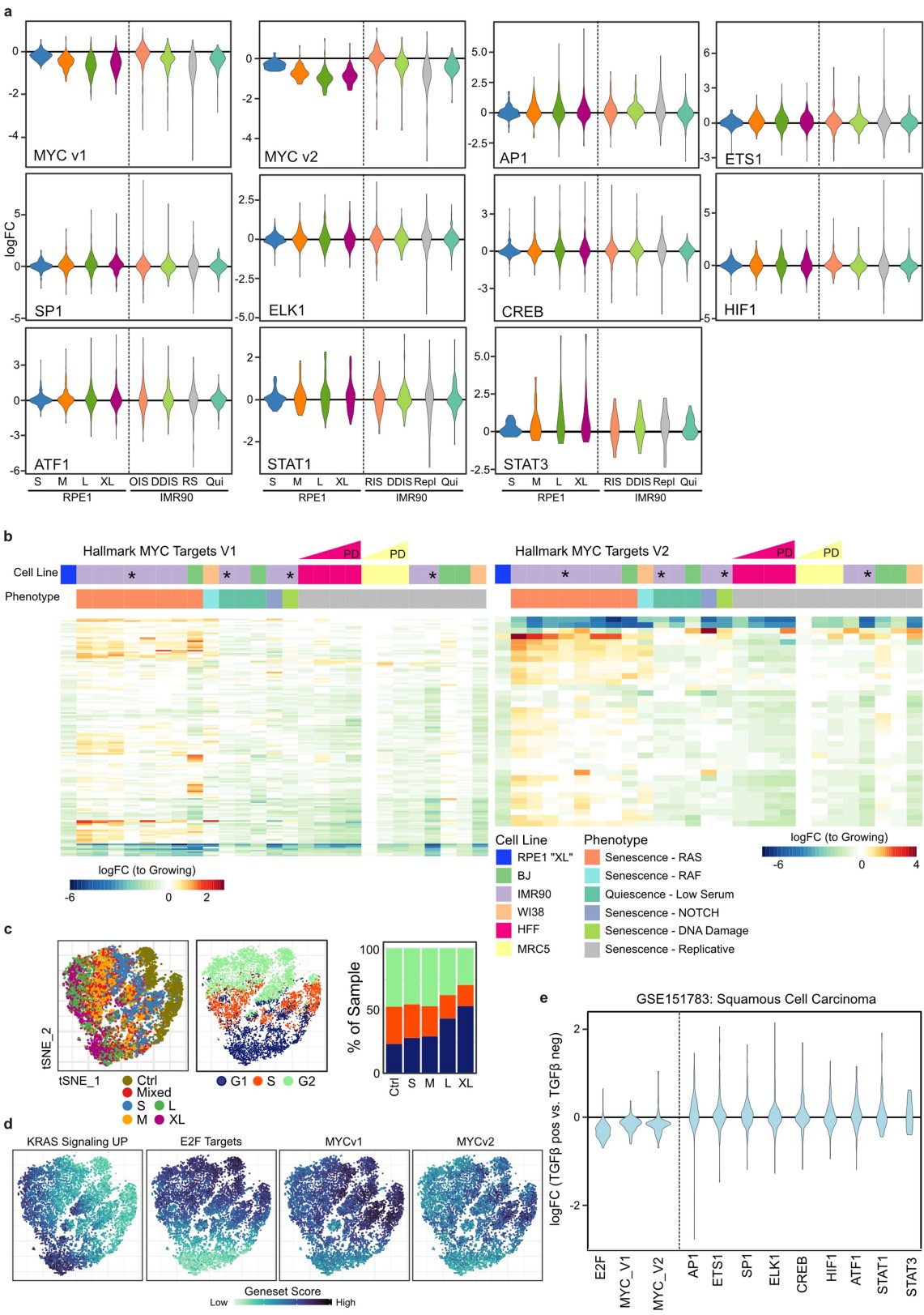

**Extended Data Fig. 5** | See next page for caption.

**Extended Data Fig. 5 | Meta-analysis of senescence-associated transcriptomic changes of MYC target genes. a**, Differential expression of TF-targets in each RPE1 subpopulation as well as indicated IMR90 cells. Known downstream TFs in the RAS-MAPK pathway were analysed. MYC (v1, v2) in RPE1 were duplicated from Fig. 1j for comparison. **b**, Gene expression datasets were downloaded from NCBI GEO (Supplementary Table 2), comparing different stress-induced cellular phenotypes associated with reduced cell cycling. * indicates IMR90 datasets that were utilised in (**a**). The datasets were processed using the same analysis pipeline, colours indicate log2-fold change of individual MYC-target genes (MSigDB Hallmark) between each of the conditions and their corresponding growing controls. PD = Population Doubling. Samples are in the order detailed in Supplementary Table 2. **c,d**, scRNA-seq analysis in RPE1 subpopulations (n = 9,047 cells). For RAS-induced samples, we used both individual subpopulations (n = 1/subpopulation) and a pooled sample of all subpopulations (n = 1) as a replicate. Control was a mix of all subpopulations (no 4OHT, n = 1). Each sample was Hashtagged, pooled and run as the same run. Cell-cycle phases were annotated using Seurat's inbuilt CellCycleScoring function and gene markers for S- and G2/M-associated genes (**c**). Indicated gene signatures (MsigDB Hallmarks) were scored (**d**). **e**, Changes in expression levels of MYC- and E2F-target genes (MSigDB Hallmark) in tumour-initiating cells (TGFβ-reporter positive) compared to the rest of the tumour cells (TGFβ-reporter negative) in a HRAS$^{G12V}$-driven mouse squamous cell carcinoma model (Supplementary Table 2). The downstream TFs in the RAS-MAPK pathway were included as a comparison.

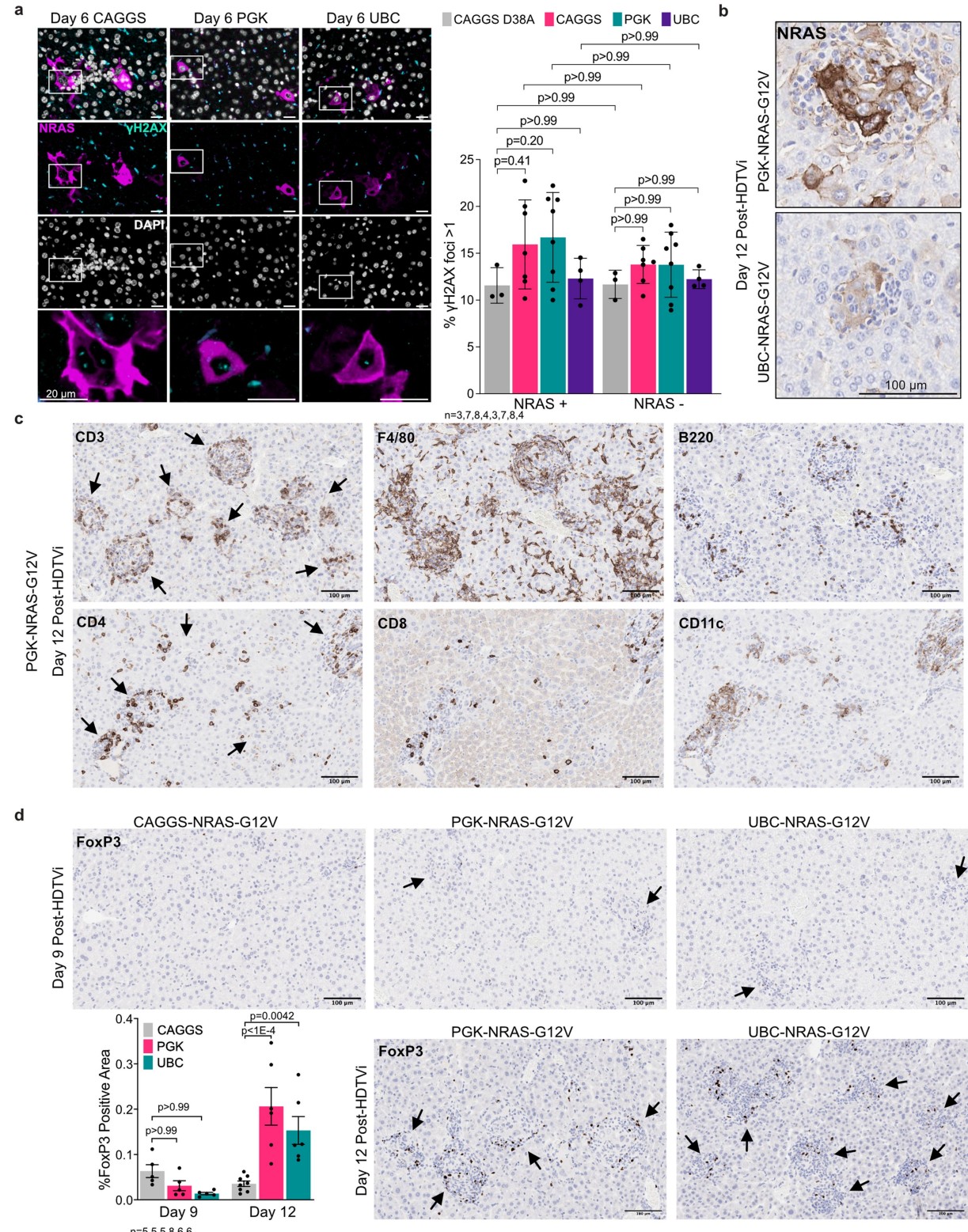

**Extended Data Fig. 6 | Characterisation of DNA damage and immune cell clusters in NRAS^G12V-injected livers. a**, NRAS-positive hepatocytes with γH2AX foci. Cells with >1 foci within the nucleus were counted as positive. Note nonspecific autofluorescence mainly from red blood cells. Values, mean (s.d.). p-values are Two-way mixed-effects ANOVA followed by post-hoc t-tests with Bonferroni correction. Scale bars = 20 μm. **b**, Persistent immune cell clusters in sub-OIS-NRAS^G12V livers. Selected areas of NRAS IHC Day 12 post-HDTVi in Fig. 3b are magnified. **c**, Representative IHC for indicated immune cell markers in persistent immune cell clusters. Each row consists of serial sections. **d**, Representative IHC for FoxP3 (Treg marker) in indicated samples. Values, mean (s.d.). p-values are Two-way ANOVA followed by post-hoc t-tests with Bonferroni correction. Scale bars = 100 μm. Arrows indicate immune cell clusters (c-d). n = number of mice (**a**,**d**).

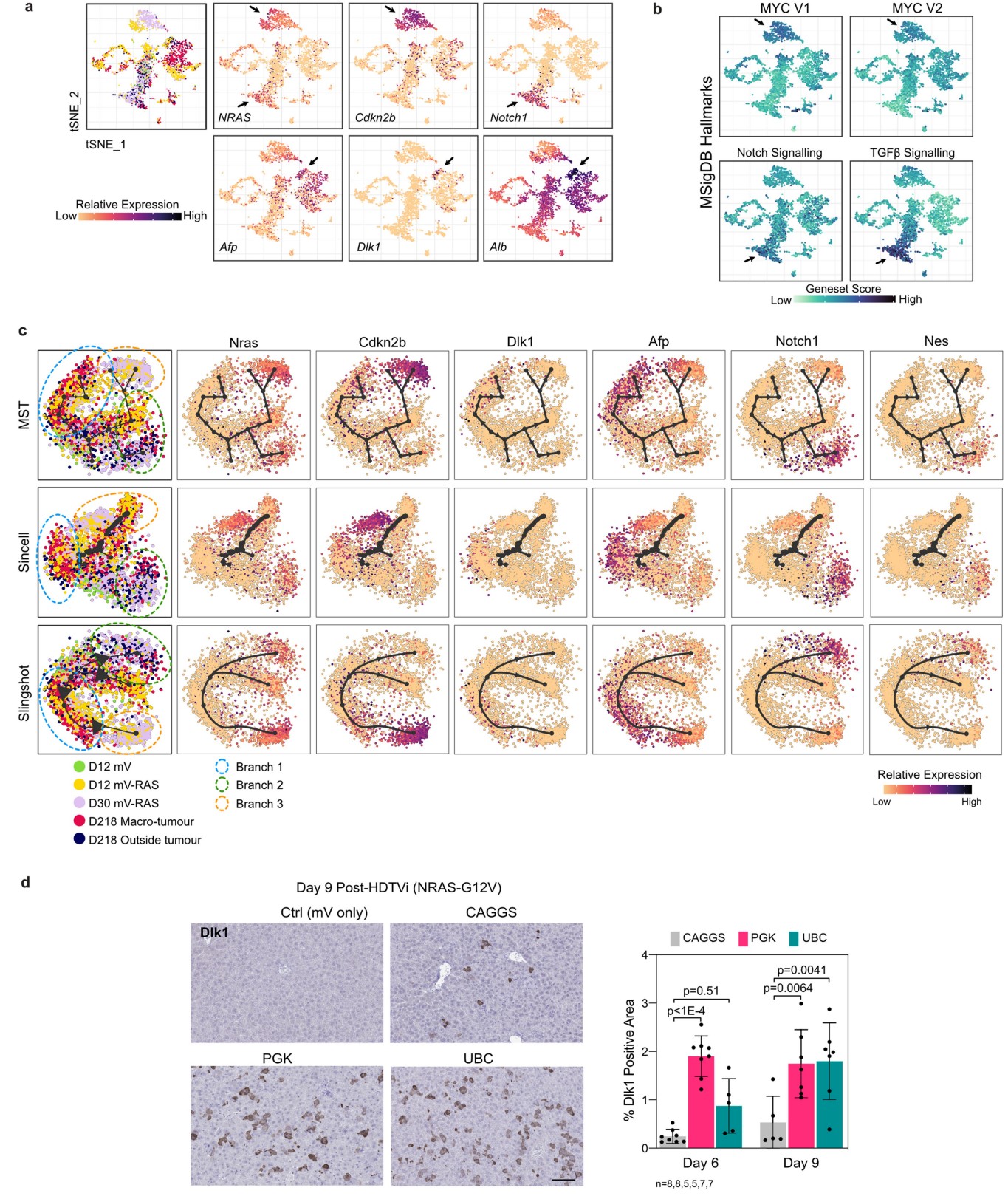

**Extended Data Fig. 7 | Additional characterisation of hepatocyte response to different oncogenic NRAS dosages. a,b**, tSNE embeddings coloured by sample of origin (as in Fig. 3f) and genes of interest (**a**), and geneset score (**b**). Arrows indicate cell clusters expressing high NRAS (with either OIS or Notch/ TGFβ signature, left) or progenitor markers (right). **c**, Alternative trajectory inference analyses to Fig. 3i for scRNA-seq, using different algorithms on the same data as derived from the time series cohort of mice injected with low-dose NRAS. Branches indicated on the left-most panel in dotted circles are numbered according to Fig. 3g. **d**, IHC validation for Dlk1 on Day 9 post HDTVi (left) and quantification of % Dlk1 positive area (right, n represents number of mice). Values, mean (s.d.). p-values are two-way ANOVA followed by post-hoc t-tests with Bonferroni correction. Scale bar = 100 μm.

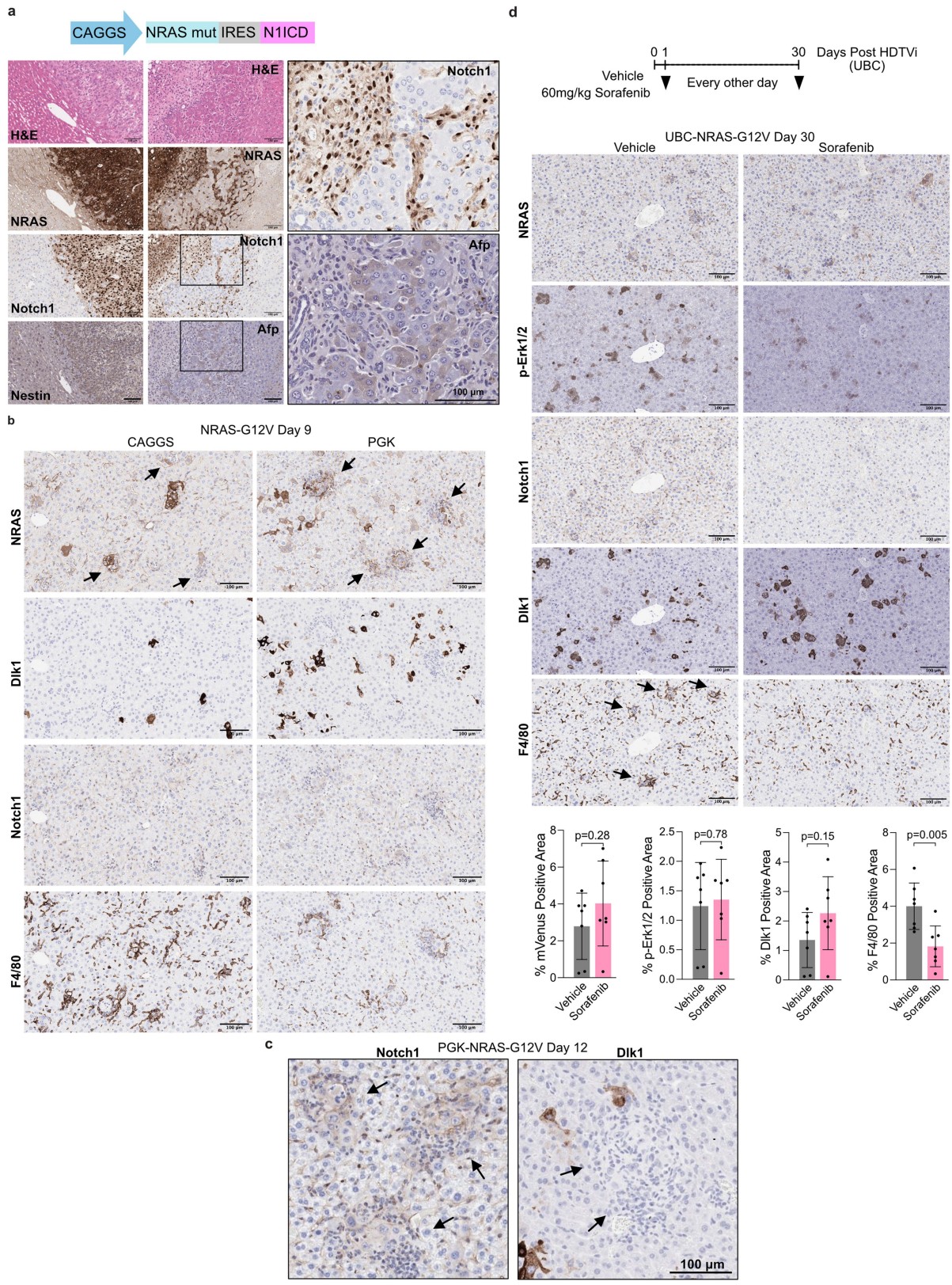

**Extended Data Fig. 8** | See next page for caption.

**Extended Data Fig. 8 | Characterisation of tumours associated with distinct TIC events. a**, Representative IHC for the indicated proteins in tumours induced by HDTVi of NRAS$^{G12V}$-IRES-N1ICD construct in mice (n = 6), showing co-expression of Notch1 and Nestin and mutual exclusivity between Notch1+ tumours and Afp+ cells (high magnification panels, right). Of note, while the tumour cells were poorly differentiated, the Afp+ cells maintained histological features of hepatocytes. **b,c**, Representative IHC for the indicated proteins on Day 9 (**b**) or Day 12 (**c**) livers after NRAS$^{G12V}$-HDTVi with the indicated dosages. Each column (**b**) represents serially sectioned images (n ≥ 5 mice). Magnified images from Fig. 3k are shown in (**c**) (n ≥ 6 mice). Scale bars = 100 μm (b) or 200 μm (**c**). Arrows indicate immune cell clusters. Dlk1 was mostly excluded from immune cell clusters. Notch1 staining was typically clearer in Day 12 (c), involved in 'persistent' immune cell clusters. **d**, Sorafenib treatment led to reduced accumulation of macrophage in UBC-NRAS$^{G12V}$-livers. Representative IHC for indicated proteins (n = 7 mice per condition). Serial sections were utilised in each condition. Scale bars = 100 μm. Values, mean (s.d.). p-values are unpaired t-test.

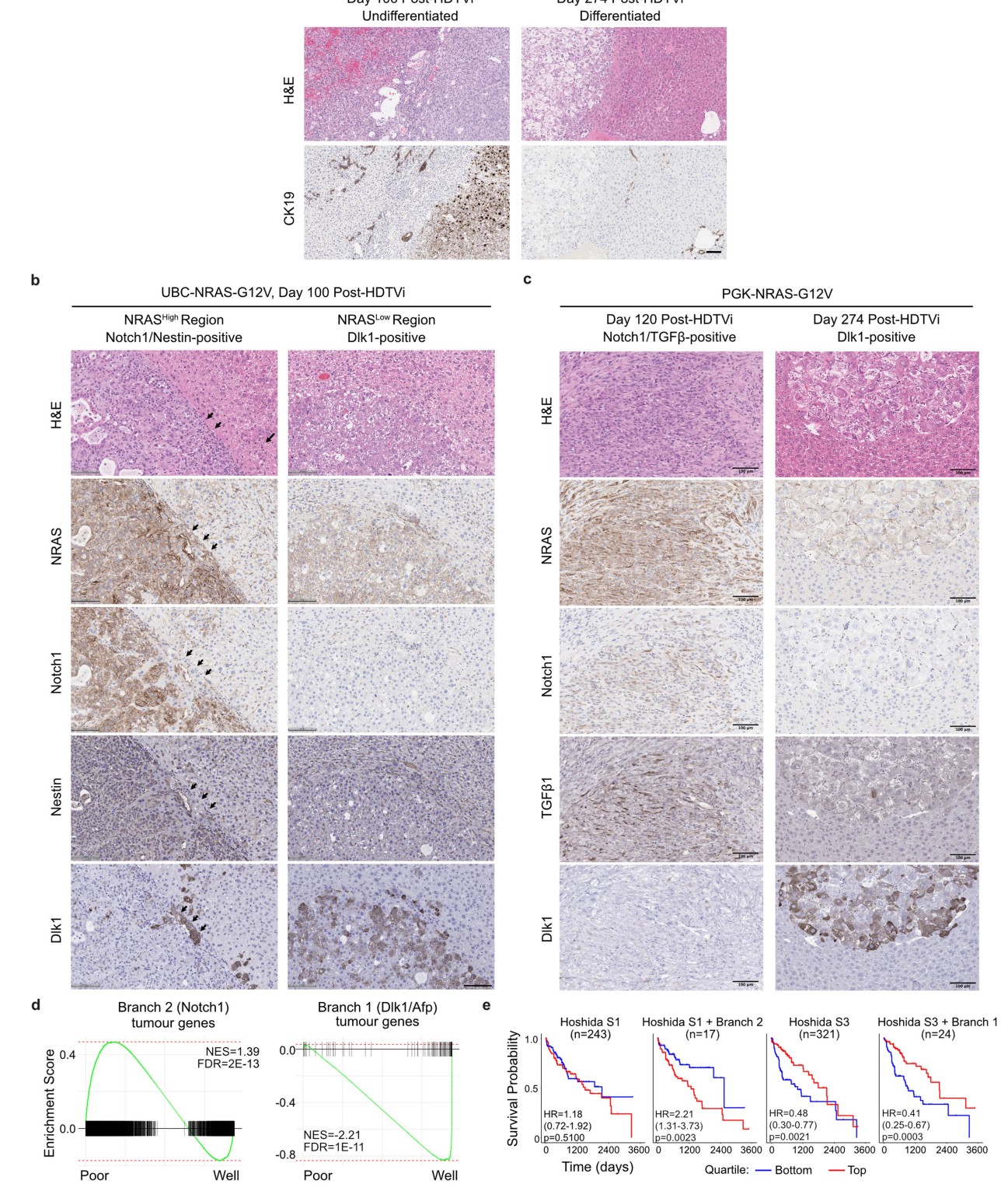

**Extended Data Fig. 9 | Validation of gene signatures in two TIC branches.**
**(a-c)** Representative serially sectioned IHC images of indicated tumours for the indicated proteins. All undifferentiated (DS 4) tumours (n = 5 mice) were CK19-positive (a). Dichotomous expression of either Notch1 or Dlk1 are shown even in the same tumour (b). 5 out of 6 Notch1-positive tumours were Tgfβ1-positive (c). Scale bars = 100 μm. Arrows indicate Dlk1-positive cells within the Notch1/Nestin-tumour. Of note, the Dlk1-positive cells tend to be well-differentiated and express low level of NRAS. **(d)** Random walk plots for geneset enrichment analysis for the indicated genesets against ranked genes between poorly- and well-differentiated HCC based on human liver cancer cell lines. **(e)** Kaplan-Meier analysis for the indicated gene signatures in TCGA-LIHC (Liver hepatocellular carcinoma) dataset. n values indicate number of genes in signature. HR = Hazard ratio for top quartile vs. bottom quartile, p = Log-rank p-value.

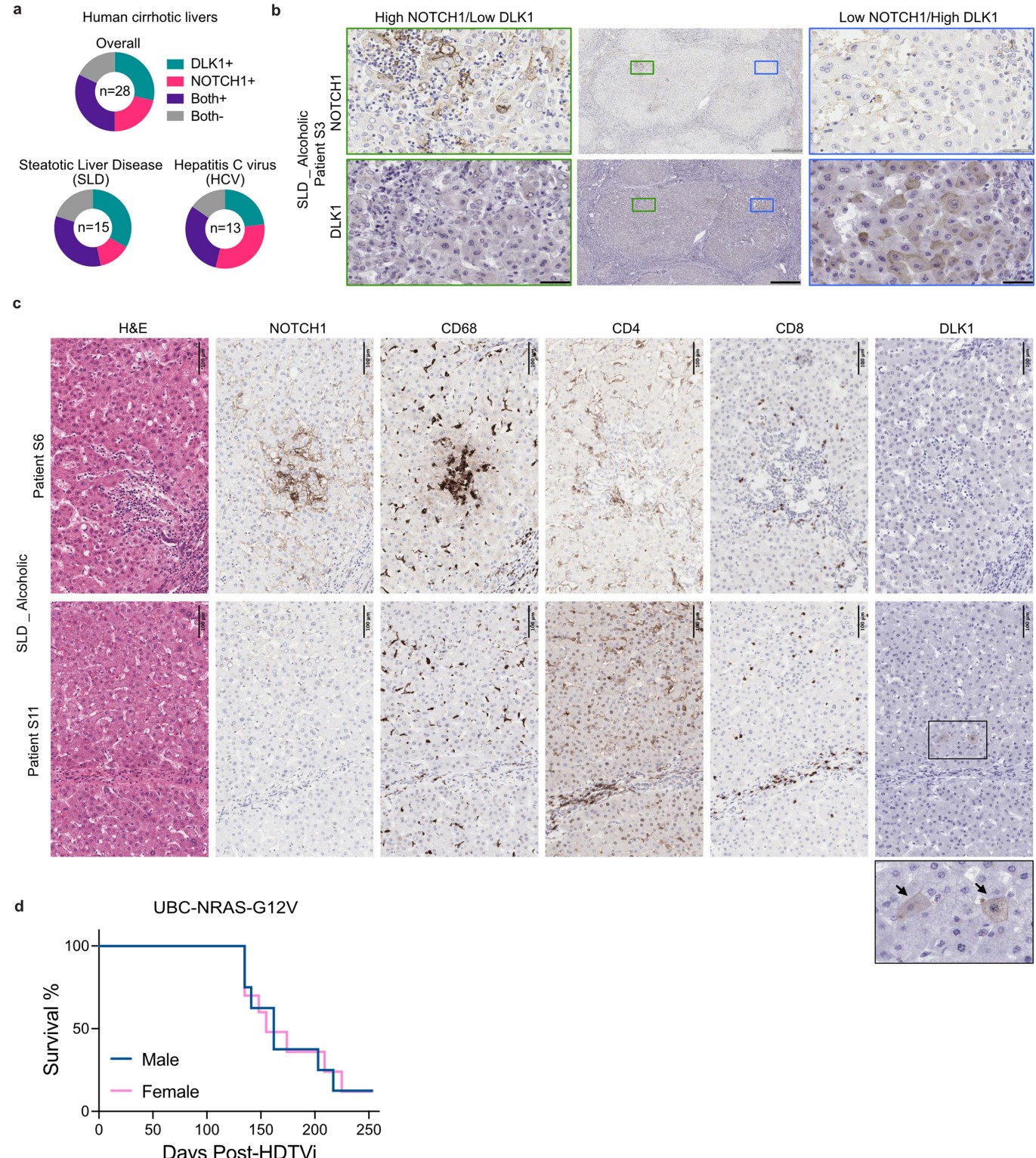

**Extended Data Fig. 10 | Immune cell clusters around NOTCH1-positive hepatocytes in human cirrhotic livers. a**, Frequency of NOTCH1 and/or DLK1-positive hepatocytes in liver cirrhosis patients examined. Steatotic liver disease (SLD) includes both non-alcoholic fatty liver disease (NAFLD) and alcohol-related liver disease (ALD). **b**, Representative IHC for the indicated proteins in a patient with SLD-related cirrhosis, showing dichotomous expression of NOTCH1 and DLK1 in the same liver. Scale bar = 500 μm in the centre panels and 50 μm in the magnified panels. **c**, Representative IHC for the indicated proteins in patients with steatotic liver disease-related cirrhosis. Serial sections were utilised for all patients in (**a**). Selected area is magnified with arrows, indicating DLK1-positive hepatocytes. **(d)** Kaplan-Meier analysis for male and female mice injected with the UBC-NRAS^(G12V) plasmids (n = 8 per condition). IHCs were performed in all patient samples (Supplementary Table 3). Scale bars = 100 μm.

# Reporting Summary

## Statistics

For all statistical analyses, confirm that the following items are present in the figure legend, table legend, main text, or Methods section.

| n/a | Confirmed | |
|---|---|---|
| ☐ | ☒ | The exact sample size (*n*) for each experimental group/condition, given as a discrete number and unit of measurement |
| ☐ | ☒ | A statement on whether measurements were taken from distinct samples or whether the same sample was measured repeatedly |
| ☐ | ☒ | The statistical test(s) used AND whether they are one- or two-sided<br>*Only common tests should be described solely by name; describe more complex techniques in the Methods section.* |
| ☒ | ☐ | A description of all covariates tested |
| ☐ | ☒ | A description of any assumptions or corrections, such as tests of normality and adjustment for multiple comparisons |
| ☐ | ☒ | A full description of the statistical parameters including central tendency (e.g. means) or other basic estimates (e.g. regression coefficient) AND variation (e.g. standard deviation) or associated estimates of uncertainty (e.g. confidence intervals) |
| ☐ | ☒ | For null hypothesis testing, the test statistic (e.g. *F*, *t*, *r*) with confidence intervals, effect sizes, degrees of freedom and *P* value noted<br>*Give P values as exact values whenever suitable.* |
| ☒ | ☐ | For Bayesian analysis, information on the choice of priors and Markov chain Monte Carlo settings |
| ☒ | ☐ | For hierarchical and complex designs, identification of the appropriate level for tests and full reporting of outcomes |
| ☒ | ☐ | Estimates of effect sizes (e.g. Cohen's *d*, Pearson's *r*), indicating how they were calculated |

*Our web collection on statistics for biologists contains articles on many of the points above.*

## Software and code

Policy information about availability of computer code

| Data collection | IF images were collected using LAS X 3.7.5.24914 or LAS X 4.7.0 software (Leica). |
|---|---|
| Data analysis | Analysis and Visualisation of flow cytometry data: FlowJo (v10.9.0). RNAseq: FastQC (https://www.bioinformatics.babraham.ac.uk/projects/fastqc/) and Cutadapt (v1.10) for pre-processing, STAR aligner (v2.7.6a) and subread (v1.5.3) for alignment and feature counting, edgeR (v3.20.9) for differential analysis. scRNAseq: CellRanger (v3.1.0) for alignment, Seurat (v4.0.4), Monocle (v2.26.0 and v3.16.0) for analysis. General statistical analysis: GraphPad Prism 10.2.1, General data visualisation: ggplot2 (v3.4.2). Heatmaps: pheatmap (v1.0.12), Upset plot: UpsetR (v1.4.0). Survival analysis: survminer (v0.4.7). Geneset Enrichment Analysis: fgsea (v3.18). |

For manuscripts utilizing custom algorithms or software that are central to the research but not yet described in published literature, software must be made available to editors and reviewers. We strongly encourage code deposition in a community repository (e.g. GitHub). See the Nature Portfolio guidelines for submitting code & software for further information.

## Data

Policy information about availability of data

All manuscripts must include a data availability statement. This statement should provide the following information, where applicable:

- Accession codes, unique identifiers, or web links for publicly available datasets
- A description of any restrictions on data availability
- For clinical datasets or third party data, please ensure that the statement adheres to our policy

Liver OIS scRNAseq (Fig. 1b-g), RPE1 RNAseq (Fig. 2 and Extended Data Fig. 2), RPE1 scRNAseq (Extended Data Fig. 5 c-d) and liver tumorigenesis scRNAseq (Fig. 3) have been deposited at GEO (GSE222339, GSE222337, GSE249489 and GSE222338 respectively, under the Super Series GSE222951). Pancreatic timecourse scRNAseq data was downloaded from GSE141017. TCGA data was downloaded from the National Cancer Institute's GDC Portal (https://gdc.cancer.gov). Senescence-associated RNAseq data (Extended Data Fig. 3a) was downloaded from GEO: GSE74324, GSE61130, GSE72407, GSE72404, GSE127116, GSE45833, GSE85082, GSE45833, GSE63577, GSE53356. Squamous Cell Carcinoma tumor-initiating cell data (Extended Data Fig. 3b) was downloaded from GSE151783. The reference human (hg38) and mouse (mm10) genomes were downloaded from ensembl.org.

## Research involving human participants, their data, or biological material

Policy information about studies with human participants or human data. See also policy information about sex, gender (identity/presentation), and sexual orientation and race, ethnicity and racism.

| | |
|---|---|
| Reporting on sex and gender | Sex information was collected for human cirrhosis patients, with consent for sharing of anonymised individual-level data. No significant difference in phenotype was observed between sex in this cohort, therefore sex information was not considered for further analyses. |
| Reporting on race, ethnicity, or other socially relevant groupings | Ethnicity information was collected for human cirrhosis patient, with consent for sharing of anonymised individual-level data. However, as this patient cohort lacks the power to systematically study differences in phenotype by ethnicity, and this point is also not the focus on the current manuscript, ethnicity information was not considered in any of the analyses of the current study. |
| Population characteristics | Population characteristics of patients were collected and summarised in Supplementary Table 3. |
| Recruitment | Tissue sections were obtained from patients undergoing liver transplantation, with either alcohol-related liver disease, nonalcoholic fatty liver disease or hepatitis C-associated liver disease (indicated in Supplementary Table 3). |
| Ethics oversight | All biological samples were collected with informed consent from Addenbrooke's Hospital, Cambridge, UK, according to procedures approved by the Office for Research Ethics Committees Northern Ireland (ORECNI) (20/NI/0109). All participants consented to publication of research results. This information is indicated in the Methods section of the manuscript. |

Note that full information on the approval of the study protocol must also be provided in the manuscript.

# Field-specific reporting

Please select the one below that is the best fit for your research. If you are not sure, read the appropriate sections before making your selection.

☒ Life sciences ☐ Behavioural & social sciences ☐ Ecological, evolutionary & environmental sciences

For a reference copy of the document with all sections, see nature.com/documents/nr-reporting-summary-flat.pdf

# Life sciences study design

All studies must disclose on these points even when the disclosure is negative.

| | |
|---|---|
| Sample size | For SAbgal and BrdU percent positivity, sample size was based on our previous study (PMID16901784). At least 200 cells were counted per condition from >3 biological replicates. Western blots and flow cytometry are reproduced in at least 3 independent experiments. Quantification for IF experiments was performed in at least 2 independent experiments. For HDTVi studies and associated experiments e.g. tumour incidence, IHC, sample size was based on our previous study (PMID27525720). For IHC percent positivity, whole liver positivity was measured from >3 biological replicates per condition. Intensity was measured from at least 200 cells per liver, from >3 biological replicates. For tumour incidence experiments, data are shown from >9 mice per condition. Human liver cirrhosis data is based on 28 patients. |
| Data exclusions | No data was excluded from the analysis |
| Replication | Liver OIS scRNAseq: 2 mice (G12V) and 1 mouse (D38A) respectively. RPE1 RNAseq: 5 biological replicates per condition. RPE1 scRNAseq: 1 monocultured biological replicate per subpopulation, plus a pooled second replicate (and uninduced control). Liver tumorigenesis scRNAseq: 2 mice per timepoint. Western blots, immunofluorescence, IHC and flow cytometry are reproduced in at least 3 independent experiments. All attempts at replication were successful. |
| Randomization | Plate layout randomisation was applied for RNAseq library preparation. For all scRNAseq experiments, cells from the different conditions were pooled prior to library prep then processed as a single sample, therefore randomisation is not possible nor necessary. For in vivo experiments, mice were randomised into groups for HDTV with the different constructs in each experiment. For in vitro experiments, the nature of the |

experimental setup is that dose-dependency is established in pre-defined subpopulations prior to RAS induction, therefore randomisation is not possible. Cells were randomised into "uninduced" and "induced" plates on the day of induction.

Blinding | Tumours were graded by a board-certified pathologist blinded to experimental design. Mouse liver tumour palpation for Kaplan-Meier analysis was performed weekly by an animal technician blinded to experimental condition. DLK1/NOTCH1 status of human samples was assessed with no knowledge of patient clinical information. No blinding was applied to other experiments, where data analysis are based on objectively measurable data i.e. quantifiable measurements.

# Reporting for specific materials, systems and methods

We require information from authors about some types of materials, experimental systems and methods used in many studies. Here, indicate whether each material, system or method listed is relevant to your study. If you are not sure if a list item applies to your research, read the appropriate section before selecting a response.

## Materials & experimental systems

| n/a | Involved in the study |
|---|---|
| ☐ | ☒ Antibodies |
| ☐ | ☒ Eukaryotic cell lines |
| ☒ | ☐ Palaeontology and archaeology |
| ☐ | ☒ Animals and other organisms |
| ☐ | ☒ Clinical data |
| ☒ | ☐ Dual use research of concern |
| ☒ | ☐ Plants |

## Methods

| n/a | Involved in the study |
|---|---|
| ☒ | ☐ ChIP-seq |
| ☐ | ☒ Flow cytometry |
| ☒ | ☐ MRI-based neuroimaging |

## Antibodies

Antibodies used | anti-GFP (Abcam ab13970), anti-Ras (Abcam ab52939, EP1125Y), anti-p-Erk1/2 (Cell Signaling Technology #9101), anti-CK8 (DSHB MABT329, TROMA-1), anti-CK19 (DSHB MABT913, TROMA-III), anti-mouse Nestin (Chemicon MAB353, rat-401), anti-human Nestin (Chemicon MAB5326, 10C2), anti-Afp (Santa Cruz sc-8399, C3), anti-mouse Dlk1 (R&D Systems #FAB8634T, 1168B), anti-human DLK1 (R&D Systems MAB1144, 211309), anti-Notch1 (Cell Signaling Technology #3608, D1E11), anti-TGF beta (Cell Signaling Technology #3709, 56E4), anti-mouse CD4 (Abcam ab183685, EPR19514), anti-mouse CD8α (Cell Signaling Technology #98941, D4W2Z), anti-mouse F4/80 (Serotec MCA497, CLA3-1), anti-mouse FoxP3 (eBioscience 14-5773, FJK-16s), anti-human CD4 (Dako M7310, 4B12), anti-human CD8 (Thermo Fisher Scientific RM-9116-S, SP16), anti-human CD68 (Novocastra NCL-L-CD68, 514H12), anti-BrdU (BD Biosciences Cat # 555627, 3D4), anti-phospho-Histone H2A.X (Ser139) (Merck Cat # 05-636, JBW301), anti-b-actin (Sigma Cat # A5441, AC15), anti-HRAS (Santa Cruz Biotechnology Cat # sc29, F235), anti-GFP (Clontech Cat # 632377), anti-IL6 (R&D Biosystems Cat # MAB2061, Clone#1936), anti-IL8 (R&D Biosystems Cat # MAB208, Clone#6217), anti-Cyclin A (Sigma Cat # c4710, CY-A1), anti-p21 (Santa Cruz Cat # sc-6246, F5)

Validation | Anti-b-actin (A5441, Sigma) Used for: WB. Species Against: Human. This antibody was validated by the company and used in our previous studies (PMID27525720, PMID29743479).
Anti-HRAS (sc29, Santa Cruz Biotechnology) Used for: WB. Species Against: Human. This antibody was validated by the company and used in our previous study (doi:10.1038/s43587-021-00147-y).
Anti-GFP (632377, Clontech) Used for: WB. Species Against: Human. This antibody was validated by the company and used in our previous study (PMID16901784).
Anti-IL6 (MAB2061, R&D Biosystems) Used for: WB. Species Against: Human. This antibody was validated by the company and used in our previous study (PMID33730589).
Anti-IL8 (MAB208, R&D Biosystems) Used for: WB, IF. Species Against: Human. This antibody was validated by the company and used in our previous study (PMID33730589).
Anti-Cyclin A (c4710, Sigma) Used for: WB. Species Against: Human. This antibody was validated by the company.
Anti-p21 (sc-6246, Santa Cruz) Used for: WB. Species Against: Human. This antibody was validated by the company.
Anti-phospho-Histone H2A.X (05-636, Merck) Used for: IF. Species Against: Human, Mouse. This antibody was validated by the company.
Anti-BrdU (555627, BD Biosciences) Used for: IF. Species Against: Human. This antibody was validated by the company and used in our previous study (PMID33730589).
Anti-Nras (ab52939. Abcam) Used for: IHC. Species Against: Mouse. This antibody was validated by the company and used in our previous study (PMID27525720).
Anti-GFP (ab13970, Abcam) Used for: IHC. Species Against: Mouse. This antibody was validated by the company.
Anti-Notch1 (3608, Cell Signalling Technology) Used for: IHC. Species Against: Mouse. This antibody was validated by the company and used in our previous study (PMID27525720).
Anti-Nestin (MAB353, Chemicon) Used for: IHC. Species Against: Mouse. This antibody was validated by the company.
Anti-Nestin (MAB5326, Chemicon) Used for: IHC. Species Against: Mouse. This antibody was validated by the company.
Anti-Dlk1 (FAB8634T, R&D Systems) Used for: IHC. Species Against: Mouse. This antibody was validated by the company.
Anti-DLK1 (MAB1144, R&D Systems) Used for: IHC. Species Against: Mouse. This antibody was validated by the company.
Anti-Afp (sc8399, Santa Cruz Biotechnology) Used for: IHC. Species Against: Mouse. This antibody was validated by the company.
Anti-CK19 (TROMA-III, DSHB) Used for: IHC. Species Against: Mouse. This antibody was validated by the company.
Anti-CK8 (TROMA-I, DSHB) Used for: IHC. Species Against: Mouse. This antibody was validated by the company.
Anti-pErk1/2 (9101, Cell Signalling Technology) Used for: IHC. Species Against: Mouse. This antibody was validated by the company.
Anti-TGF beta (3709, Cell Signaling Technology) Used for: IHC. Species Against: Mouse. This antibody was validated by the company.
Anti-mouse CD4 (ab183685, abcam) Used for: Flow. Species Against: Mouse. This antibody was validated by the company.

Anti-mouse CD8a (98941, Cell Signaling Technology) Used for: Flow. Species Against: Mouse. This antibody was validated by the company.
Anti-mouse F480 (MCA497, Serotec) Used for: Flow. Species Against: Mouse. This antibody was validated by the company.
Anti-mouse FoxP3 (14-5773, eBioscience) Used for: Flow. Species Against: Mouse. This antibody was validated by the company.
Anti-human CD4 (M7310, Dako) Used for: Flow. Species Against: Human. This antibody was validated by the company.
Anti-human CD8 (RM-9116-S, Thermo Fisher Scientific) Used for: Flow. Species Against: Human. This antibody was validated by the company.
Anti-human CD68 (NCL-L-CD68, Novocastra) Used for: Flow. Species Against: Human. This antibody was validated by the company.
Anti-TGFbeta (3709, Cell Signaling Technology) Used for: IHC. Species Against: Mouse. This antibody was validated by the company.

# Eukaryotic cell lines

Policy information about cell lines and Sex and Gender in Research

| Cell line source(s) | RPE1-hTert (ATCC), TIG3 (JCRB) |
|---|---|
| Authentication | Cells were obtained directly from the respective source cell banks. No authentication was performed by the authors of this manuscript. |
| Mycoplasma contamination | Cells were regularly tested for Mycoplasma contamination and always found to be negative. |
| Commonly misidentified lines (See ICLAC register) | No commonly misidentified lines were used in this study. |

# Animals and other research organisms

Policy information about studies involving animals; ARRIVE guidelines recommended for reporting animal research, and Sex and Gender in Research

| Laboratory animals | C57BL/6 and CB17/Icr-Prkdcscid/IcrIcoCrl mice were used in this study. HDTV injections were carried out on mice between 6 and 8 weeks of age. |
|---|---|
| Wild animals | No wild animals were used in this study. |
| Reporting on sex | Only female mice were used in all experiments apart from in the sex comparison long-term cohort, where one cohort of male mice was used. |
| Field-collected samples | No field-collected samples. |
| Ethics oversight | The CRUK CI Animal Welfare and Ethics Review Board (AWERB; Institutional Animal Care and Use Committee) approved all animal experiments performed in this study. All animal work was conducted in accordance with UK law. |

Note that full information on the approval of the study protocol must also be provided in the manuscript.

# Clinical data

Policy information about clinical studies
All manuscripts should comply with the ICMJE guidelines for publication of clinical research and a completed CONSORT checklist must be included with all submissions.

| Clinical trial registration | No active clinical trials were performed in this study. |
|---|---|
| Study protocol | No active clinical trials were performed in this study. |
| Data collection | Tissue sections were obtained from patients undergoing liver transplantation, with either alcohol-related liver disease, nonalcoholic fatty liver disease or hepatitis C-related liver disease (indicated in Supplementary Table 3). |
| Outcomes | No active clinical trials were performed in this study. |

# Plants

| | |
|---|---|
| Seed stocks | *Report on the source of all seed stocks or other plant material used. If applicable, state the seed stock centre and catalogue number. If plant specimens were collected from the field, describe the collection location, date and sampling procedures.* |
| Novel plant genotypes | *Describe the methods by which all novel plant genotypes were produced. This includes those generated by transgenic approaches, gene editing, chemical/radiation-based mutagenesis and hybridization. For transgenic lines, describe the transformation method, the number of independent lines analyzed and the generation upon which experiments were performed. For gene-edited lines, describe the editor used, the endogenous sequence targeted for editing, the targeting guide RNA sequence (if applicable) and how the editor was applied.* |
| Authentication | *Describe any authentication procedures for each seed stock used or novel genotype generated. Describe any experiments used to assess the effect of a mutation and, where applicable, how potential secondary effects (e.g. second site T-DNA insertions, mosiacism, off-target gene editing) were examined.* |

# Flow Cytometry

## Plots

Confirm that:

☒ The axis labels state the marker and fluorochrome used (e.g. CD4-FITC).

☒ The axis scales are clearly visible. Include numbers along axes only for bottom left plot of group (a 'group' is an analysis of identical markers).

☒ All plots are contour plots with outliers or pseudocolor plots.

☒ A numerical value for number of cells or percentage (with statistics) is provided.

## Methodology

| | |
|---|---|
| Sample preparation | RPE1-hTert and TIG3 cells grown in culture were trypsinised into suspension and immediately run on the flow sorter. Hepatocytes and immune cells were isolated from the liver using a collagenase-based dissociation protocol and immediately run on the flow sorter. |
| Instrument | Analysis of fluorochrome intensity was performed on a MacsQuantVYB (Miltenyi Biotech) for RPE1 and TIG3 experiments, and on a LSRFortessa Cell Analyzer or BD FacsSymphony (BD) for the mouse immune profiling experiments. Cell sorting was performed using a FACS Aria sorter (Becton Dickinson). |
| Software | Analysis was performed using FlowJo (v10.9.0, Becton Dickinson) |
| Cell population abundance | Abundance of cell populations post-sort ranged from 2% to 50% depending on application. Purity of fractions was determined by returning cells to culture and then re-running flow cytometry analysis 7-30 days post-analysis to demonstrate separation of subpopulations established by the flow sorting experiment. |
| Gating strategy | Debris was gated out using FSC-A/SSC-A gate, then singlets were selected for using FSC-H/FSC-A gate. For in vivo preparations, a live/dead dye was added for gating out unviable cells. |

☒ Tick this box to confirm that a figure exemplifying the gating strategy is provided in the Supplementary Information.

