## [Peer Review File · Nature]

Manuscript Title: Titration of RAS alters senescent state and influences tumour initiation

Editorial Notes:

Redactions – unpublished data

Reviewer Comments & Author Rebuttals

Reviewer Reports on the Initial Version:

Referees' comments:

Referee #1 (Remarks to the Author):

This study nicely shows the dependency of downstream phenotypes, from senescence to tumor initiation, on the dosage of RAS.

Oncogenic RAS proteins promote cell transformation leading to tumorigenesis. Paradoxically, activation of the RAS oncogene can also induce cellular senescence, an irreversible cell growth arrest, and an important tumor suppression mechanism in vivo. Here, the authors develop oncogenic RAS-dose titration models in vitro and in vivo, revealing a RAS-dose-driven non-linear continuum of downstream phenotypes. Using an inducible NRASG12V mouse liver model, authors show that low/ intermediate levels of oncogenic RAS promote progenitor-like features and malignant transformation, while high levels of oncogenic RAS induce cellular senescence, accompanied by immunosurveillance. Interestingly, titrating down the level of RAS induces immune resistance and subsequently tumor development.

This great study presents a very interesting, novel concept. I believe addressing the questions below will further straighten the conclusions and provide additional incites into the scope of this study.

Major conceptual questions:

I would suggest a better understanding of both the cell-autonomous (points 1,2) and non-cell-autonomous (points 3,4) effects of RAS dosage.

1. The authors show the dependency of cellular fate in early tumorigenesis on the levels of RAS expression, with low expression promoting undifferentiated tumors, and high leading to oncogene-induced senescence. However, authors also indicate that in both in vivo liver model (Fig. 1D) and in vitro model of RPE1 cells (Fig. 2L), oncogenic RAS promotes a unique progenitor state, a part of the OIS

phenotype. RAS expression was reported previously to predispose cells to secondary hits, which can ultimately promote senescence bypass and malignant transformations.

Can authors define in which case there will be a stronger clonal selection of RAS-induced cells leading to increased cell plasticity, development of a tumor, and worse survival?

2. Oncogene RAS can promote aberrant DNA replication leading to activation of the DNA damage response. Markers of such activation, like ATM, ATR, Chk1, Chk2, γ H2A, and 53BP1, are expressed during OIS. Could the extent of DNA damage response by a fate-determining factor controlling cells to either enter senescence or tumorigenesis? At least, the extent of activation of DNA damage response at different levels of RAS should be studied.

3. Authors show that immune cell recruitment to the hepatocytes with low RAS expression is intact (Fig. 3b - top, fig, S4a), however, senescence surveillance is weaker or absent in PGK- or UBC-NRASG12V mice. Why senescence surveillance of low-expressing RAS hepatocytes is impaired? Are there any differences in the expression of the elements that were previously reported to mediate immune surveillance in this model?

3. Senescence-associated secretory phenotype (SASP) is a cell non-autonomous mechanism by which senescent cells can promote tumorigenesis of neighboring premalignant cells. Can the level of RAS expression affect cellular plasticity and tumor-initiating capacity in a paracrine manner? Would cells with blocked inflammatory response still develop into cancer at a low level of RAS? In other words – is the effect you observe purely cell-autonomous?

Minor questions:

1. The sentence: “Thus, OIS can be viewed as a barrier to spontaneously increasing the dosage of RAS to the ‘critical’ level during tumor initiation” does not fit the main concept of the paper and thus might be misleading as a part of the introduction.

2. In the supplementary figure gating strategy is not clear.

3. Fig. 1C. shows increased and heterogenous expression of both NRAS and mVenus across observed clusters. However, from the presented data the conclusion that differences in NRAS dose is a primary driver of the clustering cannot be driven. Evidence that is more direct is necessary in order to support such a conclusion.

4. Data on Myc-target genes in Fig 1F, G shall be better substantiated in the main text explaining why it is presented and how it supports the message of Fig 1.

5. Myc-regulated transcriptional network consists of hundreds of genes spanning a variety of different cellular processes. It shall be explained what subset of Myc-target genes is defined as V1 and V2. What is the biological meaning of the upregulation of V1 gene sets in OIS cluster 4?

6. In Fig. 1H. legend is confusing. Shall it be designated as only "Non-OIS" or "OIS" respectively to the color of the bar?

Referee #2 (Remarks to the Author):

In this manuscript Adelyne S.L. Chan and colleagues discuss the role of oncogenic RAS-induced senescence (OIS) in an escalation model - and describe some mechanistic underpinnings in mice and men.

The authors tackle the question as to whether different doses of RAS would affect OIS and quality. As OIS requires a high level of oncogenic stress to be an autonomous tumor suppressor mechanism associated with pre-malignancy, it has remained unclear what lower oncogenic dosage would evoke.

Notably, when titrated down, NRASG12V-expressing hepatocytes became immune-resistant, and developed tumors. Since the relationship between OIS and tumor initiation remains elusive, modelling the phenotype by a range of oncogenic RAS levels in a normal or non-transformed diploid cellular context is of high interest, novel and conceptually very appealing.

Still, there are several issues that should be addressed and experiments needed to make the manuscript more clear and go a bit further in depth on the mechanistic underpinning.

Figure 1: The idea to look at the natural differences of RAS-dosage and how this translates into different OIS signatures, or signatures in general is very nice. It would be great to have a spatial overview (transcriptome) of this heterogeneity connecting RAS expression difference with specific targets or even immune cell status (e.g. CD4 T cell attraction – although this is not necessary in this context). Also a better visualization of the candidates – (e.g. by a heat map) would be fantastic.

Figure 2: Is convincing. It would be helpful to corroborate these findings also in other cells than RPE1.

Figure 3: An immune cell characterization of the liver tissue in relation to tumor development and RAS dosage would be important – this can on one hand be done directly on the tissue by histochemistry but also by flow cytometry analysis – to better define the immune cell subsets (e.g. T cells subsets) and numbers.

This would be interesting in relation to suppressive immune cells that could be generated by a lower RAS dosage. Also in relation to the reduced survival.

Can the authors show where in the livers Notch and TGFβ signaling is active?

Figure 4: In relation to the DLK1 or Notch positive TICs – what immunological environment are they surrounded by?

In regards to the dichotomous expression of NOTCH1 and DLK1: would it be possible to isolate the cells separately and characterize them transcriptionally – as well as functionally (transplantation)?

An increase of the human cohort as well as an analysis of cirrhotic patients with different etiology and history would be useful.

Referee #3 (Remarks to the Author):

Overall, this paper presents an interesting link between the level of RAS signaling and senescence. They have generated numerous scRNA-seq datasets from their mouse model that enables them to track RAS mutations directly to transcriptional phenotypes. While they present an interesting hypothesis and datasets, the bioinformatics analyses must be significantly strengthened and provide insufficient evidence of many of the key claims of this paper. Examples are provided below.

1. The link between RAS signaling and mutant RAS in the public domain data analyses is weak and not well connected to the –original dataset analyzed by the authors. Different genes are explored in each, and it's unclear whether the signatures are preserved across all genes evaluated in Fig 1d,e in the other datasets. The MSigDB hallmark geneset contains signatures associated with both KRAS up and downregulation. It's unclear how these are combined. Likewise, these often contain many pathways indirectly associated with RAS signaling due to activation of other growth factor receptors or parallel pathways associated with RAS. More direct analysis would be obtained from comparing the same genes between datasets and incorporation of transcription factor targets downstream of RAS (as was done for MYC) in addition to the hallmark pathways

2. There are concerns about the TCGA analysis presented in the manuscript. It is unclear why PDAC and HNSCC are the only two datasets selected, particularly when RAS mutations are not a common driver on HNSCC. Tumor types such as lung that have more pervasive RAS mutations would be valuable. Moreover, since these are bulk RNA-seq data, additional cell types in the TME including notably fibroblasts that are pervasive in these tumor types, may play a critical role on both RAS signaling and senescence. These must be considered in the analysis as well as the addition of other tumor types for evaluation and link to human data. Also, given that an HCC model is used it's unclear why that tumor type is only evaluated for survival and not the other mechanisms explored in the paper. Moreover, given the pervasiveness of RAS mutations in PDAC tumors and extensive public domain scRNA-seq data of epithelial cells in KRAS mutant human tumors (e.g., PDAC) as well as human HCC datasets, it's also unclear why TCGA is used instead of these widely available resources.

3. A major focus of this manuscript is the transition of cell cycle of RAS mutant populations. The many tools available for annotating cell cycle states from scRNA-seq data would be more appropriate than the pathway analyses used in many of the figures.

4. In Extended Data Fig 2 and Fig 2i-k, it's unclear how the genes were selected among the thousands that are statistically significant. It seems cherry picked vs falling out from the data.

5. There are concerns about the extensive reliance on pseudotime for many of the conclusions in this manuscript when it has been shown to be highly sensitive to the embedding and not always the optimal method for a given dataset. Evaluation of the robustness of the findings across multiple trajectory inference methods, such as dynverse, would better support the data.

Referee #4 (Remarks to the Author):

The study from Adelyne Chan et al., investigates the effects of different levels of oncogenic RAS on cellular senescence phenotypes. The researchers developed in vitro and in vivo models to examine the impact of varying RAS levels. In the liver model, different levels of RAS led to distinct hepatocyte clusters with characteristics of either senescence or progenitor-like cells. When RAS levels were decreased, the hepatocytes became immune-resistant and formed tumors. Time-series analysis identified two types of tumors: early-onset aggressive undifferentiated and late-onset differentiated hepatocellular carcinoma. Similar trends were observed in a pancreatic tumor model. This elegant study suggest that fine-tuning oncogenic RAS levels may have implications for cancer development and therapeutic interventions. Major limitations of the study are the poor report of the number of animals used, lack of depth in senescence and cancer-related characterizations and superficial description statistical analysis

More specifically:

1. Sample size and statistical analysis: The result section does not provide information about the sample size or statistical analysis performed in the study. The text briefly mentions control groups but does not provide detailed information about their criteria.
2. The text briefly mentions the implications of the findings in early tumorigenesis but does not provide a thorough discussion of the potential future directions and practical implications of the study. Potential therapeutic targets within the Ras/MAPK signaling pathway in HCC should be considered. It would be valuable to include experiments that test a therapeutic strategy targeting this pathway and how it may impact subclonal fitness and tumor progression.
3. Given that liver cancer affects males and females differently, it would be valuable to include a gender-specific analysis or speculation, to explore potential differences in the impact of RAS signaling and its role in tumorigenesis. This could involve stratifying the human data by gender and examining the relationship between RAS signaling, senescence, and tumor initiation separately for males and females. By comparing the molecular pathways activated by RAS and their consequences on senescence and tumor initiation in both sexes, the finding could shed light on the gender-specific factors that contribute to liver cancer development.

Minor points:

Text Line 320. Not clear sentence-please reformulate

Text Line 322. This sentence is too speculative, given that cirrhosis doesn't develop from RAS mutations.

Author Rebuttals to Initial Comments:

RESPONSE TO REVIEWERS' COMMENTS

We thank the reviewers for their constructive comments on our manuscript. They have been very helpful in strengthening our manuscript. We are particularly excited to have a much clearer view of the two distinct types (Notch1-type and Dlk1-type) of tumour-initiating cells (TICs) and their immune connections (**Reviewer fig. 1**) - this is relevant with multiple points from all reviewers.

Point-by-point responses they may later progress into distinct types of tumours

Referee #1 (R1):

This study nicely shows the dependency of downstream phenotypes, from senescence to tumor initiation, on the dosage of RAS.

Oncogenic RAS proteins promote cell transformation leading to tumorigenesis. Paradoxically, activation of the RAS oncogene can also induce cellular senescence, an irreversible cell growth arrest, and an important tumor suppression mechanism in vivo. Here, the authors develop oncogenic RAS-dose titration models in vitro and in vivo, revealing a RAS-dose-driven non-linear continuum of downstream phenotypes. Using an inducible NRAS^{G12V} mouse liver model, authors show that low/ intermediate levels of oncogenic RAS promote progenitor-like features and malignant transformation, while high levels of oncogenic RAS induce cellular senescence, accompanied by immunosurveillance. Interestingly, titrating down the level of RAS induces immune resistance and subsequently tumor development.

This great study presents a very interesting, novel concept. I believe addressing the questions below will further straighten the conclusions and provide additional incites into the scope of this study.

Major conceptual questions:

I would suggest a better understanding of both the cell-autonomous (points 1,2) and non-cell-autonomous (points 3,4) effects of RAS dosage.

1. The authors show the dependency of cellular fate in early tumorigenesis on the levels of RAS expression, with low expression promoting undifferentiated tumors, and high leading to oncogene-induced senescence. However, authors also indicate that in both in vivo liver model (Fig. 1D) and in vitro model of RPE1 cells (Fig. 2L), oncogenic RAS promotes a unique progenitor state, a part of the OIS phenotype. RAS expression was reported previously to predispose cells to secondary hits, which can ultimately promote senescence bypass and malignant transformations. Can authors define in which case there will be a stronger clonal selection of RAS-induced cells leading to increased cell plasticity, development of a tumor, and worse survival?

This is a fundamental question in tumour initiation biology. This probably depends on the nature of the secondary hit. For example, in our CAGGS-NRAS^{G12V} (high RAS) + Notch1-ICD (constitutive active Notch1 mutant) cohort, the mice develop aggressive tumours with strong NRAS staining, suggesting that Notch1-ICD-expressing tumour-initiating cells (TICs) escape oncogene-induced senescence (OIS), thus high RAS can have a selective advantage in the presence of active Notch signalling (Extended Data Fig. 8a).

This is consistent with other data, showing that early-onset Notch1-type tumours express higher RAS compared to late-onset differentiated Dlk1-type tumours (e.g. Fig. 4a). Notably, in fetal liver development, Notch promotes cholangiocyte differentiation from bi-directional hepatoblasts, whereas Dlk1, a non-canonical Notch ligand with a Notch inhibitory activity, is associated with hepatocyte differentiation¹. Although our Notch-type tumours were pathologically undifferentiated and lacked obvious cholangio features, they nonetheless showed common molecular (e.g. CK19+, activated RAS) and clinical (aggressiveness) features with cholangio components in primary liver cancer. We speculate that secondary hits associated with enhanced Notch signalling (and perhaps other related factors such as TGFβ and Nestin) may promote clonal selection for those high-RAS cells with a distinct progenitor-like state.

In addition to the cell-autonomous aspect, our new data suggests that Notch signalling may also affect the local immune reaction. In this mouse model, oncogenic NRAS-expressing hepatocytes often attract immune cells, forming immune cell clusters, which persist in PGK/UBC-NRAS^{G12V} livers beyond day 12 (Extended Data Fig. 6b-d). Interestingly, most of the Notch1-type, but not Dlk1-type, hepatocytes were involved in the persisting immune cell clusters (**Reviewer fig. 1**; Fig. 3k, Extended Data Fig. 8b-c), suggesting that Notch1-type hepatocytes are immunogenic but macrophages within the immune ‘niche’ are unable to eliminate them. This is consistent with our previous study showing that Notch signalling has inhibitory effects on OIS surveillance in the CAGGS-NRAS^{G12V} mice². Moreover, inhibiting RAS/MAPK signalling in PGK/UBC-NRAS^{G12V} mice with Sorafenib (a multi-kinase inhibitor targeting RAF and a number of upstream receptor tyrosine kinases³) led to reduced macrophage infiltration/aggregation, which is associated with Notch1+ hepatocytes (Extended Data Fig. 8d) (see also our response to R4-#2).

Reviewer fig. 1. Distinct immune microenvironments in two types of TIC candidates. Modified from Fig. 3k and Extended Data Fig. 8c (mice) and Fig. 4e (humans) (see also Extended Data Fig. 10c for humans).

Together, it is likely that second hits that modulate Notch signalling may affect clonal selection both in a cell-autonomous and non-cell-autonomous manner. We have included these new data and edited the results accordingly.

2. Oncogene RAS can promote aberrant DNA replication leading to activation of the DNA damage response. Markers of such activation, like ATM, ATR, Chk1, Chk2, γ H2A, and 53BP1, are expressed during OIS. Could the extent of DNA damage response by a fate-determining factor controlling cells to either enter senescence or tumorigenesis? At least, the extent of activation of DNA damage response at different levels of RAS should be studied.

To address this question in the mouse liver model, we first leveraged our scRNA-seq data (day 6). We have employed DNA damage-related genesets from the entire MSigDB (not limited to 'hallmarks') and computed enrichment scores across the 4 cell clusters with different *NRAS* levels. The heatmap representation in the new Extended Data Fig. 1e shows that DNA damage-related genesets were often upregulated in high-RAS Clusters 3 and 4, although the changes are subtle.

Next, we conducted co-immunofluorescence of γ H2AX and *NRAS* in mouse livers injected *NRAS*^{G12V} under different promoters (day 6). We observed no significant increase in the frequency of cells with DNA damage foci, compared to non-oncogenic *NRAS*^{G12V/D38A}-expressing hepatocytes, or non-transduced hepatocytes in the same livers, at least day 6 (Extended Data Fig. 6a). This lack of significance may be due to the heterogeneity of RAS level, particularly in CAGGS livers (see Fig. 3b, *NRAS* intensity).

We also examined our In vitro systems, RPE1 and new TIG3 cells. TIG3 is a human diploid fibroblast line, and as expected, TIG3 cells exhibited a more typical OIS phenotype in high-RAS conditions (new Extended Data Fig. 3c-e, please also refer to our response to R2-#2), compared to RPE1 cells, which displayed a slow-cycling phenotype. Consistently, we observed an increase in the intensity of γ H2AX immunofluorescence (a common OIS feature) in high-RAS-TIG3 cells. In contrast, we failed to detect a similar increase in γ H2AX intensity in RPE1 cells (Extended Data Fig. 3f). We find this very interesting, and potentially relevant in the senescence spectrum. Notably, in the liver model mentioned above, some DNA damage-related genesets appeared lower in Cluster 2 (which is enriched for hepatoblast/progenitor features) (Extended Data Fig. 1e). Although a more comprehensive investigation is required in the liver model, our in vitro data support the OIS spectrum concept, and it is tempting to speculate that low DDR might be associated with slow-cycling (low metabolic burden)/tumour initiating states in our models. We have added these new results and edited the main text accordingly. We thank the reviewer for drawing our attention to this point.

3. Authors show that immune cell recruitment to the hepatocytes with low RAS expression is intact (Fig. 3b - top, fig, S4a), however, senescence surveillance is weaker or absent in PGK- or UBC-*NRAS*^{G12V} mice. Why senescence surveillance of low-expressing RAS hepatocytes is impaired? Are there any differences in the expression of the elements that were previously reported to mediate immune surveillance in this model?

This is an important question (please also see our responses to R2-#1, #3). In this mouse liver model, senescence surveillance involves both CD4⁺ T cells and macrophages. In the original manuscript, although UBC/PGK-*NRAS*^{G12V}-expressing hepatocytes recruit immune cells, they were resistant to immune surveillance, leading to the accumulation of persistent immune cell clusters around RAS-expressing hepatocytes beyond day 12 (note, by this time point, most CAGGS-driven high-RAS hepatocytes are eliminated).

To directly compare these three RAS dosages, we have performed immune cell profiling using flow cytometry at day 9, the typical time-point where the high CAGGS-NRAS^{G12V} dosage exhibits the most prominent immune reaction⁴. Consistent with previous studies⁵, we detected significant recruitment of CD4+ and CD8+ T-cells, NK cells, immature monocytes (iMC), and macrophages in the CAGGS-NRAS^{G12V} livers (new Fig. 3e). Strikingly, however, such immune cell recruitment was not apparent in PGK- or UBC-NRAS^{G12V} livers, except for myeloid cells, suggesting that the immune response in UBC/PGK-RAS dosages is generally weaker but sustained.

To gain mechanistic insight, we have conducted a more comprehensive analysis of differentially expressed genes in our scRNA-seq data (the CAGGS-NRAS^{G12V} cohort d6 - Fig. 1), focusing on the secretome. Hepatocytes in Clusters 3 and 4 (with relatively high levels of NRAS^{G12V}) expressed genes associated with the 'Cytokine-cytokine receptor interaction' term, including inflammatory SASP genes, such as *Il1a*, *Il1b*, and *Ccl2* (new Extended Data Fig. 1d).

More specifically, *Ccl2* secretion from NRAS^{G12V}-expressing hepatocytes is indeed known to be required for the recruitment of iMC and thus senescence surveillance in this liver OIS model⁶. In addition, our recent study showed that *Ptgs2/Cox2* is also critical for senescence surveillance in this model: we showed that *Cox2* loss in CAGGS-NRAS^{G12V} condition results in the accumulation of immunosuppressive Treg⁴. *Cox2* is an enzyme involved in generating prostaglandins and modulates the inflammatory SASP. Notably, *Ptgs2/Cox2* was only detected in the Cluster 4 (OIS) hepatocytes in our scRNA-seq data (we have replaced *Cxcl16* with *Ptgs2* in Fig. 1d). Consistently, Treg cells (FoxP3 IHC) appeared to progressively accumulate in PGK/UBC-NRAS^{G12V} livers (new Extended Data Fig. 6d).

Together, these suggest that weaker activation of the inflammatory SASP and progressive accumulation of Treg cells at lower levels of NRAS^{G12V}-expressing hepatocytes might in part contribute to their immune resistance.

We have added these new results and edited the text accordingly.

4. Senescence-associated secretory phenotype (SASP) is a cell non-autonomous mechanism by which senescent cells can promote tumorigenesis of neighboring premalignant cells. Can the level of RAS expression affect cellular plasticity and tumor-initiating capacity in a paracrine manner? Would cells with blocked inflammatory response still develop into cancer at a low level of RAS? In other words – is the effect you observe purely cell-autonomous?

This is a good point. As shown in Fig. 3d, the low level of NRAS^{G12V} can lead to tumour development even in SCID mice, supporting the autonomous tumorigenic effects. However, the overall phenotype was much milder in SCID mice. We are tempted to speculate that the immune response in the low-RAS context might also contribute to tumorigenesis in this context, although the SCID mice are a different strain background, thus the interpretation needs experimental validation.

Nevertheless, as discussed in our response to R1-major point #3 and R2-#3, sub-OIS PGK/UBC-NRAS^{G12V} hepatocytes tended to form persistent immune cell clusters and these

immune clusters contained abundant F4/80+ macrophage-like cells (new Extended Data Fig. 6c). The exact nature and origin of these macrophages with no apparent phagocytic activity remain to be elucidated. It is possible that they might contribute to a tumour-initiating cell (TIC) 'niche', akin to tumour-associated macrophages (TAMs). Thus, we do not exclude the synergistic tumour-promoting effects between autonomous and non-autonomous activities in the sub-OIS-RAS setting. Dissection of the 'TIC niche' will be a major focus in the future. We have added this point to the discussion section in the revised manuscript.

Minor questions:

1. The sentence: "Thus, OIS can be viewed as a barrier to spontaneously increasing the dosage of RAS to the 'critical' level during tumor initiation" does not fit the main concept of the paper and thus might be misleading as a part of the introduction.

We agree. We have removed this sentence. We thank the reviewer for pointing this out.

2. In the supplementary figure gating strategy is not clear.

In addition to the original hepatocyte sorting, we have added gating/sorting strategies for in vitro systems (RPE1 and TIG3 cells) and immune cell profiling in new Supplementary Information.

3. Fig. 1C. shows increased and heterogenous expression of both NRAS and mVenus across observed clusters. However, from the presented data the conclusion that differences in NRAS dose is a primary driver of the clustering cannot be driven. Evidence that is more direct is necessary in order to support such a conclusion.

An alternative way to represent this single-cell data, similar to that of the tumorigenesis dataset used later in the paper, is to order cells in pseudotime. Progression along the pseudotime corresponded well with the clusters shown in the original figure (the pseudotime values of Cluster 1 < Cluster 2 < Cluster 3 < Cluster 4). Unbiased analysis of the genes driving this pseudotime shows *NRAS* in the top 50 hits. We have now included this information as new Extended Data Fig. 1a-c.

4. Data on Myc-target genes in Fig 1F, G shall be better substantiated in the main text explaining why it is presented and how it supports the message of Fig 1.

To substantiate the relevance of Myc targets, we have added other transcription factor (TF) signatures (also in our response to R3-#1). Notably, MYC is a direct downstream TF in the RAS-MAPK/ERK pathway, where MYC is a nuclear substrate of ERK⁷. Signatures of other TFs that are nuclear substrates of ERK or downstream kinases showed a largely linear pattern along the cell clusters, highlighting the unique non-linear relationship between RAS and MYC signatures (new Extended Data Fig. 1e). We have edited the main text to include these new results.

In addition, we have conducted the same analysis in our RPE1 datasets and mouse squamous cell carcinoma models. Again, only MYC signatures exhibited an overall reduction in

oncogenic RAS-expressing cells (new Extended Data Fig. 5a, e), supporting the unique role of MYC in these RAS-driven slow-cycling cells.

5. Myc-regulated transcriptional network consists of hundreds of genes spanning a variety of different cellular processes. It shall be explained what subset of Myc-target genes is defined as V1 and V2. What is the biological meaning of the upregulation of V1 gene sets in OIS cluster 4?

Notably, the correlation between RAS dosage and Myc signature was well conserved in both V1 and V2 at the single cell level, thus the data reinforcing the unique pattern of the correlation - generally a negative correlation but a subset of Myc targets were increased in OIS clusters. We have reinforced this point in the revised manuscript.

Here is some additional background about these two versions of MYC targets in MSigDB Hallmark genesets⁸. Hallmark genesets are a computational refinement of a group of existing 'founder' genesets encompassing both ontology-based and perturbation-based genesets. Different founder sets yield MYC_V1 and V2, which partially overlap. Both have been widely utilised: both correlate well with cell proliferation and cancer aggressiveness in clinical datasets⁹.

The biological meaning of the upregulation of specific Myc-targets in OIS remains unclear. Importantly, as shown in Extended Data Fig. 5b, Myc-targets are invariably downregulated in cell cycle arrest states (quiescence and other types of senescence). The only exception was OIS, in which Myc targets were largely intact or upregulated. MYC is a master regulator of biosynthesis and reduced MYC activity has been implicated in slow-cycling (low metabolic) states. We speculate that high MYC activity causes a high metabolic load and cellular stress, thus promoting senescence in the presence of mitotic signalling. Therefore, the MYC activity might contribute to the switch between slow-cycling and OIS states in RAS-expressing cells. This will be an interesting direction for follow-up studies in the future.

6. In Fig. 1H. legend is confusing. Shall it be designated as only "Non-OIS" or "OIS" respectively to the color of the bar?

We have changed the colour bar annotation in the figure panel to "Non-OIS" and "OIS" and edited the legend accordingly.

Referee #2 (R2):

In this manuscript Adelyne S.L. Chan and colleagues discuss the role of oncogenic RAS-induced senescence (OIS) in an escalation model - and describe some mechanistic underpinnings in mice and men.

The authors tackle the question as to whether different doses of RAS would affect OIS and quality. As OIS requires a high level of oncogenic stress to be an autonomous tumor suppressor mechanism associated with pre-malignancy, it has remained unclear what lower oncogenic dosage would evoke.

Notably, when titrated down, NRASG12V-expressing hepatocytes became immune-resistant, and developed tumors. Since the relationship between OIS and tumor initiation remains

elusive, modelling the phenotype by a range of oncogenic RAS levels in a normal or non-transformed diploid cellular context is of high interest, novel and conceptionally very appealing.

Still, there are several issues that should be addressed and experiments needed to make the manuscript more clear and go a bit further in depth on the mechanistic underpinning.

1. Figure 1: The idea to look at the natural differences of RAS-dosage and how this translates into different OIS signatures, or signatures in general is very nice. It would be great to have a spatial overview (transcriptome) of this heterogeneity connecting RAS expression difference with specific targets or even immune cell status (e.g. CD4 T cell attraction – although this is not necessary in this context). Also a better visualization of the candidates – (e.g. by a heat map) would be fantastic.

This is an important question, particularly considering the persistent immune clusters around PGK/UBC-NRAS^{G12V} hepatocytes (e.g. Extended Data Fig. 6b).

We have examined the spatial relationship of Dlk1+ or Notch1+ hepatocytes with immune cell clusters with immunohistochemistry (IHC). We have utilised serially sectioned images as much as possible. Interestingly, most Notch1-positive hepatocytes were involved in the immune cell clusters, whereas Dlk1-positive hepatocytes were largely excluded (**Reviewer fig. 1A**). Remarkably, a similar trend was also found in human cirrhotic livers (**Reviewer fig. 1B**). Consistent with our previous study², sustained activation of Notch signalling may contribute to the immune resistance of PGK/UBC-NRAS^{G12V} hepatocytes. These new results (Fig. 3k, Fig. 4e, Extended Data Fig. 8c, and Extended Data Fig. 10c) further emphasise the distinct nature of the two types of TICs (see also our response to R2-#5).

These persisting immune cells were mainly a mixture of macrophage/myeloid and T cells. Interestingly, immunosuppressive Treg (FoxP3 IHC) cells appeared to progressively accumulate in those immune cell clusters in PGK/UBC-NRAS^{G12V} livers (new Extended Data Fig. 6c, d).

We also used flow cytometry for further quantification. Consistent with previous studies⁵, we detected significant recruitment of CD4+ and CD8+ T-cells, NK cells, immature monocytes (iMC), and macrophages in the CAGGS-NRAS^{G12V} livers at day 9. Strikingly, however, such immune cell recruitment was not apparent in PGK- or UBC-NRAS^{G12V} livers, except for myeloid cells. These data suggest that the overall immune cell recruitment was weaker in PGK/UBC-NRAS^{G12V} compared to the classical high CAGGS-NRAS^{G12V} context but Notch1-type hepatocytes, in particular, are involved in persistent immune cell clusters (day 12).

In addition, as suggested, we have added a new heatmap representation of secretory factors^{2,10} across cell-clusters (Clusters 1-4) derived from our scRNA-seq (day 6, Fig. 1b). Pathway enrichment performed on genes grouped based on their expression level across the cell-clusters showed different functional categories being associated with different levels of NRAS^{G12V} (new Extended Data Fig. 1d, also our response to R1-major point #3). Consistent with our immune cell profiling, factors associated with “Cytokine-cytokine receptor interaction” were enriched in the higher NRAS^{G12V}-expressing cells (Clusters 3 and 4). In addition, the term, “Complement and coagulation cascade”, was enriched in lower NRAS^{G12V}-expressing cells (Cluster 2), reinforcing the functional augmentation in these hepatocytes.

These new findings have now been incorporated into the main text. We thank the reviewer for raising these questions (also R2-#5).

2. Figure 2: Is convincing. It would be helpful to corroborate these findings also in other cells than RPE1.

This is a valid point. Although the flow sorting procedure is highly stressful for 'normal' diploid cells, we have managed to establish a TIG3 (human diploid fibroblasts) model using the same 'predictable reporter' system. In a mixed population of TIG3 with a wide range of HRAS^{G12V} dosage, the survival benefit of the low-RAS cells was recapitulated and high-RAS TIG3 cells show the classic senescence-like phenotype, including reduced proliferation, increased SA- β -gal activity, and prominent inflammatory secretory phenotype, such as IL6 and IL8. These results provide additional support for the existence of the OIS spectrum. These new data can be found in Extended Fig. 3c-f and we have edited the main text.

3. Figure 3: An immune cell characterization of the liver tissue in relation to tumor development and RAS dosage would be important – this can on one hand be done directly on the tissue by histochemistry but also by flow cytometry analysis – to better define the immune cell subsets (e.g. T cells subsets) and numbers.

This would be interesting in relation to suppressive immune cells that could be generated by a lower RAS dosage. Also in relation to the reduced survival.

This is related to R1-major point #3 and R2-#1. In this mouse model, the high-NRAS^{G12V}-driven classical OIS hepatocytes are eliminated via senescence surveillance (mostly by day 12), which involves both CD4+ T cells and macrophages⁵.

We have conducted new experiments to profile immune cells through flow cytometry in response to different dosages of NRAS^{G12V} in mouse livers. Interestingly, unlike the classical OIS (CAGGS-driven NRAS^{G12V}) model, sub-OIS (PGK- and UBC-driven NRAS^{G12V}) hepatocytes induced minimum T-cell response at day 9 after NRAS^{G12V} introduction (the typical timepoint for immune profiling in this liver OIS model). Thus, overall T cell recruitment was reduced in PGK/UBC-NRAS^{G12V} livers.

We have also assessed Treg cells using Foxp3 IHC. While there was no accumulation in all conditions at day 9, Treg cells appeared to progressively accumulate in those immune clusters in PGK/UBC-NRAS^{G12V} contexts at day 12 (Extended Data Fig. 6d).

This is consistent with the weaker (or even lack of) induction of *Ccl2* and *Ptgs2/Cox2* in sub-OIS hepatocyte clusters in scRNA-seq, since we and others have previously shown that these are essential for senescence surveillance in this model (also see our response to R1-major #3)⁶. These new immune cell profiling data can be found in Fig. 3e and Extended Data Fig. 6c-d.

4. Can the authors show where in the livers Notch and TGF β signaling is active?

Our scRNA-seq data suggested a close correlation between Notch and Tgfb signalling, represented by the expression of *Notch1* and *Tgfb1*, respectively, in the early onset tumour type. Thus, we expect that both these signalling pathways are active in this type of tumour. To confirm this, we have conducted new IHC staining of Notch1, Tgfβ1, and Dlk1 in these two types of tumours, showing that Notch1 and Tgfβ1 co-stained in the early-onset aggressive tumours (5 out of 6 Notch1-positive tumours were Tgfβ1-positive). Representative figures are now included as new Extended Data Fig. 9c.

5. Figure 4: In relation to the DLK1 or Notch positive TICs – what immunological environment are they surrounded by?

We thank the reviewer for raising this critical point; new data substantially improved our model.

We first asked this in our mouse liver model. As mentioned in our response to R2-#1 and #3, in PGK/UBC-NRAS^{G12V} livers, immune cell clusters persisted around NRAS-expressing hepatocytes beyond day 12. Strikingly, those hepatocytes were mostly positive for Notch1 staining, whereas Dlk1 were positive in NRAS-expressing hepatocytes outside these immune cell clusters, reinforcing the distinct nature of these two types of TICs (**Reviewer fig. 1A**).

Then we have reevaluated all the human tissues (we have now added 15 new patients with hepatitis C virus, HCV - see R2-#7 below) with serial section IHC using markers for T cells (CD4, CD8) and myeloid cells/macrophages (CD68) alongside NOTCH1 and DLK1. Strikingly, we have found that NOTCH1-positive, but not DLK1-positive cells, were highly associated with clusters of those immune cells (**Reviewer fig. 1B**).

These data together suggest that Notch1-positive cells were more immunogenic but resistant to immune surveillance. This is consistent with our previous study, showing that blocking Notch signalling in RAS-expressing hepatocytes promotes senescence surveillance². These new results can be found in Fig. 3k and Extended Data Fig. 8c for mice and Fig. 4e and Extended Data Fig. 10c for human samples.

6. In regards to the dichotomous expression of NOTCH1 and DLK1: would it be possible to isolate the cells separately and characterize them transcriptionally – as well as functionally (transplantation)?

This is an interesting idea, particularly for functional assays. However, flow sorting hepatocytes using cell surface markers (e.g. Dlk1 or Notch1) have been technically challenging possibly due to the involvement of the enzymatic digestion during the single hepatocyte isolation procedure, disrupting cell surface protein epitopes. Indeed, despite multiple attempts, we detected intracellular, but not cell surface, Dlk1 after single cell isolation in our mouse model. In addition, cell transplantation of isolated hepatocytes to the liver via trans-spleen injection typically uses transformed/liver cancer cells⁶, and injecting directly isolated primary cells is also highly challenging.

Therefore, to further dissect the difference between these two types of TICs, we instead have focused on their spatial relationships with immune cells as was also suggested by this reviewer. As mentioned above (R2-#1, #5), Notch1-expressing hepatocytes, but not Dlk1-

expressing hepatocytes, were typically involved in the persistent immune cell clusters. This trend was strikingly recapitulated in human cirrhotic livers.

As discussed in our response to R4-#2, these data highlight a potential functional and clinical relevance. Early treatment of the RAS-injected mice with Sorafenib, a multi-kinase inhibitor, led to a reduced level of pERK1/2 and a significant decline of F4/80+ macrophage/myeloid cell aggregation, suggesting a reduction in Notch1-type TICs (new Extended Data Fig. 8d).

We are currently establishing a Dlk1-lineage tracing mouse model¹¹, which, in conjunction with our PGK/UBC-NRAS^{G12V} liver model, would allow us to monitor Dlk1-type TICs and selectively eliminate them. We are hoping to address this key question in the future.

7. An increase of the human cohort as well as an analysis of cirrhotic patients with different etiology and history would be useful.

We have included 15 more cirrhotic patients with different aetiology (Hepatitis C Virus, HCV) on top of the original 13 patients (steatotic liver disease, including alcoholic liver disease and non-alcoholic fatty liver diseases). We identified positive DLK1 staining within hepatocytes of 17 out of 28 cirrhotic human livers, whereas NOTCH1 staining was identified in 15 out of 28. Notably, those 9 patients exhibited positive staining for both markers, but in spatially different regions.

In addition, we have reevaluated all samples, including the original ones, for both NOTCH1 and DLK1 IHC with immune markers using serially sectioned samples (new Fig. 4e, Extended Data Fig. 10c) (please see our response to R2-#5).

Referee #3 (R3):

Overall, this paper presents an interesting link between the level of RAS signaling and senescence. They have generated numerous scRNA-seq datasets from their mouse model that enables them to track RAS mutations directly to transcriptional phenotypes. While they present an interesting hypothesis and datasets, the bioinformatics analyses must be significantly strengthened and provide insufficient evidence of many of the key claims of this paper. Examples are provided below.

1. The link between RAS signaling and mutant RAS in the public domain data analyses is weak and not well connected to the original dataset analyzed by the authors. Different genes are explored in each, and it's unclear whether the signatures are preserved across all genes evaluated in Fig 1d,e in the other datasets. The MSigDB hallmark geneset contains signatures associated with both KRAS up and downregulation. It's unclear how these are combined. Likewise, these often contain many pathways indirectly associated with RAS signaling due to activation of other growth factor receptors or parallel pathways associated with RAS. More direct analysis would be obtained from comparing the same genes between datasets and incorporation of transcription factor targets downstream of RAS (as was done for MYC) in addition to the hallmark pathways.

We have broken down this into three points:

i) Choice of genes/gene sets across different datasets: Senescence is highly diverse between cell types and triggers. While some markers, such as upregulation of CDK inhibitors (*CDKN1a/p21*, *CDKN2a/p16*, and *CDKN2b/p15*), are relatively conserved, it is advised to combine them since no single markers are common in all types of senescence¹². Other senescence markers, particularly non-autonomous factors (such as secretory factors), are more cell-type specific¹³. Moreover, some progenitor features (plasticity) are also highly cell-type specific, e.g. the ‘hepatoblasts feature’ for the liver (Fig. 1e) and *Pdx1* for the pancreas (Extended Data Fig. 2c), while others are relatively pervasive (e.g. *Prom1* and *Notch1*) (*Notch1* is now added to the pancreatic data in Extended Data Fig. 2c).

Thus, the senescence spectrum involves both relatively common and tissue-specific gene sets: the latter, in particular, is associated with cell functionality and cell identity/plasticity. Different genes were explored in different datasets, as the data shown are derived from a variety of cell types. In the revised manuscript, we have expanded examples to include the same genes (gene sets) whenever appropriate (e.g. Fig. 1, Extended Data Fig. 2c).

ii) KRAS signatures: We apologise for the oversight in labels on the figure and thank the reviewer for pointing this out. Data from the signatures on the original figure show KRAS_SIGNALLING_UP, as a readout for KRAS upregulation, without combining. We have clarified this in our revised manuscript (Fig. 1g, legend).

iii) RAS-MAPK/ERK downstream transcription factor (TF) analysis: This is a good point. Indeed, MYC is a direct nuclear substrate of ERK. We have performed additional TF analysis in the scRNA-seq dataset from the liver, focusing on known TFs that are nuclear substrates of ERK and/or downstream kinases (e.g. RSK, MSK) and can be regulated in an ERK-dependent manner. This includes AP1, ETS1, SP1, ELK1, CREB, HIF1, ATF3, STAT1 and STAT3⁷. Interestingly, unlike the MYC activity, which was rather reduced in low-RAS clusters (Fig. 1f), these TFs’ activities were largely linear along the cell clusters (new Extended Data Fig. 1e).

Similarly, in our RPE1 cells and mouse squamous cell carcinoma models, which represent RAS-induced slow-cycling states, only MYC signatures exhibited an overall reduction in oncogenic RAS-expressing cells (new Extended Data Fig. 5a, e).

These new data support the unique correlation between reduced MYC activity and RAS-driven tumour-initiating and slow-cycling cells. We have incorporated these new findings into the revised manuscript. We thank the reviewer for this suggestion.

2. There are concerns about the TCGA analysis presented in the manuscript. It is unclear why PDAC and HNSCC are the only two datasets selected, particularly when RAS mutations are not a common driver on HNSCC. Tumor types such as lung that have more pervasive RAS mutations would be valuable. Moreover, since these are bulk RNA-seq data, additional cell types in the TME including notably fibroblasts that are pervasive in these tumor types, may play a critical role on both RAS signaling and senescence. These must be considered in the analysis as well as the addition of other tumor types for evaluation and link to human data. Also, given that an HCC model is used it’s unclear why that tumor type is only evaluated for survival and not the other mechanisms explored in the paper. Moreover, given the pervasiveness of RAS mutations in PDAC tumors and extensive public domain scRNA-seq

data of epithelial cells in KRAS mutant human tumors (e.g., PDAC) as well as human HCC datasets, it's also unclear why TCGA is used instead of these widely available resources.

As suggested, we have extended our analysis to link our data and public datasets.

Our main focus is preneoplasia and tumour-initiating cells (TICs) and such data are highly limited in humans. Thus, in advanced cancer datasets, we only focused on survival as a clinical readout. Because of this, we have used TCGA as they provide prognostic information on larger patient numbers, as opposed to scRNA-seq datasets which (a) tend to only have small patient numbers per dataset and (b) are generally less well-annotated with clinical information. We agree that the focus on PDAC and HNSCC was too narrow, and as such, we have added other tumour types including lung in the TCGA analysis (new Extended Data Fig. 2d). The data are largely consistent except for colon adenocarcinoma (COAD). The reason for this is currently unclear.

As suggested, we have explored public human cancer scRNA-seq datasets: we have identified two datasets (lung and PDAC), which are particularly relevant in our study. Consistent with the bulk TCGA analysis, KRAS level tended to be upregulated in cancer compared to normal adjacent cells (new Extended Data Fig. 2e, f).

TME (tumour microenvironment) is an interesting question and 'TME' in preneoplastic contexts is still an emerging research area. In the revised manuscript, we have added new data focusing on potential TICs and immune components in human and mouse livers. As discussed in our response to R2-#5, we found that the two types of TIC candidates are associated with distinct immune microenvironments (e.g. **Reviewer fig. 1**). Notch-type was particularly involved in persistent immune cell clusters. These new results can be found in Extended Data Fig. 6b-d, 8b-c (mice) and Fig. 4e and Extended Data Fig. 10c (humans).

[REDACTED]

As suggested by this reviewer, we also checked fibroblastic features: the immune clusters around NRAS^{G12V}-expressing hepatocytes were highly fibrotic (collagen-rich), involving a-SMA+ 'fibroblasts'. These cells could be hepatic stellate cells (liver resident fibroblasts). It is possible that these cells play key roles in the formation of the 'TIC niche'. In addition, our previous study suggested that liver sinusoidal endothelial cells (LSECs) serve as amplification machinery of OIS surveillance in mouse livers¹⁴. The interaction between distinct TICs and these various stromal components will be a future focus, using spatial omics approaches.

3. A major focus of this manuscript is the transition of cell cycle of RAS mutant populations. The many tools available for annotating cell cycle states from scRNA-seq data would be more appropriate than the pathway analyses used in many of the figures.

We use those tools routinely but we find it often difficult to apply them to datasets mainly consisting of non-/slow-cycling cells, like our liver data. Most of the cell cycle phase genesets are derived from synchronised bulk¹⁵ and single-cell¹⁶ experiments in established cell lines, not in a native tissue context. Particularly for us, it is difficult to distinguish between G0 (nearly all normal hepatocytes), OIS and TICs by these tools. As shown in **Reviewer fig. 3A**, the ‘E2F-targets’ signature (MSigDB) in our scRNA-seq data from CAGGS-NRAS^{G12V} liver (day 6) shows low expression in most cells as we previously showed that very few hepatocytes incorporate EdU (5-ethynyl-2'-deoxyuridine) in this context⁴. We also annotated cell cycle states^{17,18}, showing a random pattern (**Reviewer fig. 3A**).

Reviewer fig. 3. A. Cell cycle annotation in liver scRNA-seq (CAGGS-NRAS). Not included in the manuscript.

B. Cell cycle annotation in RPE1 scRNA-seq. **C.** Negative correlation between RAS and MYC signatures. Modified from Extended Data Fig. 5c, d.

However, as suggested by the reviewer, we agree that cell cycle annotation might provide additional insights into our culture model, which recapitulates the transition from highly proliferative to non-/slow-cycling states (Fig. 2). We have generated new scRNA-seq data in RPE1 cells for the main subpopulations (S, M, L, XL, and controls). Here, we find a highly consistent pattern between cell cycle annotation and cell cycle genes (e.g. ‘E2F-targets’) (**Reviewer fig. 3B**). Interestingly, in tSNE space, RAS signalling and cell cycle profile were orthogonal, wherein an increased percentage of cells in G1(G0)-phase with residual S-phase cells were found in higher HRAS^{G12V} subpopulations. This is consistent with our original cell biology results. In bulk RNA-seq (Fig. 2j, k), both MYC and E2F (cell cycle) targets were similarly downregulated in RAS-expressing RPE1 cells, and indeed these two signatures do have some overlap. However, at the single-cell level, MYC targets appeared more correlative with the RAS signature than the E2F signature/cell cycle profile. Thus reduced MYC activity is not simply due to reduced cell proliferation in this case, reinforcing the negative correlation between RAS and MYC signatures in the sub-OIS contexts (**Reviewer fig. 3C**). New RPE1 scRNA-seq data can be found in Extended Data Fig. 5c-d, and we have edited the main text accordingly.

4. In Extended Data Fig 2 and Fig 2i-k, it's unclear how the genes were selected among the thousands that are statistically significant. It seems cherry picked vs falling out from the data.

Original Extended Data Fig. 2 (now Extended Data Fig. 4a-c) is an unbiased analysis of differentially expressed genes and identifies the upregulation of the “Senescence and autophagy” term (we have added emphasis to indicate this in the new Extended Data Fig. 4c) and downregulation of “Cell cycle” terms, both characterising senescence-associated phenotypes. Fig. 2i was manually curated using these gene sets, based on genes commonly associated with senescence in the literature¹⁹ that we believe an audience would find useful

as easily accessible information. We have clarified this in the legend. Fig. 2j-k represents the most notable examples from the unbiased analysis of MSigDB hallmarks. We have included entire hallmarks as new Extended Data Fig. 4d.

5. There are concerns about the extensive reliance on pseudotime for many of the conclusions in this manuscript when it has been shown to be highly sensitive to the embedding and not always the optimal method for a given dataset. Evaluation of the robustness of the findings across multiple trajectory inference methods, such as *dynverse*, would better support the data.

We thank the reviewer for their suggestion of the *dynverse* package that allows for comparing multiple trajectory inference methodologies²⁰, including pseudotime/Monocle which was used in our original analysis.

For the tumorigenesis dataset presented (Fig. 3), we have now included trajectory inference results from 3 additional methods, *MST*, *Slingshot* and *Sincell* (new Extended Data Fig. 7a). These were selected from the available methods in *dynverse* using the inbuilt guidelines for recommended methods given the parameters of the particular dataset. While there are the expected differences in data appearance due to the different embeddings used, our original conclusions as presented in the paper are conserved. To summarise, these are:

1. There are 2 distinct clusters of high RAS-expressing cells, one consistent with OIS (expresses markers such as p15, and restricted to early time points) and the other consistent with our “Notch-branch” tumours (also previously described in Hoare 2015).
2. There is a further separate cluster of lower RAS-expressing cells, present at the tumour timepoint, expressing hepatocytic markers including Dlk1. This is consistent with our “Dlk-branch” tumours described in the rest of the paper.

Notably, the mutual exclusivity of these branches (Dlk1 vs. Notch1) was validated by immunohistochemistry (IHC) and therefore conclusions are not solely drawn from the pseudotime analysis. Furthermore, these disparate branches are highly correlative with tumours with different histopathological properties. These additional trajectory inference results reinforce the key takeaway from the pseudotime analysis, the existence of disparate types of tumour-initiating cells (TICs). The new results can be found in Extended Data Fig. 7a, and we have edited the main text accordingly. We thank the reviewer for this helpful suggestion.

Referee #4 (R4):

The study from Adelyne Chan et al., investigates the effects of different levels of oncogenic RAS on cellular senescence phenotypes. The researchers developed in vitro and in vivo models to examine the impact of varying RAS levels. In the liver model, different levels of RAS led to distinct hepatocyte clusters with characteristics of either senescence or progenitor-like cells. When RAS levels were decreased, the hepatocytes became immune-resistant and formed tumors. Time-series analysis identified two types of tumors: early-onset aggressive undifferentiated and late-onset differentiated hepatocellular carcinoma. Similar trends were observed in a pancreatic tumor model. This elegant study suggest that fine-tuning oncogenic RAS levels may have implications for cancer development and therapeutic interventions.

Major limitations of the study are the poor report of the number of animals used, lack of depth in senescence and cancer-related characterizations and superficial description statistical analysis

More specifically:

1. Sample size and statistical analysis: The result section does not provide information about the sample size or statistical analysis performed in the study. The text briefly mentions control groups but does not provide detailed information about their criteria.

We apologise for the lack of clarity. We have included the information about sample size and statistical analysis in the main text, legends and Methods. We have also added a 'Statistical analysis' section in the Methods. We thank the reviewer for pointing this out.

2. The text briefly mentions the implications of the findings in early tumorigenesis but does not provide a thorough discussion of the potential future directions and practical implications of the study. Potential therapeutic targets within the Ras/MAPK signaling pathway in HCC should be considered. It would be valuable to include experiments that test a therapeutic strategy targeting this pathway and how it may impact subclonal fitness and tumor progression.

We agree this is a very important point. We indeed expect that our findings have clinical implications, particularly in early diagnosis and cancer prevention.

Our new results (e.g. **Reviewer fig. 1**) suggest that NOTCH1-positive hepatocytes are more immunogenic but resistant to immune surveillance (see our response to R2-#1, #5). Interestingly, we previously showed that inhibiting Notch signalling by ectopic expression of a dominant negative form of co-activator mastermind1 (dnMAML1) promotes OIS surveillance². Thus, it is possible that Notch inhibition in a high-RAS context might divert cells toward a less harmful OIS state.

In addition, this reviewer raised an interesting point: considering our data linking RAS dosage and cell fates, early intervention of the RAS-MAPK pathway might modulate the tumour-initiating process. For this, we decided to use Sorafenib, a clinically approved first-line drug for advanced HCC. This multi-kinase inhibitor disrupts the RAS/RAF/MEK/ERK cascade by targeting RAF and a number of upstream receptor tyrosine kinases³.

To ask whether this modality can be applicable to the preneoplastic stage, we treated mice with Sorafenib every two days for 30 days following a sub-OIS dosage of NRAS^{G12V} (UBC-driven mVenus-P2A-NRAS^{G12V}) injection. At day 30 following NRAS^{G12V} hydrodynamic tail vein injection (HDTV), mice were culled for histology (Extended Data Fig. 8d). As expected, pERK staining was substantially reduced by the Sorafenib treatment. We found no difference in the abundance of hepatocytes expressing the transgene, thus lowering the RAS-MAPK signalling did not affect immune surveillance. Remarkably, however, the F4/80 macrophage accumulation, which was associated with Notch1-type TICs (please also see our responses to R1-#1, #4, R2-#5), was significantly reduced by the treatment with no alteration (or an increase if any) in the abundance of Dlk1-positive hepatocytes (Extended Data Fig. 8c).

Considering that the RAS level was generally higher in Notch1-type than Dlk1-type hepatocytes, we speculate that prophylactic Sorafenib, or inhibition of the RAS-MAPK pathway, might reduce the inflammatory burden, leading to a more benign state, much like Dlk1-type.

Together, an early intervention of the key signatures identified in our study may have tumour preventative potentials: i) targeting Notch signalling in high-RAS context to improve OIS surveillance and ii) targeting RAS-MAPK signalling to reduce Notch1-type TICs/myeloid accumulation.

Since the IHC staining for NOTCH+ regions in human cirrhotic livers is highly robust, we envisage that our study will open up a new avenue for a focused investigation of the NOTCH1+ niche (a potential TIC candidate for the aggressive liver tumours) in human livers in the future. We thank the reviewer for this suggestion.

The new Sorafenib data can be found in Extended Data Fig. 8d, and we have provided a discussion on the potential for early intervention of the TICs.

3. Given that liver cancer affects males and females differently, it would be valuable to include a gender-specific analysis or speculation, to explore potential differences in the impact of RAS signaling and its role in tumorigenesis. This could involve stratifying the human data by gender and examining the relationship between RAS signaling, senescence, and tumor initiation separately for males and females. By comparing the molecular pathways activated by RAS and their consequences on senescence and tumor initiation in both sexes, the finding could shed light on the gender-specific factors that contribute to liver cancer development.

As the reviewer points out, the sex disparity in primary liver cancer is well-documented. Thus, it would be very interesting to know whether there are any sex biases in the preneoplastic stage. Historically, the hydrodynamic tail vein injection (HDTV_i) technique, which involves rapid bolus injection of a high volume of liquid (10% of body weight), has been mainly utilised in female mice due to the technical challenge posed by the higher body weight of male mice. While we are still accumulating our experience in male mice HDTV_i, we have conducted a pilot cohort experiment in both sexes using UBC-*NRAS*^{G12V} construct. We found no significant difference in the abundance of *NRAS*^{G12V}-expressing hepatocytes at day 12, thus at least the immune resistance of a low level of oncogenic RAS-expressing cells is conserved in male animals. We will follow this up with in-depth analyses of the sex-dependent TIC potential and associated immune cell profiling in the future. We thank the reviewer for this suggestion.

[REDACTED]

We have also stratified the human data shown in Fig. 4d by sex: Interestingly, the cancer genes that are associated with the two TIC branches might be more male-dominated. However, notably, the numbers of patients at risk are fewer in female patients, thus, the data is probably inconclusive. We hope we can expand the analysis in the future for this question.

[REDACTED]

Minor points:

Text Line 320. Not clear sentence-please reformulate

“spatial mutual exclusivity between them was observed”: now this sentence has been changed to as follows:

*“Notably, 9 patients exhibited positive staining for both markers **in spatially different regions.**”*

Text Line 322. This sentence is too speculative, given that cirrhosis doesn't develop from RAS mutations.

We agree with this point, and we have removed the last phrase, “they may later progress into distinct types of tumours”.

Reviewer References

1. Miyajima, A., Tanaka, M. & Itoh, T. Stem/Progenitor *Cells* in Liver Development, Homeostasis, Regeneration, and Reprogramming. *Cell Stem Cell* 14, 561–574 (2014).
2. Hoare, M. *et al.* NOTCH1 mediates a switch between two distinct secretomes during senescence. *Nat Cell Biol* 18, 979–992 (2016).

3. Keating, G. M. Sorafenib: A Review in Hepatocellular Carcinoma. *Target. Oncol.* 12, 243–253 (2017).
4. Gonçalves, S. et al. COX2 regulates senescence secretome composition and senescence surveillance through PGE2. *Cell Reports* 34, 108860 (2021).
5. Kang, T.-W. et al. Senescence surveillance of pre-malignant hepatocytes limits liver cancer development. *Nature* 479, 547–551 (2011).
6. Eggert, T. et al. Distinct Functions of Senescence-Associated Immune Responses in Liver Tumor Surveillance and Tumor Progression. *Cancer Cell* 30, 533–547 (2016).
7. Sugiura, R., Satoh, R. & Takasaki, T. ERK: A Double-Edged Sword in Cancer. ERK-Dependent Apoptosis as a Potential Therapeutic Strategy for Cancer. *Cells* 10, 2509 (2021).
8. Liberzon, A. et al. The Molecular Signatures Database Hallmark Gene Set Collection. *Cell Syst.* 1, 417–425 (2015).
9. Schulze, A., Oshi, M., Endo, I. & Takabe, K. MYC Targets Scores Are Associated with Cancer Aggressiveness and Poor Survival in ER-Positive Primary and Metastatic Breast Cancer. *Int. J. Mol. Sci.* 21, 8127 (2020).
10. Gonzalez, R. et al. Screening the mammalian extracellular proteome for regulators of embryonic human stem cell pluripotency. *Proc National Acad Sci* 107, 3552–3557 (2010).
11. Driskell, R. R. et al. Distinct fibroblast lineages determine dermal architecture in skin development and repair. *Nature* 504, 277–281 (2013).
12. Gorgoulis, V. et al. Cellular Senescence: Defining a Path Forward. *Cell* 179, 813–827 (2019).
13. Olan, I. & Narita, M. Senescence: An Identity Crisis Originating from Deep Within the Nucleus. *Annu Rev Cell Dev Bi* 38, 219–239 (2022).
14. Yin, K. et al. Senescence-induced endothelial phenotypes underpin immune-mediated senescence surveillance. *Gene Dev* 36, 533–549 (2022).
15. Whitfield, M. L. et al. Identification of Genes Periodically Expressed in the Human Cell Cycle and Their Expression in Tumors. *Mol. Biol. Cell* 13, 1977–2000 (2002).
16. Macosko, E. Z. et al. Highly Parallel Genome-wide Expression Profiling of Individual Cells Using Nanoliter Droplets. *Cell* 161, 1202–1214 (2015).
17. Butler, A., Hoffman, P., Smibert, P., Papalexi, E. & Satija, R. Integrating single-cell transcriptomic data across different conditions, technologies, and species. *Nat. Biotechnol.* 36, 411–420 (2018).
18. Tirosh, I. et al. Dissecting the multicellular ecosystem of metastatic melanoma by single-cell RNA-seq. *Science* 352, 189–196 (2016).
19. Salama, R., Sadaie, M., Hoare, M. & Narita, M. Cellular senescence and its effector programs. *Gene Dev* 28, 99–114 (2014).
20. Saelens, W., Cannoodt, R., Todorov, H. & Saeys, Y. A comparison of single-cell trajectory inference methods. *Nat Biotechnol* 37, 547–554 (2019).

Reviewer Reports on the First Revision:

Referees' comments:

Referee #2 (Remarks to the Author):

The review has made this manuscript much stronger. The authors have undertaken big efforts to reply to my comments as well to the comments of the other referees.

Both their experiments as well as their data analyses are now convincing also in respect to the spatial overview of this heterogeneity connecting RAS expression difference with specific targets or even immune cell status. Also a better visualization of the candidates – (e.g. by a heat map) was performed.

It was also nice to see their hypothesis reproduced in another in vitro model: In a mixed population of TIG3 fibroblasts with a wide range of HRASG12V dosage, the survival benefit of the low-RAS cells was recapitulated and high-RAS TIG3 cells show the classic senescence-like phenotype, including reduced proliferation, increased SA- β -gal activity, and prominent inflammatory secretory phenotype, such as IL6 and IL8.

Moreover, the authors characterized the immunological environment in relation to the DLK1 or Notch positive TICs. Finally, additional human samples were included. In summary, this has become a great manuscript that will be of high interest to the broad readership of Nature in general as well as the big audience interested in senescence and cancer.

Referee #3 (Remarks to the Author):

The authors have addressed all my comments.

Referee #4 (Remarks to the Author):

I thank the authors for some of the initial concerns, in particular in regards to reporting the statistical analysis. In contrast, the characteristics of the patients are still poorly reported, e.g. age, sex.

The study on Sorafenib at the preneoplastic stage can be quite informative for pre-clinical mechanistic studies. However, as the authors stated, Sorafenib is a first-line therapeutic drug for advanced HCC, with documented side effects that early-stage patients should not suffer if treated for prevention. To improve the clinical relevance and widen the scope of the study, less invasive experimental settings targeting Ras-MAPK pathway at the preneoplastic early stage should be used.

As the authors agreed, primary liver cancer such as HCC is sexually biased. While mouse weight is a crucial factor for the technical success of HDTV_i, it is surprising to see such a big sex disparity in weights at 6~8 weeks of age. Even though the study is heavily based on female mice, the authors should check more parameters in the male mice to validate their findings in both sex groups. The data in Reviewer Fig.4 does not seem to be sufficient to support this.

Author Rebuttals to First Revision:

Referee #1

The authors adequately addressed my comments and I am satisfied with the changes. This interesting study is now ready for publication.

Referee #2

The review has made this manuscript much stronger. The authors have undertaken big efforts to reply to my comments as well to the comments of the other referees. Both their experiments as well as their data analyses are now convincing also in respect to the spatial overview of this heterogeneity connecting RAS expression difference with specific targets or even immune cell status. Also a better visualization of the candidates – (e.g. by a heat map) was performed.

It was also nice to see their hypothesis reproduced in another in vitro model: In a mixed population of TIG3 fibroblasts with a wide range of HRASG12V dosage, the survival benefit of the low-RAS cells was recapitulated and high-RAS TIG3 cells show the classic senescence-like phenotype, including reduced proliferation, increased SA- β -gal activity, and prominent inflammatory secretory phenotype, such as IL6 and IL8.

Moreover, the authors characterized the immunological environment in relation to the DLK1 or Notch positive TICs. Finally, additional human samples were included. In summary, this has become a great manuscript that will be of high interest to the broad readership of Nature in general as well as the big audience interested in senescence and cancer.

Referee #3

The authors have addressed all my comments.

Reviewers #1, #2 and #3 have no further questions. We thank them for their supportive comments.

Referee #4

I thank the authors for some of the initial concerns, in particular in regards to reporting the statistical analysis. In contrast, the characteristics of the patients are still poorly reported, e.g. age, sex.

The cirrhosis patient demographics information was summarised in Supplemental Table 3, which contains the age, sex, aetiology of liver disease, co-morbidities, body mass index, smoking status and liver function tests. However, to improve the granularity of the data presented, we have added the patient ethnicity and platelet count.

In Extended Data Fig. 10a, we provided the frequency of NOTCH and/or DLK1-positive hepatocytes in liver cirrhosis patients examined. As expected for chronic liver disease, the number of female patients is quite low (6 out of 28 patients), thus the frequency of NOTCH and/or DLK1-positive hepatocytes in individual sexes is not very informative. Instead, we also have added this information to the demographic table (new Supplemental Table 3).

The study on Sorafenib at the preneoplastic stage can be quite informative for pre-clinical mechanistic studies. However, as the authors stated, Sorafenib is a first-line therapeutic drug for advanced HCC, with documented side effects that early-stage patients should not suffer if

treated for prevention. To improve the clinical relevance and widen the scope of the study, less invasive experimental settings targeting Ras-MAPK pathway at the preneoplastic early stage should be used.

The reasons for the choice of Sorafenib in our experiment include: i) it is already approved in patients with HCC, and ii) it is a multi-kinase inhibitor, including RAF and upstream kinases. The latter point is potentially useful for dealing with feedback effects that RAF/MAPK inhibitors might cause.

The current study focuses on a new paradigm, the senescence spectrum, which opens up unique opportunities in cancer interception and our Sorafenib data in the mouse liver model represents an exciting proof-of-concept for cancer prevention.

As mentioned below, we are currently developing the idea of cancer drug 'repurposing' for cancer prevention as a main follow-up study, which will include testing less invasive therapeutic regimens with a focus on reducing side effects, as the reviewer suggests. We believe this is beyond the scope of the current manuscript.

Future Direction: Cancer drug 'repurposing' for cancer prevention

As the reviewer indicates, our research focuses on early intervention for tumour prevention and the side effects of treatments should be minimal. Sorafenib does have reported side effects, such as skin reactions, diarrhoea/anorexia, and fatigue, but **Sorafenib is typically administered as a chronic treatment for advanced HCC patients as long as the patients tolerate it and it remains effective.**

However, for preventative treatments, we envision shorter, transient, and sporadic administration to modulate cell-fate decisions or targeted elimination of pre-neoplastic senescent/tumour-initiating cells. In our experiment (Extended Data Fig. 8d), mice were treated with Sorafenib for one month, already resulting in a significant reduction in immune cell clustering. We suspect that even shorter durations of treatment may be effective.

Sorafenib is currently only used for advanced-stage patients in HCC. We are excited about the possibility of repurposing such drugs for cancer prevention.

This has implications beyond the RAS/MAPK pathway, and our current study and datasets identified other possible targets, either separately or in combination. One example is NOTCH signalling: NOTCH inhibitors are promising anti-cancer drugs, but their on-target side effects (such as severe diarrhoea) hinder long-term administration. Interestingly, we previously showed that genetic NOTCH inhibition in RAS-expressing hepatocytes promotes senescence surveillance (Hoare et al. Nat Cell Biol. 2016: PMID 27525720). Thus, it is possible that lower dosages and shorter durations of NOTCH inhibitors (**thus minimising potential side effects**) might reduce the liver cancer risk via diverting toward the senescence route and promoting senescence surveillance.

These points were briefly touched on in the manuscript and we hope to systematically investigate this idea in the future.

As the authors agreed, primary liver cancer such as HCC is sexually biased. While mouse weight is a crucial factor for the technical success of HDTV_i, it is surprising to see such a big sex disparity in weights at 6~8 weeks of age. Even though the study is heavily based on female mice, the authors should check more parameters in the male mice to validate their findings in both sex groups. The data in Reviewer Fig.4 does not seem to be sufficient to support this.

As we described in our previous responses, the weight difference between male and female mice is a crucial factor for the success of the hydro-dynamic tail vein injection (HDTV_i) technique: this is the main reason for the female bias in this study. The bar graph shows the evidence for the significant weight difference between sexes of 8w-old C57BL/6 mice we have used for HDTV_i experiments at the CRUK Cambridge Institute animal facility. This is highly consistent with the data available for C57BL/6 at the Jackson Lab website: <https://www.jax.org/jax-mice-and-services/strain-data-sheet-pages/body-weight-chart-000664>

Nevertheless, we are actively engaged in utilising male mice for this technique. In our previous responses, we showed no significant difference in the abundance of ectopic NRAS^{G12V}-expressing hepatocytes between sexes at day 12 in the UBC (low-RAS) cohort. Now, as shown here, our ongoing long-term monitoring of this cohort confirms the tumorigenic activity of low-RAS in male mice and suggests that there is no difference in tumour development between male and female mice. Therefore, we feel that further monitoring and additional assays are unlikely to provide significant insights **in this particular context**.

The sex disparity in HCC represents a fundamental question but is highly elusive, **likely intertwined with other risk factors, such as age and background chronic diseases**. We are currently designing a follow-up study utilising our recently developed age-accelerated mouse models (Cassidy et al. Nat Commun 2020: PMID 31949142). These models comprise two variations: i) age acceleration with chronic liver inflammation/fibrosis and ii) age acceleration without apparent liver disorders. We will characterise low-RAS-mediated liver tumour initiation in these clinically more relevant models with various risk factors (sex disparity, ageing, and chronic liver diseases). **Given the complexity of the biological system, we view this as an independent research question.**

We have reinforced this point in our revised manuscript (at the end of the discussion).

“While our preclinical models are focused on young female mice, a better understanding of specific TICs and their microenvironments, along with other factors like sex, age, and background chronic liver diseases, may offer therapeutic insights for early intervention in tumorigenesis.”

We thank this reviewer for raising these inspiring questions for our future research directions.

Reviewer Reports on the Second Revision:

Referees' comments:

Referee #2 (Remarks to the Author):

The revised version by Chan et al, "Titration of RAS alters senescent state and influences tumour initiation" covers in its current form all main aspects that were asked by all 4 referees.

The aspect of sexual dimorphism is an important one and the authors also utilized male mice for the HDTV_i technology.

The authors showed no significant difference in the abundance of ectopic NRAS^{G12V}-expressing hepatocytes between female and male mice at day 12 (UBC(low-RAS) cohort).

Moreover, the authors describe/display a long-term monitoring of the same cohort, which confirms a "tumorigenic activity" of low-RAS in male mice, suggesting that there is no difference in tumour development between male and female mice.

The authors state in their letter of response "Therefore, we feel that further monitoring and additional assays are unlikely to provide significant insights in this particular context".

I would propose that the authors describe their long-term monitoring cohort in the manuscript, and integrate this point in the discussion.

Moreover, the authors could set this in the context with their points mentioned at the end of the discussion - this would clarify the current knowledge and what is still to be discovered in this context:

"While our preclinical models are focused on young female mice, a better understanding of specific TICs and their microenvironments, along with other factors like sex, age, and background chronic liver diseases, may offer therapeutic insights for early intervention in tumorigenesis."

Referee #4 (Remarks to the Author):

Thanks for the explanations to my previous comments. However, the point raised before remained open as no further experimental evidence was added

Author Rebuttals to Second Revision:

Referee #2 (Remarks to the Author):

The revised version by Chan et al, "Titration of RAS alters senescent state and influences tumour initiation" covers in its current form all main aspects that were asked by all 4 referees.

The aspect of sexual dimorphism is an important one and the authors also utilized male mice for the HDTV_i technology.

The authors showed no significant difference in the abundance of ectopic NRASG12V-expressing hepatocytes between female and male mice at day 12 (UBC(low-RAS) cohort).

Moreover, the authors describe/display a long-term monitoring of the same cohort, which confirms a "tumorigenic activity" of low-RAS in male mice, suggesting that there is no difference in tumour development between male and female mice.

The authors state in their letter of response "Therefore, we feel that further monitoring and additional assays are unlikely to provide significant insights in this particular context".

I would propose that the authors describe their long-term monitoring cohort in the manuscript, and integrate this point in the discussion session.

Moreover, the authors could set this in the context with their points mentioned at the end of the discussion - this would clarify the current knowledge and what is still to be discovered in this context:

"While our preclinical models are focused on young female mice, a better understanding of specific TICs and their microenvironments, along with other factors like sex, age, and background chronic liver diseases, may offer therapeutic insights for early intervention in tumorigenesis."

We thank Reviewer #2 for their further input. As suggested, we have added the survival curves from our long-term sex cohort to Extended Data Figure 10, demonstrating no significant differences in tumour formation between male and female mice in this model, and integrated a mention to this in the discussion.

Referee #4 (Remarks to the Author):

Thanks for the explanations to my previous comments. However, the point raised before remained open as no further experimental evidence was added.

As suggested by Reviewer #2 (please see our response above), we have included new data regarding sex differences and integrated this point into the discussion.